# UniZero: Generalized and Efficient Planning with Scalable Latent World Models

**Yuan Pu**[*]                                                                  *puyuan@pjlab.org.cn*
*Shanghai Artificial Intelligence Laboratory*

**Yazhe Niu**[*]                                                               *niuyazhe314@gmail.com*
*The Chinese University of Hong Kong*

**Zhenjie Yang**                                                             *jayyoung0802@gmail.com*
*Shanghai Jiao Tong University*

**Jiyuan Ren**                                                          *rjy22@mails.tsinghua.edu.cn*
*Shanghai Artificial Intelligence Laboratory*

**Hongsheng Li**[†]                                                             *hsli@ee.cuhk.edu.hk*
*The Chinese University of Hong Kong*

**Yu Liu**                                                                   *liuyuisanai@gmail.com*
*Shanghai Artificial Intelligence Laboratory*

**Reviewed on OpenReview:** *https://openreview.net/forum?id=Gl6dF9soQo*

## Abstract

Learning predictive world models is crucial for enhancing the planning capabilities of reinforcement learning (RL) agents. Recently, MuZero-style algorithms, leveraging the value equivalence principle and Monte Carlo Tree Search (MCTS), have achieved superhuman performance in various domains. However, these methods struggle to scale in heterogeneous scenarios with diverse dependencies and task variability. To overcome these limitations, we introduce UniZero, a novel approach that employs a transformer-based world model to effectively learn a shared latent space. By concurrently predicting latent dynamics and decision-oriented quantities conditioned on the learned latent history, UniZero enables joint optimization of the long-horizon world model and policy, facilitating broader and more efficient planning in the latent space. We show that UniZero significantly outperforms existing baselines in benchmarks that require long-term memory. Additionally, UniZero demonstrates superior scalability in multitask learning experiments conducted on Atari benchmarks. In standard single-task RL settings, such as Atari and DMControl, UniZero matches or even surpasses the performance of current state-of-the-art methods. Finally, extensive ablation studies and visual analyses validate the effectiveness and scalability of UniZero's design choices. Our code is available at https://github.com/opendilab/LightZero.

## 1 Introduction

Reinforcement Learning (RL) has emerged as one of the pivotal approaches for achieving artificial general intelligence (AGI). Despite significant advancements in this field, traditional RL methods often struggle with complex tasks (Ni et al., 2024; Samsami et al., 2024). To address this limitation, researchers have focused on developing predictive world models to enhance the planning capabilities and sample efficiency (Janner et al., 2019; Hafner et al., 2020; Hansen et al., 2023; Assran et al., 2023; Sutton et al., 2022). Notably, MuZero-style

---

[1][*] These authors contributed equally to this work.
[2][†] Corresponding author.

algorithms (Schrittwieser et al., 2019; Antonoglou et al., 2021; Hubert et al., 2021b; Niu et al., 2024) leverage the *value equivalence principle* (Grimm et al., 2020) and Monte Carlo Tree Search (MCTS) (Świechowski et al., 2023) to facilitate planning within learned latent spaces, achieving exceptional performance in domains such as board games (Schrittwieser et al., 2019; Silver et al., 2017) and Atari (Schrittwieser et al., 2019). However, these achievements are largely restricted to settings requiring short-term memory and single-task learning (Ni et al., 2024; Hausknecht & Stone, 2017). Significant challenges still persist in scaling these methods to heterogeneous scenarios, such as environments requiring long-term memory, diverse action spaces, or multitask learning, which limits their applicability to broader and more complex domains.

In the fields of language and vision, the emergence of multi-head attention mechanisms (Vaswani et al., 2017) has fundamentally transformed the development of general-purpose foundation models. By leveraging large-scale and diverse datasets (Jia et al., 2024) in a simple next-token prediction framework, these mechanisms have driven significant advancements across a wide range of applications (Brown et al., 2020; Peebles & Xie, 2023). Recently, there has been growing interest in extending these techniques to decision-making domains. Notably, some studies (Chen et al., 2021; Janner et al., 2021; Reed et al., 2022) treat reinforcement learning as a sequence modeling problem, focusing on the supervised offline training of return-conditioned policies. Meanwhile, other works propose a two-stage online learning paradigm, where the policy and dynamics model are optimized independently (Hafner et al., 2023; Micheli et al., 2022; Robine et al., 2023).

Empirical studies (Parisotto et al., 2019; Ni et al., 2024) demonstrate the effectiveness of transformer-based architectures in capturing diverse *backward* memory capabilities within a unified, generalized framework. Similarly, Monte Carlo Tree Search (MCTS) is widely recognized for its efficiency in *forward* planning. Integrating these two paradigms offers a promising pathway for enhancing both retrospective and prospective cognitive functions in artificial intelligence. However, research on the seamless combination of transformers and MCTS remains sparse. This gap raises a critical question: *Can transformer architectures enhance the efficiency and scalability of planning in complex, heterogeneous decision-making tasks characterized by diverse dependencies and variability?* This paper represents an initial exploration of this question.

We first systematically analyze the limitations of MuZero-style architectures using a representative Atari game under stacked or non-stacked input conditions, simulating varying dependency ranges required for optimal control. Our findings, summarized in Table 1 and Figure 1, reveal two core limitations inherent to MuZero-style algorithms. First, the recurrent design introduces an intrinsic *entanglement* between latent representations and historical information, resulting in a bottleneck that impedes efficient information propagation. This entanglement further complicates the integration of self-supervised regularization losses (elaborated in Section 2 and 3.1). Second, the architecture suffers from an *under-utilization* of trajectory data during training, which restricts its ability to fully exploit the accumulated experiential data. Consequently, these limitations reduce data efficiency and scalability, particularly in heterogeneous scenarios characterized by diverse dependencies and task variability. A comprehensive analysis and experimental validation of these limitations is presented in Section 3.1.

To tackle these challenges, we introduce *UniZero*, a novel framework depicted on the right part of Figure 1. UniZero leverages a transformer-based world model to efficiently learn a task-agnostic shared latent space by disentangling latent states from implicit latent histories (see Section 3.2). Specifically, UniZero integrates domain-specific *encoders* to map diverse inputs—such as images, proprioceptive data, and discrete or continuous actions—into a unified latent representation. The unified latent states and actions across varying time steps and tasks are then processed by a transformer *backbone*, enabling temporal and contextual modeling. To support decision-making, UniZero employs specialized *heads*, which model both latent dynamics (e.g., the predicted next latent state and reward) and decision-critical quantities (e.g., policy and value) conditioned on the latent history produced by the transformer backbone. This unified design allows for the *joint* optimization of long-horizon world models and policies (Schrittwieser et al., 2019; Eysenbach et al., 2022), effectively mitigating the inconsistencies inherent to two-stage learning frameworks (Hafner et al., 2023). With its unified and modularized architecture and training paradigm, UniZero holds the promise of becoming a scalable foundational model for decision-making in various heterogeneous scenarios.

To validate the effectiveness of UniZero, we conduct extensive experiments on the VisualMatch benchmark, which requires long-term memory. UniZero significantly outperforms multiple baseline algorithms across

various memory lengths, demonstrating its capability to model long-term dependencies. Additionally, we investigate UniZero's multitask learning capabilities on 26 Atari games, demonstrating its potential as a versatile and general-purpose agent. In standard RL tasks, such as Atari and DMControl environments, UniZero also achieves comparable performance, underscoring its broad applicability. Comprehensive ablation studies and visual analyses further affirm the effectiveness and scalability of UniZero's design choices.

The main contributions of this paper are summarized as follows:

- We identify key limitations in existing MuZero-style architectures and introduce *UniZero*, a unified and modular approach that addresses these shortcomings. UniZero leverages a transformer-based latent world model to learn a task-agnostic and shared latent representation, facilitating more robust and generalizable decision-making across diverse and heterogeneous environments.

- UniZero significantly outperforms existing baselines in benchmarks that require long-term memory. To the best of our knowledge, UniZero is **the first online MCTS-based agent** to achieve performance comparable to single-task settings on the full Atari 100K benchmark using a single model.

- UniZero achieves comparable results on standard RL benchmarks, including Atari and DMControl environments, rivaling leading algorithms. Extensive ablation studies and visual analyses validate the effectiveness and scalability of its design choices.

## 2  Background

**Reinforcement Learning** (RL) Sutton & Barto (2018) is a foundational framework for addressing sequential decision-making problems, typically formalized as Markov Decision Processes (MDPs). An MDP is defined by the tuple $\mathcal{M} = (\mathcal{S}, \mathcal{A}, \mathcal{P}, \mathcal{R}, \gamma, \rho_0)$, where $\mathcal{S}$ represents the state space, and $\mathcal{A}$ denotes the action space. The transition dynamics $\mathcal{P} : \mathcal{S} \times \mathcal{A} \times \mathcal{S} \to [0, 1]$ specify the probability of transitioning from one state to another given an action. The reward function $\mathcal{R} : \mathcal{S} \times \mathcal{A} \to \mathbb{R}$ assigns scalar rewards to state-action pairs. The discount factor $\gamma \in [0, 1)$ regulates the trade-off between immediate and future rewards, while $\rho_0$ defines the initial state distribution. The objective of RL is to derive an optimal policy $\pi^* : \mathcal{S} \to \mathcal{A}$ that maximizes the expected cumulative discounted return: $\pi^* = \arg\max_\pi \mathbb{E}_\pi \left[ \sum_{t=0}^\infty \gamma^t r_t \right]$. In many real-world scenarios, the Markov property is often violated, necessitating the use of Partially Observable Markov Decision Processes (POMDPs) Sondik (1971). POMDPs generalize MDPs by introducing an observation space $\mathcal{O}$, where the agent receives observations $o \in \mathcal{O}(s)$ that provide partial information about the underlying state. In environments characterized by long-term dependencies, optimal decision-making requires leveraging the observation history $\tau_{1:t} := (o_{1:t}, a_{1:t-1})$ Ni et al. (2024). To manage computational complexity, it is common to use a truncated history of length $H$ rather than the complete observation history. Consequently, policies and value functions are defined over this truncated history and are expressed as $\pi(a_t \mid \tau_{t-H+1:t})$ and $v(a_t \mid \tau_{t-H+1:t}) = \mathbb{E}_{\pi, \mathcal{M}} \left[ \sum_{i=t}^\infty \gamma^{i-t} r_i \mid \tau_{t-H+1:t} \right]$, respectively.

**MuZero** Schrittwieser et al. (2019) achieves superhuman performance in complex visual domains Bellemare et al. (2013) without requiring prior knowledge of the environment's dynamics. It combines MCTS with a learned model comprising 3 networks: ① *Encoder*: $s_t^0 = h_\theta(o_1, \ldots, o_t)$. At time step $t$ and hypothetical (also called recurrent/unroll) step 0 (omitting $t$ when clear), this network encodes past observations $(o_1, \ldots, o_t)$ into a *latent representation* (or equivalently latent state), which initializes the root node of MCTS. ② *Dynamics Network*: $\hat{r}^k, s^k = g_\theta(s^{k-1}, a^k)$. This network predicts the next latent state $s^k$ and reward $\hat{r}^k$ based on the current latent state $s^{k-1}$ and action $a^k$. ③ *Prediction Network*: $\mathbf{p}^k, v^k = f_\theta(s^k)$. Given a latent state $s^k$, this network outputs a policy $\mathbf{p}^k$ (action probabilities) and a value $v^k$. MuZero performs MCTS in the learned latent space, with the encoder generating the root node $s^0$. Each edge in the search tree stores statistics, including $N(s, a), P(s, a), Q(s, a), R(s, a), S(s, a)$, representing visit counts, policy, value, reward, and state transitions, respectively. The MCTS process consists of three phases: *Selection*, *Expansion*, and *Backup* (Appendix B.1.1). After search, the visit counts $N(s, a)$ at the root node $s^0$ are normalized to derive an improved policy $\pi$. An action is sampled from this policy for interaction with the environment. During training, MuZero optimizes the following end-to-end loss function, incorporating separate terms for policy,

value, and reward losses ($l^{\mathrm{p}}$, $l^{\mathrm{v}}$, and $l^{\mathrm{r}}$, respectively), along with a regularization term for weight decay:

$$l_t(\theta) = \sum_{k=0}^{K} \left[ l^{\mathrm{p}}(p_t^k, \pi_{t+k}) + l^{\mathrm{v}}(v_t^k, z_{t+k}) + l^{\mathrm{r}}(\hat{r}_t^k, r_{t+k}) \right] + c||\theta||^2. \tag{1}$$

Notably, MuZero and its variants (Danihelka et al., 2022; Antonoglou et al., 2021; Hubert et al., 2021b; Ye et al., 2021) exhibit two key characteristics: (1) during training, only the initial step of the sequence observation is used, (2) predictions at each time step rely on a *latent representation* obtained recursively. We refer to architectures adhering to these principles as *MuZero-style architectures*.

**Self-supervised Regularization.** The visualization of latent representations learned by the MuZero agent, as presented in de Vries et al. (2021), indicates that in the absence of a specific training objective to align latent representations with observations, *mismatches* may emerge between the latent representations $s_t^k$ predicted by the dynamics network and the *observation embeddings* $z_{t+k}$ (or equivalently $z_t^k$) generated by the encoder. These discrepancies make the planning process unstable. Moreover, since the primary training objective in RL is based on scalar rewards, this information can be insufficient, particularly in sparse-reward scenarios (Badia et al., 2020). In contrast, *observation embeddings*, typically encoded as compact tensors, provide richer training signals than scalars. Integrating auxiliary *self-supervised objectives* into MuZero to regularize the latent representations is crucial to improving sample efficiency and stability. Specifically, de Vries et al. (2021) proposed a contrastive regularization loss: $l^z(\theta) = \sum_{k=0}^{H} ||z_t^k - s_t^k||_2^2$ which penalizes the error between the *observation embeddings* $z_t^k = \mathrm{sg}\{h_\theta(o_t^k)\}$ and their corresponding dynamics predictions $s_t^k$. Inspired by the Sim-Siam framework (Chen & He, 2021), EfficientZero (Ye et al., 2021) introduced a self-supervised consistency loss (SSL), defined as the negative cosine similarity between the *projections* of predicted latent representations $s_t^k$ and the actual *observation embeddings* $z_t^k$.

## 3 UniZero

In this section, we begin by analyzing the two main limitations of MuZero-style architectures, as discussed in Section 3.1, especially in their ability to handle tasks that require capturing long-term dependencies. To address these limitations, in Section 3.2, we introduce a novel approach termed UniZero, which is fundamentally a modular latent world model. We provide a comprehensive description of its architectural design, along with the joint optimization procedure for both the model and the policy. In Section 3.3, we explore how to conduct efficient MCTS from a long-term perspective within the learned latent space. For more details on the algorithm's implementation, please refer to Appendix B.

### 3.1 Main Limitations in MuZero-style Architectures

As shown in left of Figure 1, during training, MuZero-style architectures encode *only* the initial observation $o_t$ (which may include stacked frames) and the entire sequence of actions (omitting actions when context is clear) into the first latent state. In effect, no additional information beyond the first step is explicitly provided. This design results in an *under-utilization* of trajectory data, particularly in scenarios with long trajectories or long-term dependencies. Additionally, MuZero employs *dynamics heads* to unroll several hypothetical steps during training, i.e., $\hat{r}^k, s^k = g_\theta(s^{k-1}, a^{k-1})$, as depicted in the dark gray section of Figure 1. Consequently, the recursively unrolled latent representation $s_t^k$ becomes tightly coupled with historical information, a phenomenon we term as *entanglement*. This entanglement is fundamentally incompatible with the SSL loss, as discussed later. During inference, MuZero encodes the current observation $o_t$ into a latent representation, which subsequently serves as the root node in the MCTS. While this approach is effective in MDPs, it is likely to fail in POMDP due to the lack of historical information in the root state. Another alternative extension is to employ the recursively predicted latent representation, $s_{t-k}^k$, as the root node. Here, $z_{t-k}$ (also denoted as $z_{t-k}^0$) represents the observation embedding obtained by encoding the true observation at time $t - k$. The state $s_{t-k}^k$ is then computed by recursively predicting $k$ steps forward using the dynamics network while following the sequence of actions in the trajectory. For clarity, the case of $k = 2$ is illustrated in Figure 1. However, this design introduces accumulative errors, ultimately leading to suboptimal performance. To systematically investigate the limitations of MuZero and the impact of different designs, we propose three variants of the MuZero-style algorithm and compare them with our UniZero architecture:

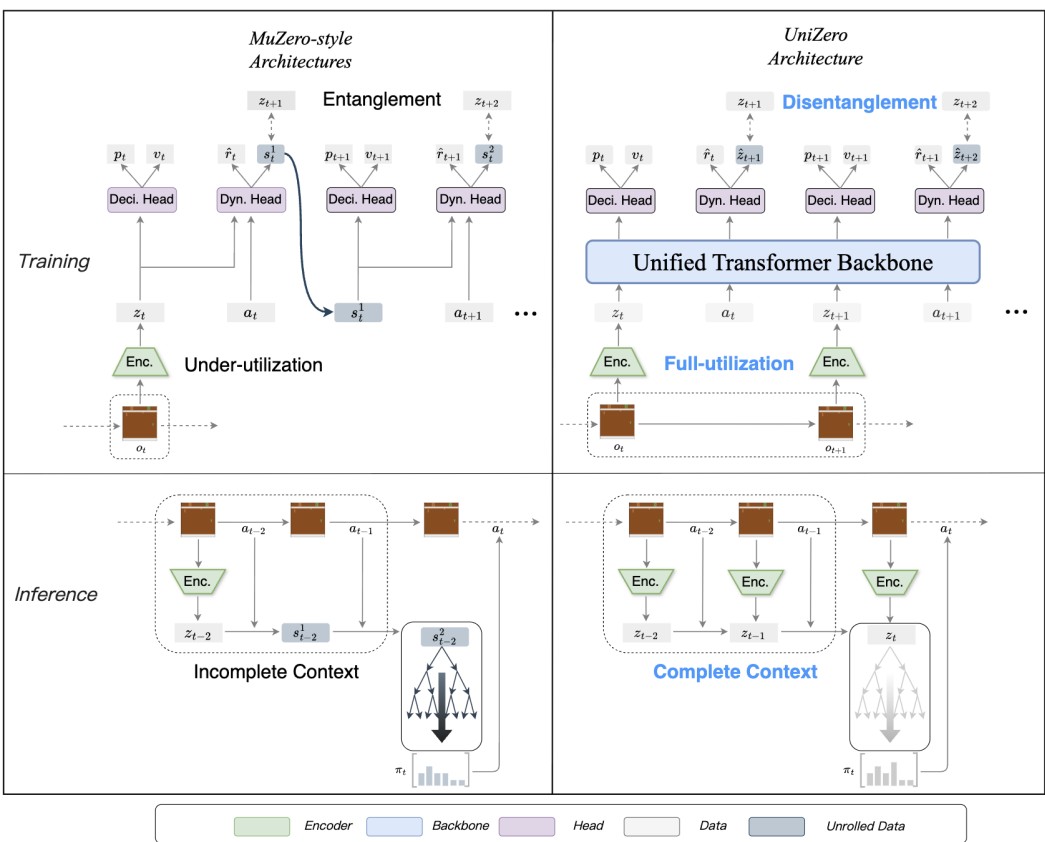

**Figure 1: Comparison between the *UniZero* (Ours) and *MuZero-style* architectures during training and inference. Left**: In the MuZero-style architecture, the *recursively* unrolled latent representation $s_t^k$ is tightly *entangled* with historical information. During training, it solely utilizes the initial observation of the sequence, resulting in inefficient utilization of information (*under-utilization*). During inference, the recursively predicted latent representation $s_{t-k}^k$ (with $k = 2$ for clarity) serves as the root node in MCTS, which is prone to inaccuracies due to accumulated errors. These issues are particularly pronounced in tasks requiring *long-term dependency modeling*. **Right**: UniZero employs a **modular latent world model** comprising an encoder, a unified transformer backbone, and decision/dynamics heads. This design explicitly *disentangles* latent states from implicit latent history and leverages all observations during training (*full-utilization*). During inference, the directly encoded latent state $z_t$ is used as the root node. By utilizing a more complete and accessible context $M = (z_{t-H_{\text{infer}}}, a_{t-H_{\text{infer}}}, \ldots, z_t, a_t)$, UniZero improves prediction accuracy and enables more effective long-term planning in the latent space.

- **(Original) MuZero**: This baseline *does not* employ any self-supervised regularization. During inference, the root latent state $z_t$ is generated by encoding only the current observation $o_t$ via the encoder.

- **MuZero w/ SSL**: As described in Section 2, this variant introduces an auxiliary self-supervised regularization loss during MuZero's training process. Theoretically, it aims to enhance sample efficiency while enabling the latent state to better retain historical information. However, the self-supervised objective focuses on accurately predicting subsequent latent states, which *deviates* from the fundamental requirement of the primary loss function—preserving the full historical context necessary for optimal decision-making. We refer to this misalignment as *incompatibility*. As a result, while this design performs well in MDP tasks, its effectiveness may be constrained in environments characterized by long-term dependencies or partial observability.

- **MuZero w/ Context**: This variant adopts the same training procedure as MuZero but modifies the inference phase by using a $k$-step *recursively* predicted latent representation, $s_{t-k}^k$, as the root node. However, the compounding prediction errors (Janner et al., 2019) inherent in recurrent unrolling result in

**Table 1: Qualitative comparison of MuZero variants.** The original *MuZero* employs only its first observation as input during training, which limits its ability to model long-term dependencies. *MuZero w/ SSL* incorporates a state regularization loss to enhance sample efficiency. Both *MuZero w/ Context* and *UniZero (RNN)* provide only partially accessible contexts due to their reliance on recurrent architectures. In contrast, *UniZero*, employing a transformer-based decoupled architecture, fully leverages the entire observation sequence during training and inference to ensure complete contextual accessibility.

| Algorithm | Obs. Full Utilization | State Regularization | Context Access |
|---|---|---|---|
| MuZero | ✗ | ✗ | ✗ |
| MuZero w/ SSL | ✗ | ✓ | ✗ |
| MuZero w/ Context | ✗ | ✗ | Partially Accessible |
| UniZero (RNN) | ✓ | ✓ | Partially Accessible |
| UniZero | ✓ | ✓ | Fully Accessible |

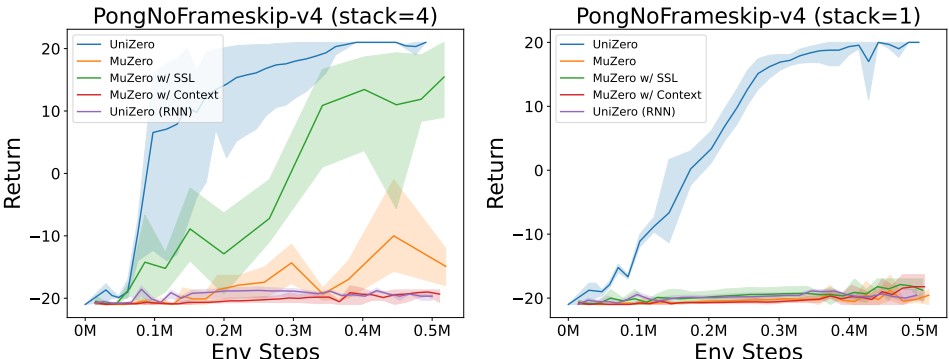

**Figure 2: Performance comparison of UniZero and MuZero variants in *Pong* under approximate MDP and POMDP settings. Left**: Results in the MDP setting. **Right**: Results in the POMDP setting. UniZero consistently outperforms all baselines across both scenarios, highlighting its robustness and adaptability. *MuZero w/ SSL* achieves superior sample efficiency in the MDP setting but fails to converge in the POMDP setting due to representation entanglement issues. Both *MuZero w/ Context* and *UniZero (RNN)* exhibit limited performance in both settings, primarily due to prediction errors caused by incomplete context representation.

an inaccurate root state. This issue, which we term as *incomplete context*, leads to significant degradation in MCTS accuracy and task performance.

- **UniZero**: Our proposed UniZero (illustrated on Figure 1) disentangles the latent states $z_t$ from the implicit latent history by leveraging a modular transformer-based world model. This architecture enables the model to fully utilize trajectory data during training while ensuring the prediction of the next latent state at every timestep. During inference, UniZero maintains a relatively complete historical context by employing a *Key-Value (KV) cache* mechanism over the most recent $H_{\text{infer}}$ steps. This design not only enriches learning with self-supervised regularization but also captures long-term dependencies effectively.

- **UniZero (RNN)**: This variant retains the same training scheme as *UniZero* but replaces the transformer backbone with a GRU (Chung et al., 2014). During inference, the GRU's hidden state is reset every $H_{\text{infer}}$ steps. However, due to the complexity of GRU training Kapturowski et al. (2019), this variant also suffers from the *incomplete context* problem. Further details are provided in Appendix B.2.

For a clearer comparison, Table 1 summarizes the qualitative differences among these variants. Furthermore, Figure 2 illustrates their performance in *Pong* under two settings: *frame_stack=4* and *frame_stack=1*, which approximately correspond to MDP and POMDP scenarios, respectively, as described in Hausknecht & Stone (2017). In the *stack4* setting, *MuZero w/ SSL* achieves good sample efficiency due to the auxiliary self-supervised regularization. However, in the *stack1* setting, it fails to converge within 500k environment

steps, largely due to the aforementioned entanglement issue. Similarly, both *MuZero w/ Context* and *UniZero (RNN)* struggle to learn effectively because of prediction errors arising from the *incomplete context* phenomenon. Specifically, for the former, these errors stem from the recurrent unrolling of latent states, while for the latter, they originate from the complexity of GRU training Kapturowski et al. (2019). In contrast, UniZero consistently outperforms all other variants, highlighting its robustness and adaptability in handling heterogeneous scenarios. Our preliminary experiments also evaluated a hybrid architecture that integrates historical and current observations via a Transformer-based history encoder to generate the initial latent state, while keeping all other training procedures identical to those of MuZero. However, its performance was poor, so we excluded it from later comparisons. We plan to revisit this variant in future research.

## 3.2 Scalable Latent World Models

Building on the insights outlined earlier, we propose the UniZero method to address the entanglement of latent representations with historical information and the under-utilization of trajectory data. This section describes the architecture of UniZero and details its training procedure for joint optimization of the model and policy.

### 3.2.1 Modular Latent World Models

**Architecture.** As shown in the top-right corner of Figure 1, UniZero is composed of four key components: the *encoders* $h_\theta$, the transformer-based *backbone*, the *dynamics head* $g_\theta$, and the *decision head* $f_\theta$. The *encoders* include both the observation and action encoders, but for simplicity, the term *encoder* refers solely to the observation encoder unless otherwise specified. Additionally, the transformer backbone is implicitly integrated into the dynamics and decision heads in certain contexts. Formally, at each time step $t$, the environmental observations and actions are denoted as $o_t$ and $a_t$, respectively. For discrete action spaces, $a_t$ may refer to an action embedding obtained via a learned embedding table, while for continuous actions, it is derived from a two-layer MLP. The latent state is represented as $z_t$, the predicted subsequent latent state as $\hat{z}_{t+1}$, and the predicted reward as $\hat{r}_t$. The policy logits and state value are denoted as $p_t$ and $v_t$, respectively. These outputs guide the MCTS procedure to enable regularized policy optimization (Grill et al., 2020). The UniZero world model $\mathcal{W}$ encompasses the following components:

$$
\begin{aligned}
&\text{Encoder:} && z_t = h_\theta(o_t) && \triangleright \text{ Maps observations to latent states} \\
&\text{Dynamics Head :} && \hat{z}_{t+1}, \hat{r}_t = g_\theta(z_{\leq t}, a_{\leq t}) && \triangleright \text{ Models latent dynamics and reward} \\
&\text{Decision Head :} && p_t, v_t = f_\theta(z_{\leq t}, a_{\leq t-1}) && \triangleright \text{ Predicts policy and value}
\end{aligned}
\tag{2}
$$

**Training.** Each time step is represented by two tokens: a latent state and an action. Details on data preprocessing and architecture are provided in Appendix B.2. The dynamics head predicts the subsequent latent state and reward, conditioned on the sequence of prior latent states and actions $(z_{\leq t}, a_{\leq t})$. Simultaneously, the decision head predicts the policy and value based on latent states and actions up to time steps $t$ and $t-1$, i.e., $(z_{\leq t}, a_{\leq t-1})$. In MuZero-style methods, the $k$-th latent state $s_t^k$ is *recursively* derived from the initial observation via a dynamic network. In contrast, UniZero employs a transformer backbone to model the latent history $h_t = \{h_t^z, h_t^{z,a}\}$ at each time step. This design explicitly separates the latent state $z_t$ from the latent history $h_t$, mitigating two key limitations of MuZero-like algorithms. Unlike prior approaches (e.g., Hafner et al. (2023)), UniZero does not include a decoder to reconstruct $z_t$ back into $\hat{o}_t$. While reconstruction losses are often used to regularize representations, our empirical results (see Section 4.5) suggest that omitting this decoding loss does not degrade performance. This supports the hypothesis that latent states only need to encode decision-relevant information, rendering reconstruction unnecessary for decision-making tasks.

**Inference.** During inference, UniZero's latent world model leverages the complete long-term memory stored in the KV cache alongside information from the current observation to produce accurate internal predictions as root and internal nodes in the tree search. This synergistic use of components significantly improves UniZero's scalability and efficiency. Further details are provided in Section 3.3.

### 3.2.2 Joint Optimization of Model and Policy

In this paper, our primary focus is on online reinforcement learning settings. Algorithm 1 presents the pseudo-code for the entire training pipeline. This subsection will present the core process of joint optimization of the model and policy (behavior). *UniZero* maintains a replay buffer $\mathcal{B}$ that stores trajectories $\{o_t, a_t, r_t, o_{t+1}, \pi_t\}$ (where $\pi_t$ is the MCTS improved policy, Section 3.3) and iteratively performs the following two steps:

1. **Experience Collection:** Collect experiences into the replay buffer $\mathcal{B}$ by interacting with the environment. Notably, the agent employs a policy derived from MCTS, which operates within the learned latent space.

2. **Model and Policy Joint Update:** Concurrent with data collection, UniZero performs joint updates on the decision-oriented world model, including the policy and value functions, using data sampled from $\mathcal{B}$.

The joint optimization objective for the model-policy can be written as:

$$\mathcal{L}_{\text{UniZero}}(\theta) \doteq \mathbb{E}_{(o_t, a_t, r_t, o_{t+1}, \pi_t)_0^{H-1} \sim \mathcal{B}} \Big[ \sum_{t=0}^{H-1} \Big( \beta_z \underbrace{\|\hat{z}_{t+1} - \text{sg}(\bar{h}(o_{t+1}))\|_2^2}_{\text{next-latent prediction}} + \beta_r \underbrace{\text{CE}(\hat{r}_t, r_t)}_{\text{reward prediction}} \\ + \beta_p \underbrace{\text{CE}(p_t, \pi_t)}_{\text{policy prediction}} + \beta_v \underbrace{\text{CE}(v_t, \hat{v}_t)}_{\text{value prediction}} \Big) \Big] \quad (3)$$

Note that we also maintain a soft target world model (Mnih et al., 2013) $\bar{\mathcal{W}} = (\bar{h}_\theta, \bar{g}_\theta, \bar{f}_\theta)$, which is an exponential moving average of current world model $\mathcal{W}$ 2. In Equation 3, $H$ is the training context length, sg is the `stop-grad` operator, CE denotes cross-entropy loss function, $\bar{h}(o_{t+1}) = \bar{z}_{t+1}$ is the target latent state generated by the target encoder $\bar{h}_\theta$, and $\hat{v}_t$ signifies the bootstrapped $n$-step TD target: $\hat{v}_t = \sum_{k=0}^{n-1}\{\gamma^k r_{t+k}\} + \gamma^n \bar{f}_\theta(z_{\leq t}, a_{\leq t-1})$. As the magnitudes of rewards across different tasks vary greatly, UniZero adopts reward and value predictions as discrete regression problems (Bellemare et al., 2017) and optimizes by minimizing the cross-entropy loss. $\pi_t$ represents the improved policy through MCTS shown in Section 3.3. We optimize the dynamics head to predict $\pi_t$, which essentially seems a policy distillation process. Compared to policy gradient methods (Zhang et al., 2023; Hafner et al., 2019; Schulman et al., 2017), this approach potentially offers better stability (Schrittwieser et al., 2019; Grill et al., 2020). The coefficients $\beta_z, \beta_r, \beta_p, \beta_v$ are constant coefficients used to balance different loss items. Inspired by Hansen et al. (2023), UniZero has adopted the *SimNorm* technique, which is implemented after the final layer of the encoder and the last component of the dynamics head that predicts the next latent state. Essentially, this involves applying the L1 norm constraint to regularize the latent state space. As detailed in Section 4.5, latent normalization has been empirically proven to be crucial for enhancing the stability and robustness of training.

### 3.3 MCTS in the Unified Latent Space

RL agents need a *memory M* (or equivalently *context*) to accurately model future in tasks that require long-term dependencies. To effectively implement this memory mechanism, as depicted in Figure 3 (for simplicity, we use 1 as the starting timestep in this figure), we establish a *KV Cache* (Ge et al., 2023) for the memory, denoted by: $KV_M = \{KV(z_{t-H}, a_{t-H}, \ldots, z_t, a_t)\}$. When the agent encounters a new observation $o_t$ and needs to make a decision, it first utilizes the encoder to transform this observation into the corresponding latent state $z_t$, which serves as the root node of the search tree. By querying the *KV Cache*, the keys and values from the recent memory $(z_{t-H_{\text{infer}}}, a_{t-H_{\text{infer}}}, \ldots, z_t, a_t)$ are retrieved for the transformer-based latent world model. This model recursively predicts the next latent state $\hat{z}_{t+1}$, the reward $\hat{r}_t$, the policy $p_t$, and the value $v_t$. The newly generated next latent state $\hat{z}_{t+1}$ functions as an internal node in the MCTS process. Subsequently, MCTS is executed within this latent space. Further details can be found in B.1.1. Upon completion of the search, the visit count set $\{N(z_t, a_t)\}$ is obtained at the root node $z_t$. These visit counts are then normalized to derive the *improved policy* $\pi_t$: $\pi_t = \frac{N(z_t, a_t)^{1/T}}{\sum_{b_t} N(z_t, b_t)^{1/T}}$. Here, $T$ denotes the temperature, which modulates the extent of exploration (Badia et al., 2020). Actions are then sampled from this distribution for interactions with the environment. After each interaction, we save the transition

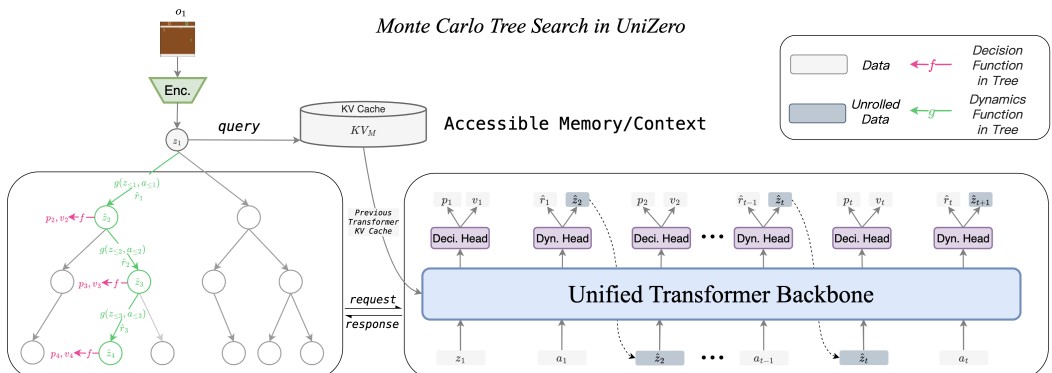

**Figure 3: MCTS in the learned latent space.** The process begins with a new observation $o_1$, which is encoded into a latent state $z_1$. This latent state serves as the root node. The previous keys and values of recent memory are retrieved from the transformer's KV Cache $KV_M$. Subsequently, the search tree utilizes the world model to predict the next latent state $\hat{z}$ (which serves as an internal node), reward $\hat{r}$, policy $p$, and value $v$, conditioned on the retrieved KV, recursively. These predictions are used to conduct MCTS, ultimately resulting in an improved policy $\pi$.

$(o_t, a_t, r_t, d_t, o_{t+1})$ along with the improved policy $\pi_t$ into the buffer, with the latter serving as the policy target in Eq. 3. By leveraging *backward memory* and *forward search*, UniZero demonstrates the potential to perform generalized and efficient long-term planning across a wide range of scenarios.

## 4 Experiments

To demonstrate the generality and scalability of UniZero, we conduct extensive evaluations across a diverse set of environments characterized by long-term and short-term dependencies, discrete and continuous action spaces, as well as single-task and multitask learning scenarios. Specifically, we evaluate UniZero on the Atari 100k benchmark (short-term dependency, discrete actions) (Bellemare et al., 2013; Kaiser et al., 2024), DMControl (short-term dependency, continuous actions) (Tunyasuvunakool et al., 2020), and VisualMatch (long-term dependency, discrete actions) (Ni et al., 2024). Through comprehensive experiments and in-depth analyses (Appendix E.1 and E.2), we aim to address the following key questions:

① How does UniZero perform on VisualMatch tasks that require long-term memory? (Section 4.2)
② In multi-task learning on Atari, how does UniZero compare to MuZero? Does it capture meaningful semantic information in the learned embeddings? (Section 4.2)
③ On single-task settings, can UniZero achieve performance on par with MuZero in the Atari 100k benchmark and the DMControl continuous control benchmark? (Section 4.4)
④ How effective and scalable are UniZero's core design choices? (Section 4.5)

### 4.1 Experimental Setup

**Environments and Baselines.** (1) **Atari 100k:** This benchmark comprises 26 Atari games, providing a diverse evaluation suite. The agent interacts for 100,000 steps (4 million frames with frame skipping of 4). Sometimes environment steps are abbreviated as *Env Steps*. (2) **DMControl:** We consider the Proprio Control Suite, including 18 continuous control tasks with a budget of 500k environment steps. Tasks include classical control, locomotion, and robotic manipulation. (3) **VisualMatch:** Designed to evaluate long-term dependencies, it tests memory through adjustable memory lengths. These grid-world tasks are divided into exploration, distraction, and reward phases, requiring the agent to recall an observed color in the exploration phase to select the correct color in the reward phase. Detailed task descriptions are provided in Appendix D.1. For the details of algorithm baselines, please refer to Appendix B.

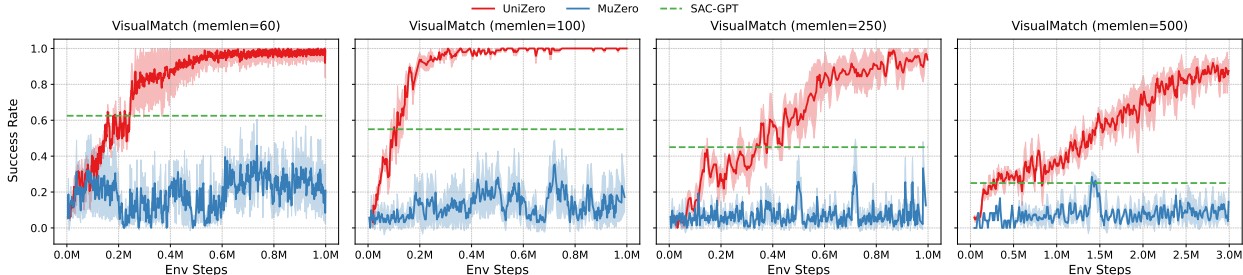

**Figure 4: Performance comparison on *VisualMatch* with increased memory lengths**. *MuZero* consistently underperformed across all tasks, primarily due to insufficient context information. The performance of *SAC-GPT* significantly deteriorated as the memory length increased. In contrast, *UniZero* maintained a high success rate even with extended memory lengths, demonstrating its superior capacity for modeling long-term dependencies. Solid lines denote means; shaded areas represent 95% confidence intervals.

## 4.2 Visual Match Benchmark

In Figure 4, we compare the performance of UniZero and MuZero on the *VisualMatch* benchmark, which requires long-term memory. The green horizontal dashed line represents the final success rate of SAC-GPT (Ni et al., 2024) after training on 3 million *environment steps*. Due to its lack of contextual information, MuZero performs poorly across all tasks, while SAC-GPT's performance degrades significantly as the memory length increases. In contrast, UniZero achieves consistently high success rates with increasing memory lengths due to its robust long-term dependency capabilities, validating the analysis presented in Section 3.1. Additional analysis of the predictions and attention maps of the trained world model is provided in Appendix E.2.

## 4.3 Multitask Learning on Atari Environments

**Table 2: Performance comparison of UniZero and MuZero across eight Atari environments in the different learning setting**. MT means multitask setting, ST means single-task setting. UniZero (MT) outperforms MuZero (MT) in most environments and achieves higher overall human normalized scores than UniZero (ST), demonstrating its scalability potential. The results on 26 Atari games can be found in Appendix 13.

| Algorithm | Alien | Boxing | Chopper | Hero | MsPacman | Pong | RoadRunner | Seaquest | *Mean* | *Median* |
|---|---|---|---|---|---|---|---|---|---|---|
| UniZero (MT) | 1003 | 5 | 3501 | 3003 | 989 | 19 | 6300 | 713 | **0.4554** | **0.4085** |
| MuZero (MT) | 590 | 1 | 1989 | 1999 | 999 | -1 | 5803 | 600 | 0.2192 | 0.0895 |
| UniZero (ST) | 580 | 3 | 2802 | 2991 | 1012 | 18 | 5503 | 750 | 0.3223 | 0.1739 |

UniZero's decoupled-yet-unified architecture proves highly effective across diverse environments with varying dependencies, enabling seamless extension to multitask learning scenarios. We first evaluate UniZero on eight Atari games: *Alien*, *Boxing*, *ChopperCommand*, *Hero*, *MsPacman*, *Pong*, *RoadRunner*, and *Seaquest*.

In the *multitask setting (MT)*, a single model is trained to perform all the considered tasks, where all tasks share a common observation space represented as a (3, 64, 64) image. To ensure consistency across tasks, we set full_action_space=True (Bellemare et al., 2013), which results in a unified action space of 18 discrete actions for each task. In the *single-task setting (ST)*, a separate model is trained independently for each task, with full_action_space=False (Bellemare et al., 2013), leading to task-specific action spaces. Unless otherwise specified, the multitask hyperparameters are consistent with those listed in Table 8, with only minimal adjustments to accommodate the larger model architecture. Specifically, the encoder uses num_channel=256, while the transformer backbone incorporates nlayer=12 and nhead=12.

**Architecture and Training in Multitask Learning.** Unlike single-task configurations, the multitask setup employs independent decision and dynamics heads for each task, following the approach of Kumar et al. (2022). This design introduces minimal additional parameters while preserving efficiency. The shared transformer backbone and encoder promote parameter reuse and generalization across tasks. Each task is

handled by a separate data collector that gathers trajectories and stores them in individual buffers. During training, we sample `task_batch_size=32` samples from different tasks, aggregate them into a minibatch, and apply the loss function defined in Equation 3 for each task. The task-specific losses are averaged to compute the total loss, which is used to perform backpropagation and network updates.

**Results.**   Table 2 and Figure 10 demonstrate that UniZero (MT) outperforms both UniZero (ST) and MuZero (MT) in terms of normalized mean and median scores across the evaluated environments within the 400K *Env Steps* setting. This result underscores the enhanced efficiency and scalability of UniZero as a latent world model for multitask training. To examine the influence of model size on multi-task learning performance, we analyze the effect of varying the transformer backbone size (nlayer=4, 8, 12) across eight Atari games (see Appendix Figure 11). Our findings reveal that increasing model size consistently improves sample efficiency across all tasks, highlighting UniZero's potential in multitask learning.

To further explore the benefits of multi-task learning, we conduct T-SNE visualizations of UniZero's latent states (Figure 12). The latent spaces exhibit distinct clustering for each game, reflecting the dynamic variations among environments. Notably, the representations of *Alien* are more dispersed, likely due to its similarity to other games, such as *MsPacman*, which belongs to the Maze category. This overlap may promote cross-task information sharing, contributing to the substantial performance improvements observed for *Alien* in the multi-task setting. Moreover, we extended our evaluation to full 26 games in Atai 100k benchmark in Appendix C. As shown in Figure 13 and Table 10, UniZero's multi-task model achieves normalized human scores comparable to its single-task version, whereas MuZero struggles with multi-task learning, showing little progress in most games. More analysis is provided in Appendix C.3.

## 4.4   Single Task Results in Non-Memory Domains

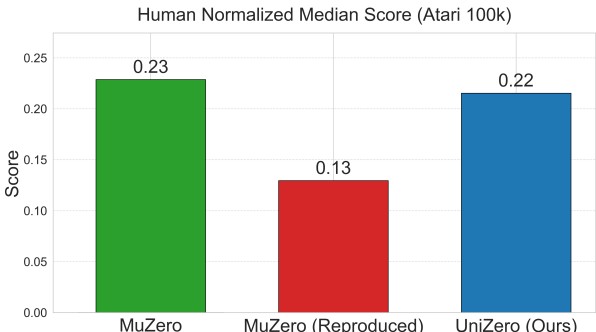

**Figure 5: Performance on the Atari 100K.** UniZero achieves a higher human-normalized median score compared to MuZero (Reproduced), demonstrating its ability to effectively model short-term dependencies. Detailed scores and curves are available in Appendix D.2.

**Atari**   We also compare the original MuZero algorithm (Schrittwieser et al., 2019), our reproduced MuZero, and UniZero, on the Atari 100K benchmark in the single-task setting, as illustrated in Figure 5. Our results show that UniZero achieves a higher human-normalized median score compared to MuZero (Reproduced) with the same code implementation framework, which indicates that UniZero effectively models short-term dependencies and demonstrates its versatility across discrete action decision-making tasks. To provide further insights, we present the complete scores and learning curves for 26 games in Appendix D.2. Additional analysis of the predictions and attention maps of the trained world model on the *Pong* game is provided in Appendix E.2.

## 4.5   Ablation Study

This subsection and Appendix E.1 evaluate key UniZero design choices:

- **Model Size**: Varies *num_layers* with training context lengths ($H = 5, 10, 20, 40$), keeping inference context length at $H_{\text{infer}} = 4$ to handle POMDP in Atari (Mnih et al., 2013).

- **Latent Normalization**: Comparison of normalization techniques that employed in the latent state, including *SimNorm* (Hansen et al., 2023), *Softmax*, and *Sigmoid*. Details can be found in Appendix B.

- **Decode Regularization**: Integrate a decoder on top of the latent state: $\hat{o}_t = d_\theta(z_t)$ (where $z_t = h_\theta(o_t)$) with additional training objective: $\mathcal{L}_{\text{decode\_reg}} = \|o_t - d_\theta(z_t)\|_1 + \mathcal{L}_{\text{perceptual}}(o_t, d_\theta(z_t))$, the first term is the reconstruction loss, and the second term is the perceptual loss (Micheli et al., 2022).

| Task | UniZero | DreamerV3 |
|---|---|---|
| acrobot-swingup | **400.3** | 154.5 |
| cartpole-balance | 952.2 | **990.5** |
| cartpole-balance__sparse | **1000.0** | 996.8 |
| cartpole-swingup | 801.3 | **850.0** |
| cartpole-swingup__sparse | **752.5** | 468.1 |
| cheetah-run | 517.6 | **585.9** |
| ball__in__cup-catch | **961.6** | 958.2 |
| finger-spin | 810.7 | **937.2** |
| finger-turn__easy | **1000.0** | 745.4 |
| finger-turn__hard | **884.5** | 841.0 |
| hopper-hop | **120.5** | 111.0 |
| hopper-stand | **602.6** | 573.2 |
| pendulum-swingup | **865.6** | 766.0 |
| reacher-easy | **993.3** | 947.1 |
| reacher-hard | **988.8** | 936.2 |
| walker-run | 587.9 | **632.7** |
| walker-stand | **976.4** | 956.9 |
| walker-walk | **954.6** | 935.7 |
| Mean | **787.2** | 743.7 |
| Median | **875.1** | 845.5 |

**DMControl**  UniZero leverages the principles of Sampled Policy Iteration (Hubert et al., 2021b), allowing for a seamless extension to continuous action spaces. Training details are provided in Appendix B. We evaluate UniZero on 18 tasks from the Proprio Control Suite in DMC, which include continuous control tasks with low-dimensional inputs and a budget of 500,000 environment steps. When compared against the state-of-the-art DreamerV3 (Hafner et al., 2023), UniZero demonstrates superior performance, achieving a higher human-normalized score, and thus showcasing its robust potential in handling diverse action spaces. Detailed learning curves for the DMC Proprio Control Suite are provided in Appendix D.3.

**Table 3: Performance between UniZero and DreamerV3 across various tasks in the DMControl.** The best performance for each task is indicated in **bold**. UniZero's higher human-normalized scores highlight its strong performance across continuous action spaces. Details are available in Appendix D.3.

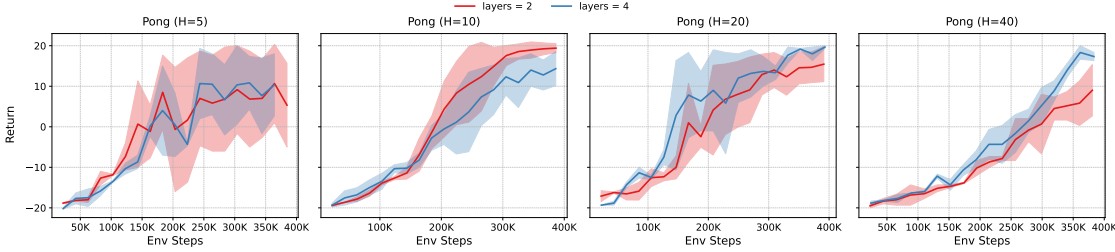

**Figure 6: Effect of model size across training context lengths ($H = 5, 10, 20, 40$) with fixed inference context length ($H_{\mathbf{infer}} = 4$) on Pong.**

Figure 6 and Figure 7 show different performance impacts for several ablation designs on *Pong* and *VisualMatch*. Additional results for *VisualMatch* are in Appendix E.1. Based on these results, we can conclude the following key findings: (1) **Training Context Length ($H$)**: Longer $H$ doesn't always improve performance, likely due to MCTS inference errors. In *Pong*, larger $H$ needs more layers to maintain performance. Consistent with prior work (Fang & Stachenfeld, 2023), longer contexts aid representation learning if prediction remains accurate. (2) **SimNorm**: Outperforms *Softmax* and *Sigmoid*, emphasizing effective latent normalization for stable training by enforcing sparsity through fixed L1 norm. (3) **Decode Regularization**: Negligible effect on both settings, indicating decision-relevant latent information matters more than observation reconstruction. Further details and visualizations are available in Appendix E.1 and Appendix E.2.

## 5 Related Work

**MCTS-based RL.** Algorithms like AlphaGo and AlphaZero (Silver et al., 2016; 2017), which combine MCTS with deep neural networks, have significantly advanced board game AI. Extensions such as MuZero (Schrittwieser et al., 2019), Sampled MuZero (Hubert et al., 2021b), and Stochastic MuZero (Antonoglou et al., 2021) have adapted this framework for environments with complex action spaces and stochastic dynamics. EfficientZero (Ye et al., 2021) and GumbelMuZero (Danihelka et al., 2022) have further increased the algorithm's sample efficiency. MuZero Unplugged (Schrittwieser et al., 2021; Xuan et al., 2024) introduced *reanalyze* techniques, enhancing performance in both online and offline settings. LightZero (Niu et al., 2024) addresses real-world challenges and introduces a open-source MCTS+RL benchmark. Studies like RAP

**Figure 7: Effect of Latent Normalization and Decode Regularization on Pong and VisualMatch (memlen=60).** *SimNorm* consistently outperforms *Softmax* and *Sigmoid*, emphasizing the critical role of proper normalization in ensuring training stability. Decode regularization exhibits a minimal effect on performance.

(Hao et al., 2023) and SearchFormer (Lehnert et al., 2024) have applied MCTS to enhance the reasoning capabilities of language models (Brown et al., 2020). We analyze the challenges MuZero faces in modeling long-term dependencies in POMDPs and propose a transformer-based latent world model to address them.

**World Models.** The concept of *world models*, as discussed in Schmidhuber (2015); Chiappa et al. (2017); Ha & Schmidhuber (2018), enables agents to predict and plan future states by learning a compressed spatiotemporal representation. Subsequent research (Hafner et al., 2023; Micheli et al., 2022; Robine et al., 2023; Zhang et al., 2023; Hansen et al., 2023; Schrittwieser et al., 2019) has enhanced world models in both architecture and training paradigms. These studies generally follow three main routes based on *training paradigms*: (1) The Dreamer series (Hafner et al., 2020; 2023) adopts an *actor-critic paradigm*, optimizing policy and value functions based on internally simulated predictions. Note that the model and behavior learning in this series are structured in a two-stage manner. Building on this, Micheli et al. (2022); Robine et al. (2023); Zhang et al. (2023) leverage Transformer-based architectures to enhance sequential data processing, achieving significant sample efficiency and robustness. (2) The TD-MPC series (Hansen et al., 2022; 2023) demonstrates substantial performance gains in large-scale tasks by learning policies through local trajectory optimization within the latent space of the learned world model, specifically utilizing the *model predictive control* algorithm (Kouvaritakis & Cannon, 2016). The model and behavior learning in this series also follow a two-stage structure. (3) Research stemming from MuZero (Ye et al., 2021; de Vries et al., 2021), grounded in the *value equivalence principle* (Grimm et al., 2020), achieves *joint optimization of the world model and policy* (Eysenbach et al., 2022; Ghugare et al., 2022a) and employs *MCTS* for policy improvement. Despite these advancements, the effective integration of these approaches remains under-explored. In our paper, we provide a preliminary investigation into integrating *scalable architectures* and *joint model-policy optimization* training paradigms. A detailed qualitative comparison is presented in Appendix 12.

## 6 Conclusion and Future work

In this paper, we investigate the efficiency of MuZero-style algorithms in heterogeneous scenarios characterized by diverse dependencies and task variability. Through qualitative analysis, we identify two key limitations of MuZero stemming from its recurrent training paradigm. To address these limitations and enhance the scalability of MuZero, we propose *UniZero*, a modular approach that integrates a transformer-based latent world model with MCTS. Experimental results demonstrate that UniZero consistently outperforms baseline methods across a wide range of settings, including discrete and continuous control, single-task and multi-task learning, as well as short- and long-term dependency modeling. Moreover, UniZero exhibits strong potential as a foundational model for large-scale multi-modal, multi-task learning, paving the way for exciting future research directions, which we aim to explore further.

## 7 Acknowledgements

This project is supported in part by the Centre for Perceptual and Interactive Intelligence (CPII) Ltd under the Innovation and Technology Commission (ITC)'s InnoHK, and in part by NSFC-RGC Project N_CUHK498/24. Hongsheng Li is a PI of CPII under the InnoHK. We extend our gratitude to several team-members of the Shanghai AI Laboratory and SenseTime for their invaluable assistance, support, and feedback on this paper and the associated codebase. In particular, we would like to thank Chunyu Xuan, Ming Zhang, and Shuai Hu for their insightful and inspiring discussions at the inception of this project.

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

# A    Environment Details

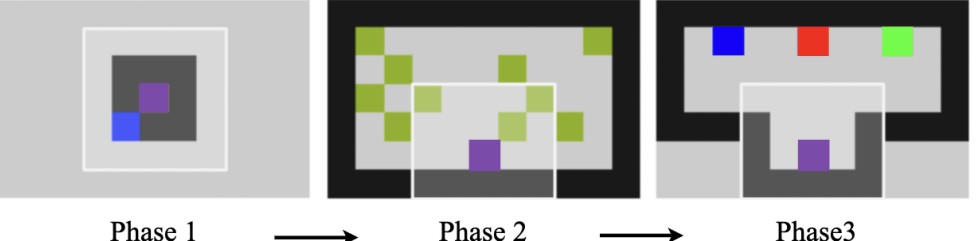

**Figure 8: VisualMatch Long-Term Dependency Benchmark**: Each task is segmented into three distinct phases—*Exploration*, *Distraction*, and *Reward*. As illustrated, the duration of Phase 2 (*Distraction*) varies and is denoted by the parameter *memory_length*, while the durations of Phases 1 and 3 remain constant. The agent (depicted in purple) operates within a partially observable Markov decision process (POMDP) environment, restricted to a 5×5 grid field of view outlined by white borders and impeded by black walls. During the *Exploration Phase* (Phase 1), the agent observes a room with a randomly assigned target color (e.g., blue). In the subsequent *Distraction Phase* (Phase 2), the environment introduces random distractions such as green apples. Finally, in the *Reward Phase* (Phase 3), the agent must navigate to the grid corresponding to the initially observed target color. Increasing the *memory_length* intensifies the requirement for the agent to retain and utilize long-term dependencies to successfully complete the task.

## A.1    VisualMatch Long-Term Dependency Benchmark

The **VisualMatch** benchmark is meticulously designed to evaluate an agent's capacity for handling long-term dependencies with adjustable memory lengths. As depicted in Figure 8, each task within this benchmark is structured as a grid-world environment and is divided into three sequential phases: *Exploration*, *Distraction*, and *Reward*.

- *Exploration Phase*: The agent observes a room exhibiting a randomly assigned RGB color.

- *Distraction Phase*: The environment introduces randomly appearing apples, serving as distractions for the agent.

- *Reward Phase*: The agent is required to select a block that matches the initial room color observed during the Exploration Phase.

In our experimental configuration, the duration of **Phase 1** (*Exploration*) is fixed at 1 step, while **Phase 3** (*Reward*) is fixed at 15 steps. The target colors in the Reward Phase are randomly selected from a predefined set of three colors: blue, red, and green.

Our setup diverges from that of Ni et al. (2024) primarily in the reward structure:

- Collecting apples during the *Distraction Phase* yields no reward.

- A reward of 1 is granted solely upon the successful completion of the goal in the *Reward Phase*, rendering the environment characterized by entirely sparse rewards. And in the *Reward Phase*, we use a `fixed_symbol_colour_map`.

- Additionally, in *VisualMatch*, the duration of the *Exploration Phase* is condensed to 1 step, compared to the 15 steps employed in Ni et al. (2024).

The VisualMatch task is thus intricately designed to test an agent's proficiency in managing long-term dependencies within its decision-making processes. The partially observable nature of the environment, with the agent confined to a 5×5 grid field of view at each step, necessitates strategic decision-making based on incomplete information, effectively simulating numerous real-world scenarios.

### A.2 Benchmark in Non-Memory Domains

**Atari 100k Benchmark** The **Atari 100k** benchmark, introduced by SimPLe (Kaiser et al., 2024), is extensively utilized in research focused on sample-efficient reinforcement learning. This benchmark encompasses 26 diverse Atari games with image-based inputs and discrete action spaces, accommodating up to 18 possible actions per game. The diversity of games ensures a comprehensive and robust evaluation of algorithmic performance across a wide range of environments. In the Atari 100k benchmark, agents are permitted to interact with each game environment for a total of 100,000 steps per game, which corresponds to 400,000 environment frames when considering frame skipping at every 4 frames. This setup emphasizes the importance of sample efficiency, as agents must learn effective policies with a limited number of interactions.

**DeepMind Control Suite** We utilize a collection of 18 continuous control tasks from the DMControl suite, specifically selected from the *proprioceptive inputs* domain. These tasks exhibit significant variability in objectives, observation spaces, and action dimensions, providing a comprehensive testbed for evaluating continuous control algorithms. All tasks are modeled as infinite-horizon continuous control environments; however, for the purpose of evaluation, we impose a fixed episode length of 1,000 steps and eliminate any termination conditions to maintain consistency across tasks.

Consistent with the methodology outlined in Hansen et al. (2023), we apply an action repeat value of 2 across all tasks. This results in an effective episode length of 500 decision steps. The primary performance metric employed is the cumulative episode return, which quantifies the agent's ability to achieve objectives across the varied control tasks. This metric provides a clear and quantifiable measure of an agent's proficiency in navigating and manipulating the diverse environments presented by the DMControl suite.

# B Implementation Details

## B.1 Algorithm Details

Here, we present the complete training pipeline of UniZero in Algorithm 1. The `training_loop` of the UniZero algorithm consists of two primary procedures:

1. `collect_experience`: This procedure gathers experiences (trajectories) $\{o_t, a_t, r_t, d_t\}$ and the improved policy $\pi_t$ derived from Monte Carlo Tree Search (MCTS) into the replay buffer $\mathcal{B}$. The agent interacts with the environment by sampling actions $a_t$ from the MCTS policy $\pi_t$, which is generated by performing `MCTS` in the learned latent space.

2. `update_world_model`: This procedure jointly optimizes the world model and the policy. UniZero updates the decision-oriented world model, policy, and value using samples from $\mathcal{B}$.

`collect_steps` in Algorithm 1 is defined as `num_episodes_each_collect` × `episode_length`. In our experiments, `num_episodes_each_collect` is typically set to 8. The parameter `world_model_iterations` in Algorithm 1 is calculated as `collect_steps` × `replay_ratio` (the ratio between collected environment steps and model training steps) (Schwarzer et al., 2023). In our experiments, `replay_ratio` is usually set to 0.25.

---

**Algorithm 1:** UniZero

---

**Procedure** `training_loop()`:

    **for** *train_iterations* **do**

        Create a Key-Value Cache for the memory: $KV_M = \{\}$

        `collect_experience(`*collect_steps*`)`

        **for** *world_model_iterations* **do**

            `update_world_model()`

        `// Due to the variations in the parameters of the world model.  We need clear the old Key-Value Cache`

        Clear the Key-Value Cache: $KV_M = \{\}$

**Procedure** `collect_experience(`$n$`)`:

    $o_0 \leftarrow$ `env.reset()`

    **for** $t = 0$ **to** $n - 1$ **do**

        $z_t \leftarrow h_\theta(o_t)$

        Sample $a_t \sim \pi_t = \pi(a_t|z_t)$, which is obtained through `MCTS(`$z_t$`, `$KV_M$`, `$\mathcal{W}$`)`

        Add the latest Key-Value Cache $KV(z_t, a_t)$ to $KV_M$

        $o_{t+1}, r_t, d_t \leftarrow$ `env.step(`$a_t$`)`

        **if** $d_t = 1$ **then**

            $o_{t+1} \leftarrow$ `env.reset()`

    $\mathcal{B} \leftarrow \mathcal{B} \cup \{o_t, a_t, r_t, o_{t+1}, \pi_t\}_{t=0}^{n-1}$

**Procedure** `update_world_model()`:

    Sample a mini-batch of sequences $\{(o_t, a_t, r_t, o_{t+1}, \pi_t)_{t=i}^{i+H-1}\} \sim \mathcal{B}$ `// where` $H$ `is the training context length.`

    Compute target TD-value $\hat{v}_t$, and target next latent state $\bar{z}_{t+1}$ according to the target world model $\bar{\mathcal{W}}$

    Optimize the world model and policy jointly according to Equation 3

**Procedure** `MCTS(`$z_t$`, `$KV_M$`, `$\mathcal{W}, sim$`)`:

    `// The following process will repeat` $sim$ `iterations/simulations, where` $i$ `represents the current simulation step.`

    **Require:** $N_i(\hat{z}, a), Q_i(\hat{z}, a), P_i(\hat{z}, a), R_i(\hat{z}, a), Z_i(\hat{z}, a)$

    Initialize root node $\leftarrow z_t$

    **repeat**

        $a^* \leftarrow \text{PUCT}(Q, P, N)$ as in Equation 4

    **until** $N_i(\hat{z}_t^l, a^l) = 0$

    Evaluate the leaf root node $\hat{z}_t^l$ using $\mathcal{W}$: $p_t^l, v_t^l \leftarrow f_\theta(\hat{z}_t^l, KV_M)$, $\hat{z}_t^{l+1}, \hat{r}_t^l \leftarrow g_\theta(\hat{z}_t^l, a^l, KV_M)$ and stored the dynamics and decision quantities into the corresponding tables $R_i(\hat{z}, a), Z_i(\hat{z}, a), P_i(\hat{z}, a)$

    **for** *each $\hat{z}$ along the search path* **do**

        $Q_{i+1}(\hat{z}, a) = \frac{N_i(\hat{z}, a) \cdot Q_i(\hat{z}, a) + \hat{v}(\hat{z})}{N_i(\hat{z}, a) + 1}$

        $N_{i+1}(\hat{z}, a) = N_i(\hat{z}, a) + 1$

    **return** $\pi_t = $ `Normalization(`$\{N_{i+1}(z_t, a_t)|a_t \in \mathcal{A}\}$`)`

---

### B.1.1 MCTS in the Learned Latent Space

As delineated in Algorithm 1, the `MCTS` procedure (Schrittwieser et al., 2019) within the learned latent space comprises 3 phases in each simulation step $i$. The total iterations/simulation steps in a single *search* process is denoted as *sim*:

- **Selection**: Each simulation initiates from the internal root state $z_t$, which is the latent state encoded by the encoder $h_\theta$ given the current observation $o_t$. The simulation proceeds until it reaches a leaf node $\hat{z}_t^l$, where $t$ signifies the search root node is at timestep $t$, and $l$ indicates it's a the leaf node. For each hypothetical timestep $k = 1, ..., l$ of the simulation, actions are chosen based on the Predictor Upper

Confidence Bound applied on Trees (PUCT) (Rosin, 2011) formula:

$$a^{k,*} = \arg\max_a \left[ Q(\hat{z}, a) + P(\hat{z}, a) \frac{\sqrt{\sum_b N(\hat{z}, b)}}{1 + N(\hat{z}, a)} \left( c_1 + \log\left( \frac{\sum_b N(\hat{z}, b) + c_2 + 1}{c_2} \right) \right) \right] \tag{4}$$

where $N$ represents the visit count, $Q$ denotes the estimated average value, and $P$ is the policy's prior probability. The constants $c_1$ and $c_2$ regulate the relative weight of $P$ and $Q$. For the specific values, please refer to Table 8. For $k < l$, the next state and reward are retrieved from the latent state transition and reward table as $\hat{z}^{k+1} = S(\hat{z}^k, a^k)$ and $\hat{r}^k = R(\hat{z}^k, a^k)$.

- **Expansion**: At the final timestep $l$ of the simulation $i$, the predicted reward and latent state are computed by the dynamics network $g_\theta$: $\hat{r}^l, \hat{z}^{l+1} = g_\theta(\hat{z}^l, a^l, KV_M)$, and stored in the corresponding tables, $R(\hat{z}^l, a^l) = r^l$ and $S(\hat{z}^l, a^l) = \hat{z}^{l+1}$. The policy and value are computed by the decision network $f_\theta$: $p^l, v^l = f_\theta(\hat{z}^l, KV_M)$. A new internal node, corresponding to state $z^l$, is added to the search tree. Each edge $(\hat{z}^l, a)$ from the newly expanded node is initialized to $\{N(s^l, a) = 0, Q(s^l, a) = 0, P(s^l, a) = p^l\}$.

- **Backup**: At the end of the simulation, the statistics along the simulation path are updated. The estimated cumulative reward at step $k$ is calculated based on $\bar{v}^l$, i.e., an $(l-k)$-TD bootstrapped value:

$$\hat{v}^k = \sum_{i=0}^{l-1-k} \gamma^i \hat{r}_{k+1+i} + \gamma^{l-k} \bar{v}^l \tag{5}$$

where $\hat{r}$ are predicted rewards obtained from the dynamics network $g_\theta$, and $\bar{v}$ are obtained from the target decision network $\bar{f}_\theta$. Subsequently, $Q$ and $N$ are updated along the search path, following the equations in the `MCTS` procedure described in 1.

Upon completion of the search, the visit counts $N(\hat{z}, a)$ at the root node $z_t$ are normalized to derive the improved policy:

$$\pi_t = \mathcal{I}_\pi(a|z_t) = \frac{N(z_t, a)^{1/T}}{\sum_b N(z_t, b)^{1/T}} \tag{6}$$

where $T$ is the temperature coefficient controlling exploration. Finally, an action is sampled from this distribution for interaction with the environment. UniZero leverages key-value (KV) caching and attention mechanisms to enhance backward memory capabilities and employs MCTS to improve forward planning efficiency. By integrating these two technological directions, UniZero significantly advances more general and efficient planning.

## B.2 Architecture Details

**Encoder.** In the Atari 100k experiment, our observation *encoder* architecture principally follows the framework described in Niu et al. (2024), utilizing the convolutional networks. A notable modification in UniZero is the addition of a *linear* layer at the end, which maps the original three-dimensional features to a one-dimensional latent state of length 768 (denoted as latent state dim, $D$), facilitating input into the transformer backbone network. Additionally, we have incorporated a *SimNorm* operation, similar to the details described in the TD-MPC2 paper (Hansen et al., 2023). Let $V$ (=8 in all our experiments) be the dimensionality of each simplex $\mathbf{g}$, constructed from $L$ (= $D$ / $V$) partitions of $\mathbf{z}$. SimNorm applies the following transformation:

$$\mathbf{z}^{sim\_norm} \doteq [\mathbf{g}_1, \ldots, \mathbf{g}_i, \ldots, \mathbf{g}_L], \quad \mathbf{g}_i = \frac{e^{\mathbf{z}_{i:i+V}/\tau}}{\sum_{j=1}^V e^{\mathbf{z}_{i:i+V}/\tau}}, \tag{7}$$

where $\mathbf{z}^{sim\_norm}$ is the simplicial embedding (Lavoie et al., 2022) of $\mathbf{z}$, $[\cdot]$ denotes concatenation, and $\tau > 0$ is a temperature parameter that modulates the sparsity of the representation. We set $\tau$ to 1. As demonstrated in 4.5, SimNorm is crucial for the training stability of *UniZero*.

**Table 4: Architecture of the encoder** for *VisualMatch*. The size of the submodules is omitted and can be derived from the shape of the tensors. LeakyReLU refers to the leaky rectified linear units used for activation, while Linear represents a fully-connected layer. *SimNorm* (Hansen et al., 2023) operations introduces natural sparsity by constraining the L1 norm of the latent state to a fixed constant, thereby ensuring stable gradient magnitudes. Conv denotes a CNN layer, characterized by kernel $= 3$, stride $= 1$, and padding $= 1$. BN denotes the batch normalization layer.

| Submodule | Output shape |
|---|---|
| Input image ($o_t$) | $3 \times 5 \times 5$ |
| Conv1 + BN1 + LeakyReLU | $16 \times 5 \times 5$ |
| Conv2 + BN2 + LeakyReLU | $32 \times 5 \times 5$ |
| Conv3 + BN3 + LeakyReLU | $64 \times 5 \times 5$ |
| AdaptiveAvgPool2d | $64 \times 1 \times 1$ |
| Linear | 64 |
| *SimNorm* | 64 |

For the encoder used in the Long-Dependency Benchmark, we employed a similar conv. network architecture, with a latent state of length 64. Specifics can be found in the related table (see Table 4).

**Dynamics Head and Decision Head.** Both the dynamics head and the decision head utilize two-layer linear networks with GELU (Hendrycks & Gimpel, 2016) activation functions. Specifically, the final layer's output dimension for predicting value and reward corresponds to the support size (refer to B.4) (Schrittwieser et al., 2019; Bellemare et al., 2017). For predicting policy, the output dimension matches the action space size. For predicting the next latent state, the output dimension aligns with the latent state dimension, followed by an additional *SimNorm* normalization operation. In the context of *Atari* games, this dimension is set to 768, whereas for *VisualMatch*, it is configured to 64.

**Transformer Backbone.** Our transformer backbone is based on the nanoGPT project, as detailed in Table 7. For each timestep input, UniZero processes **two** primary modalities. The first modality involves latent states derived from *observations*, normalized in the final layer using *SimNorm*, as discussed above. The second modality pertains to *actions*, which are converted into embeddings of equivalent dimensionality to the latent states via a learnable *nn.Embedding* layer. For continuous action spaces, these can alternatively be embedded using a learnable linear layer. Notably, rewards are *not* incorporated as inputs in our current framework. This choice is based on the rationale that rewards are determined by observations and actions, and thus do not add additional insight into the decision-making process. Furthermore, our approach does not employ a return-conditioned policy (Chen et al., 2021; Lee et al., 2022), leaving the potential exploration of reward conditions to future work. Each timestep's observed results and corresponding action embeddings are added with a *learnable positional encoding*, implemented through *nn.Embedding*, as shown in Table 5. While advanced encoding methods like rotary positional encoding (Su et al., 2023) and innovate architectures of transformer (Dao et al., 2022) exist, their exploration is reserved for future studies. Detailed hyper-parameters can be found in Appendix B.4.

**Table 5: Positional encoding** module. $w_{1:H}$ is a learnable parameter matrix with shape $H \times D$, and $H$ refers to the sequence length and $D$ refers to the latent state dimension, 768 for the *Atari*, 64 for the *VisualMatch*.

| Submodule | Output shape |
|---|---|
| Input $((z_{1:H}, a_{1:H}))$ | $2H \times D$ |
| Add $((z_{1:H}, a_{1:H}) + w_{1:H})$ | |

**UniZero (RNN).** This variant employs a training setup akin to *UniZero* but utilizes a GRU (Chung et al., 2014) as the backbone network. During training, all observations are utilized. During inference, the hidden state of the GRU is reset every $H_{\text{infer}}$ steps. The recursively predicted hidden state $h_t$ and observation embedding $z_t$ serve as the root node of the MCTS. The recursively predicted hidden state $h_t$ and predicted

**Table 6:** Details of **Transformer block**. $MHSA$ refers to multi-head self-attention and $FFN$ refers to feed-forward networks. Dropout mechanism can prevent over-fitting.

| Submodule | Module alias | Output shape |
|---|---|---|
| Input features (label as $x_1$) | - | $2H \times D$ |
| Multi-head self attention + Dropout$(p)$
Linear1 + Dropout$(p)$
Residual (add $x_1$)
LN1 (label as $x_2$) | MHSA | $2H \times D$ |
| Linear2 + GELU
Linear3 + Dropout$(p)$
Residual (add $x_2$)
LN2 | FFN | $2H \times D$
$2H \times D$
$2H \times D$
$H \times D$ |

**Table 7: Transformer-based latent world model** $(p_{1:H}, v_{1:H}, \hat{z}_{1:H}, \hat{r}_{1:H}, h^z_{1:H}, h^{z;a}_{1:H}) = f_\theta(z_{1:H}, a_{1:H})$. The hidden states $(h^z_{1:H}, h^{z;a}_{1:H})$ in the final layer of the transformer are referred to as the *implicit latent history.* Positional encoding and Transformer block are explained in Table 5 and 6.

| Submodule | Output shape |
|---|---|
| Input $((z_{1:H}, a_{1:H}))$
Positional encoding
Transformer blocks $\times N$
(implicit) Latent history $((h^z_{1:H}, h^{z;a}_{1:H}))$
Decision head $(p_{1:H}, v_{1:H})$
Dynamic head $(\hat{z}_{1:H}, \hat{r}_{1:H})$ | $2 * H \times D$ |

latent state $\hat{z}_t$ serve as the internal nodes. At the root node, due to the complexity of GRU training, the recurrent hidden state $h_t$ may not fully capture the historical information. At the internal nodes, the issue is exacerbated by the accumulation of errors, leading to inaccurate predictions and consequently limiting performance. For an illustration of the training process, please refer to Figure 9.

## B.3   Extension of UniZero to Continuous Action Spaces

To adapt *UniZero* for environments with continuous action spaces (Hubert et al., 2021b), we introduce several key modifications to both the network architecture and the MCTS procedure. These adaptations are crucial for accurate modeling and effective decision-making when actions are not confined to a discrete set but can take any value within a continuous range.

### B.3.1   Policy Network Modification

The policy network's decision head is redesigned to accommodate continuous actions by outputting parameters suitable for continuous distributions. Instead of producing logits for a finite set of discrete actions, the network now generates the mean ($\mu$) and standard deviation ($\sigma$) parameters of a Gaussian distribution for each action dimension. Specifically, for each dimension $i$ of the action space $\mathcal{A}$, the network predicts $\mu_{\theta,i}(s)$ and $\sigma_{\theta,i}(s)$, enabling the representation of a continuous action space. Formally, the policy is defined as:

$$\pi_\theta(a|s) = \mathcal{N}(\mu_\theta(s), \sigma_\theta^2(s)), \tag{8}$$

where $\mu_\theta(s) \in \mathbb{R}^{|\mathcal{A}|}$ and $\sigma_\theta(s) \in \mathbb{R}^{|\mathcal{A}|}$ are the mean and standard deviation vectors parameterized by $\theta$, and $|\mathcal{A}|$ denotes the dimensionality of the action space.

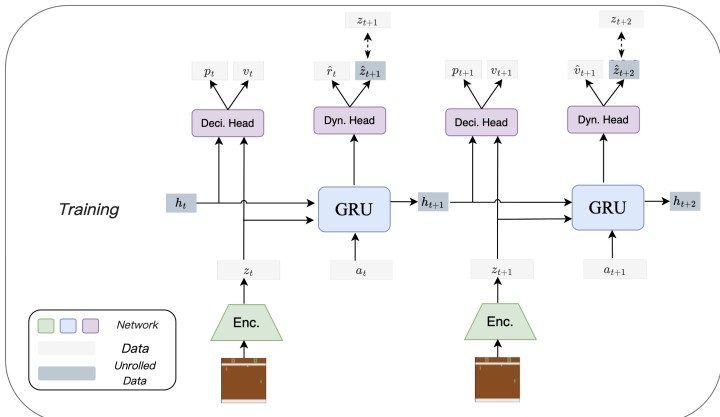

**Figure 9:** Training pipeline of *UniZero (RNN)*. During training, all observations are utilized. The recursively predicted hidden state $h_t$ and observation embedding $z_t$ serve as the root node. The recursively predicted hidden state $h_t$ and predict latent state $\hat{z}_t$ serve as the internal nodes of MCTS. During inference, the GRU hidden state is reset every $H_{\text{infer}}$ steps. However, potential inaccuracies may arise from the recursively predicted hidden state $h_t$ due to the complexity of GRU training

.

### B.3.2 MCTS Node Expansion Adaptation

In continuous action spaces, enumerating all possible actions is computationally infeasible. To address this challenge, we modify the node expansion strategy within MCTS to sample a finite set of actions from a proposal distribution derived from the policy network. Specifically, actions are sampled from the Gaussian proposal distribution $\beta(a|s)$ defined as:

$$\beta(a|s) = \mathcal{N}(\mu_\theta(s), \sigma_\theta^2(s)). \tag{9}$$

During each node expansion, a finite number $K \ll |\mathcal{A}|$ of actions are sampled from $\beta(a|s)$, with $K = 20$ in our continuous action experiments. Each sampled action $a_i$ is associated with its corresponding probability under the proposal distribution, $\beta(a_i|s) = \pi_\theta(a_i|s)$.

### B.3.3 PUCT Formula Adaptation

To maintain a balanced exploration-exploitation trade-off in continuous action spaces, the Predictor + UCT (PUCT) formula is adjusted accordingly. In this adaptation, the prior policy distribution is transformed from the original prior policy $P(\hat{z}, a)$ in Equation 4 to a uniform policy $u(\hat{z}, a)$. This modification leverages the prior implicitly without introducing explicit bias, ensuring that exploration is not disproportionately influenced by the policy.

### B.3.4 Policy Distillation from MCTS Visit Counts

After the MCTS procedure, the visit counts $N(s, a)$ at the root node are normalized to derive an improved policy estimate $\hat{\pi}_\beta(a|s)$:

$$\hat{\pi}_\beta(a|s) = \frac{N(s, a)}{\sum_b N(s, b)}. \tag{10}$$

To integrate this improved policy into the policy network, we employ a projection operator $P$. Inspired by *MuZero*, this projection minimizes the Kullback-Leibler (KL) divergence between the improved policy and the network's policy output:

$$\mathcal{L}_{\text{KL}} = \text{KL}\left(\hat{\pi}_\beta(\cdot|s) \,\|\, \pi_\theta(\cdot|s)\right). \tag{11}$$

### B.3.5 Policy Loss Calculation for Continuous Actions

The policy loss for continuous actions is computed by minimizing the KL divergence between the improved policy derived from MCTS and the policy network's output distribution. The loss function is defined as:

$$\mathcal{L}_{\text{policy}} = -\sum_i \hat{\pi}_\beta(a_i|s) \cdot \log \pi_\theta(a_i|s), \tag{12}$$

where $a_i$ are the sampled actions from the improved policy. This formulation ensures numerical stability by operating directly with probabilities rather than log probabilities.

### B.3.6 Summary of Modifications

Extending **UniZero** to accommodate continuous action spaces involves the following key modifications:

- **Policy Network Redesign**: The policy network is modified to output the mean and standard deviation parameters of Gaussian distributions for each action dimension, enabling the representation of continuous actions.

- **MCTS Node Expansion Adjustment**: The node expansion strategy within MCTS is adapted to sample a finite set of actions from the Gaussian proposal distribution, avoiding the infeasibility of enumerating all possible continuous actions.

- **PUCT Formula Adaptation**: The PUCT formula is revised to appropriately balance exploration and exploitation in the context of continuous actions, without introducing bias towards the prior policy.

- **Policy Distillation via KL Divergence Minimization**: The visit counts obtained from MCTS are distilled into the policy network by minimizing the KL divergence between the improved policy and the network's policy distribution.

These enhancements enable *UniZero* to effectively manage continuous control tasks, ensuring robust performance across diverse and complex action spaces.

### B.4 Baselines and Hyperparameters

**Baselines.** Our MuZero implementation is based on the LightZero (Niu et al., 2024) framework. Unless otherwise stated, all references to MuZero in this work denote its variant augmented with self-supervised learning regularization (MuZero w/ SSL), as discussed in Section 6. (1) **VisualMatch Baselines:** We compare against MuZero and the SAC-Discrete variant combined with the GPT backbone, as proposed in Ni et al. (2024), referred to as *SAC-GPT*. (2) **Atari 100k Baselines:** The baseline used is MuZero. (3) **DMControl Baselines:** DreamerV3 (Hafner et al., 2023) is used as the baseline, a model-based approach that optimizes a model-free policy using rollouts generated from a learned environment model. For a comprehensive comparison with prior model-based RL algorithms such as TWM (Robine et al., 2023), IRIS (Micheli et al., 2022), DreamerV3 (Hafner et al., 2023), STORM (Zhang et al., 2023), TDMPC2 (Hansen et al., 2023) and MuZero (Schrittwieser et al., 2019), please refer to Table 12.

**Hyperparameters.** We maintain a consistent set of hyperparameters across all tasks unless explicitly stated otherwise. Table 8 outlines the key hyperparameters for *UniZero*, which are closely aligned with those reported in Niu et al. (2024). Furthermore, Table 9 provides the critical hyperparameters for *MuZero w/ SSL*, *MuZero w/ Context*, and *UniZero (RNN)*.

### B.5 Computational Cost

Compared to MuZero, although UniZero may have a slightly higher computational cost per step, the difference in actual training time is not significant due to their different training paradigms—UniZero employs parallel training, whereas MuZero requires executing a recursive for-loop during training.

**Table 8: UniZero Key Hyperparameters**. Most hyperparameters are aligned with those in Niu et al. (2024) to enable fair comparisons. For brevity, *long-term* denotes the long-term dependency benchmark, *DMC* refers to the DeepMind Control Suite, and *Atari* refers to the Atari 100k benchmark.

| Hyperparameter | Value |
|---|---|
| **Planning** | |
| Number of MCTS Simulations ($sim$) | 50 |
| Number of Sampled Actions ($K$) | 20 (DMC only) |
| Inference Context Length ($H_{\text{infer}}$) | 4 (Atari); 2 (DMC); `memory_length + 16` (long-term) |
| Temperature | 0.25 |
| Dirichlet Noise ($\alpha$) | 0.3 |
| Dirichlet Noise Weight | 0.25 |
| Coefficient $c_1$ | 1.25 |
| Coefficient $c_2$ | 19652 |
| **Environment and Replay Buffer** | |
| Replay Buffer Capacity | $1,000,000$ |
| Sampling Strategy | Uniform |
| Observation Shape (Atari) | `(3, 64, 64)` (stack1); `(4, 64, 64)` (stack4) |
| Observation Shape (Long-term) | `(3, 5, 5)` |
| Observation Shape (DMC) | Varied across tasks |
| Reward Clipping | True (Atari only) |
| Number of Frames Stacked | 1 (stack1); 4 (stack4; Atari only) |
| Frame Skip | 4 (Atari); 2 (DMC) |
| Game Segment Length | 400 (Atari); 100 (DMC); `memory_length + 16` (long-term) |
| Data Augmentation | False |
| **Architecture** | |
| Latent State Dimension ($D$) | 768 (Atari, DMC); 64 (long-term) |
| Number of Transformer Heads | 8 (Atari, DMC); 4 (long-term) |
| Number of Transformer Layers ($N$) | 2 |
| Dropout Rate ($p$) | 0.1 |
| Activation Function | LeakyReLU (encoder); GELU (others) |
| Reward/Value Bins | 101 |
| SimNorm Dimension ($V$) | 8 |
| SimNorm Temperature ($\tau$) | 1 |
| **Optimization** | |
| Training Context Length ($H$) | 10 |
| Replay Ratio | 0.25 |
| Buffer Reanalyze Frequency | 0 (DMC, long-term); 1/50 (Atari); 0 in Figure 2 |
| Batch Size | 64 |
| Optimizer | AdamW |
| Learning Rate | $1 \times 10^{-4}$ |
| Next Latent State Loss Coefficient | 10 |
| Reward Loss Coefficient | 0.1 (DMC); 1 (others) |
| Policy Loss Coefficient | 0.1 (DMC); 1 (others) |
| Value Loss Coefficient | 0.1 (DMC); 0.5 (others) |
| Policy Entropy Coefficient | $1 \times 10^{-4}$ |
| Weight Decay | $10^{-4}$ |
| Max Gradient Norm | 5 |
| Discount Factor | 0.997 |
| Soft Target Update Momentum | 0.05 |
| Hard Target Network Update Frequency | 100 |
| Temporal Difference (TD) Steps | 5 |

**Table 9: Key Hyperparameters** for *MuZero w/ SSL*, *MuZero w/ Context*, and *UniZero (RNN)* on Atari.

| Hyperparameter | Value |
|---|---|
| **Planning** | |
| Number of MCTS Simulations ($sim$) | 50 |
| Inference Context Length ($H_{\text{infer}}$) | 0 (*MuZero w/ SSL*); 4 (for other two algo.) |
| Temperature | 0.25 |
| Dirichlet Noise ($\alpha$) | 0.3 |
| Dirichlet Noise Weight | 0.25 |
| Exploration Coefficient ($c_1$) | 1.25 |
| Visit Count Coefficient ($c_2$) | 19652 |
| **Environment and Replay Buffer** | |
| Replay Buffer Capacity | $1,000,000$ |
| Sampling Strategy | Uniform |
| Observation Shape (Atari) | `(3, 64, 64)` (stack1); `(4, 64, 64)` (stack4) |
| Observation Shape (Long-term) | `(3, 5, 5)` |
| Reward Clipping | True (Atari only) |
| Number of Frames Stacked | 1 (stack1); 4 (stack4; Atari only) |
| Frame Skip | 4 (Atari only) |
| Game Segment Length | 400 (Atari); `memory_length + 16` (long-term) |
| Data Augmentation | True |
| **Optimization** | |
| Training Context Length ($H$) | 10 |
| Replay Ratio | 0.25 |
| Buffer Reanalyze Frequency | 0 |
| Batch Size | 256 |
| Optimizer | SGD |
| Learning Rate Schedule | $0.2 \rightarrow 0.02 \rightarrow 0.002$ (Ye et al., 2021) |
| SSL Loss Coefficient | 2 |
| Reward Loss Coefficient | 1 |
| Policy Loss Coefficient | 1 |
| Value Loss Coefficient | 0.25 |
| Policy Entropy Loss Coefficient | 0 |
| Number of Reward/Value Bins | 101 |
| Discount Factor ($\gamma$) | 0.997 |
| Target Network Update Frequency | 100 |
| Weight Decay | $10^{-4}$ |
| Maximum Gradient Norm | 5 |
| Temporal Difference (TD) Steps | 5 |

The following computational overhead experiments were conducted on a Kubernetes cluster with the following specifications: a single NVIDIA A100 80GB GPU, a 24-core CPU, and 100GB of memory. Under these computational resources and the hyperparameter settings listed in Table 8, the training time required for UniZero is as follows:

- Training an Atari agent for 100k steps takes approximately 4 hours (see Figure 14), which is comparable to the previously reported training time for MuZero.

- On the *VisualMatch* task (with a memory length of 500), completing 1M training steps takes approximately 30 hours (see Figure 4).

It is important to note that the above results were obtained using a single GPU without special computational optimizations. Currently, we have implemented a multi-GPU version, and experiments indicate that its speedup ratio is close to linear. In the future, we plan to further explore existing Transformer optimization techniques to enhance computational efficiency.

## C  Multi-task Learning Details

In this section, we evaluate UniZero's capability to seamlessly extend to a multi-task learning setting. While UniZero demonstrates exceptional performance on single-task problems with varying levels of dependency (Section 4.1), its decoupled-yet-unified design enables it to scale effectively to multi-task environments. By leveraging a shared transformer backbone, UniZero adaptively captures diverse dependencies across tasks within a unified architecture and training paradigm. To validate its multi-task learning potential, we present results on eight Atari games: *Alien*, *Boxing*, *ChopperCommand*, *Hero*, *MsPacman*, *Pong*, *RoadRunner*, and *Seaquest*. Unless explicitly stated, the multi-task hyperparameters remain consistent with those outlined in Table 8.

### C.1  Architecture

The observation space for all tasks is standardized and consists of $(3, 64, 64)$ RGB images. We configure `full_action_space=True`, following Bellemare et al. (2013), which yields a unified action space with 18 discrete actions across all tasks. The primary architectural difference from the single-task setup is the introduction of *independent decision heads and dynamics heads* for each task (as described in Kumar et al. (2022)), which requires only a minimal increase in parameters. The core transformer backbone and encoder, however, are *shared* across all tasks, enhancing parameter efficiency and enabling shared representation learning. In our initial experiments, we also tested the shared head setting. The results indicate that compared to an independent head for each game, a shared head leads to an overall performance drop of around 50%. Although a single, powerful head should theoretically handle these tasks, achieving this practically requires a more complex design to address gradient conflict issues. Therefore, this paper mainly explores the independent head setting.

### C.2  Training

During training, each task is assigned its own data collector, responsible for sequentially gathering trajectories and storing them in separate replay buffers. For gradient updates, we sample a batch of size `task_batch_size=32` from each task, aggregate the samples into a larger minibatch, and compute the loss for each task using the objective function defined in Equation 3. The task-specific losses are averaged to obtain the total loss, which is then backpropagated to update the shared network parameters.

### C.3  Results

**Performance Comparison.**   Table 2 and Figure 10 demonstrate that UniZero (multi-task) significantly outperforms both UniZero (single-task) and MuZero (multi-task) in terms of normalized mean and median scores across the evaluated environments. This improvement underscores UniZero's scalability and effectiveness

as a latent world model for generalized agent training. Notably, UniZero achieves comparable sample efficiency across all tasks relative to single-task learning, highlighting its robust multi-task learning capabilities.

**Latent State Analysis.** To further investigate the success of multi-task learning, we analyze the latent states learned by UniZero using T-SNE visualizations (Maaten & Hinton, 2008) (Figure 12). Specifically, we sample 40 transitions from each game using the final model trained in the multi-task setup. The observation samples are encoded into 768-dimensional latent states using UniZero's representation network, which are then reduced to two dimensions via T-SNE. The visualization reveals well-defined clustering tendencies in the latent state spaces for each game, reflecting UniZero's ability to effectively capture task-specific dynamics. However, certain games, such as *Alien*, exhibit more dispersed latent representations. This dispersion may arise from *Alien*'s similarity to other environments, such as *MsPacman* (both belonging to the Maze class), allowing greater information sharing across tasks. This shared information likely contributes to *Alien*'s significant performance improvement under the multi-task learning setup compared to the single-task scenario.

**Effect of Model Size.** To investigate the relationship between model size and multi-task learning performance, and to explore whether a scaling law exists, we evaluate the impact of varying the transformer backbone size (nlayer={4, 8, 12}) on eight Atari games in the multi-task setting. The encoder and head architectures are kept constant across all configurations, while other hyperparameters remain consistent with those outlined in Table 8. As illustrated in Figure 11, increasing the model size consistently *improves* sample efficiency across all tasks, demonstrating the scalability of the model as its size grows. These results highlight the potential of larger models to serve as robust latent world models for training generalized agents.

**Extended Evaluation.** To assess the scalability of UniZero, we expand our test scope to cover the full evaluation set of 26 Atari games. As shown in Figure 13 and Table 10, the multi-task model of UniZero achieves a normalized human score comparable to that of UniZero trained on single tasks. In contrast, MuZero *fails* to scale effectively to a large number of tasks. Notably, MuZero performs poorly in multi-task training across the 26 games, making little to no significant progress in most tasks. This limitation likely stems from the substantial differences in dynamical properties across tasks, which make it difficult for its recurrent training paradigm to adapt seamlessly to such variations. As a result, severe conflicts arise during the learning process, ultimately hindering the model's improvement. In contrast, UniZero effectively leverages shared representations across tasks and models task dependencies through its Transformer backbone, thereby maintaining performance levels comparable to dedicated single-task models and demonstrating strong potential in multi-task learning.

**Additional Insights.** In our preliminary experiments, we explored several multi-task gradient correction techniques, including PCGrad (Yu et al., 2020) and CAGrad (Liu et al., 2021). However, these methods yielded minimal performance improvements in our experimental setup and were therefore excluded from the reported results. Additionally, we tested multi-task minibatch sampling strategies, where the sampling ratio for each task was inversely proportional to the average episode length of the respective task. We also experimented with augmenting the latent state space by introducing task-specific, learnable embeddings. Unfortunately, neither approach demonstrated significant benefits, and as such, they were omitted from the final analysis.

**Future Directions.** To address the limitations identified, future work will focus on investigating advanced learning dynamics and task-balancing techniques. Specifically, we plan to explore: (1) improved strategies for balancing tasks during training, (2) optimizing information reuse in MCTS, and (3) integrating multiple modalities and tasks within a unified framework. Furthermore, we aim to study advanced pretraining and fine-tuning methodologies to enhance UniZero's multi-task performance. These directions are expected to provide deeper insights into the challenges of multi-task reinforcement learning and contribute to more robust and scalable solutions.

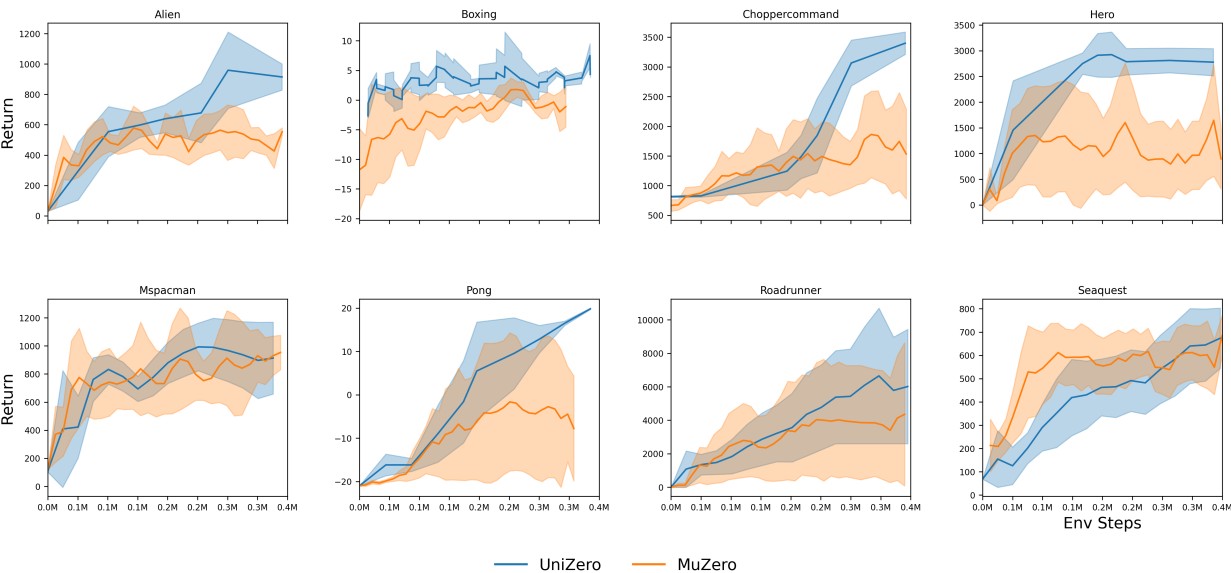

**Figure 10: Performance comparison between UniZero and MuZero on eight Atari games in multi-task settings.** UniZero demonstrates comparable sample efficiency across all tasks relative to MuZero in multi-task learning, underscoring its scalability as a latent world model for training generalized agents. The solid line represents the mean of three runs, and the shaded areas indicate the 95% confidence intervals.

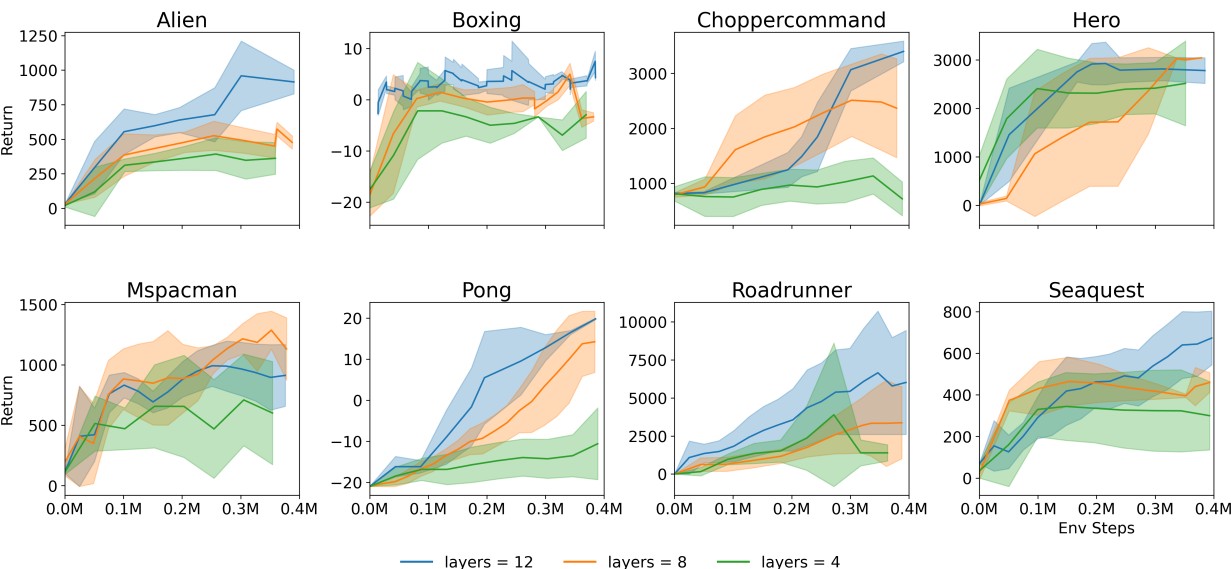

**Figure 11: Effect of model size of UniZero on eight Atari games in multi-task settings.** Increasing the model size consistently improves sample efficiency across all tasks (nlayer={4, 8, 12}), demonstrating the potential of UniZero as a scalable generalized agents.

# D   Additional Single Task Results

## D.1   Experimental Setup

To evaluate the effectiveness and scalability of the proposed **UniZero** algorithm, we conducted experiments on 26 games from the image-based **Atari 100K** benchmark. Detailed configuration settings for the Atari environment are provided in Section B.4. Observations are represented as `(3, 64, 64)` for single-frame

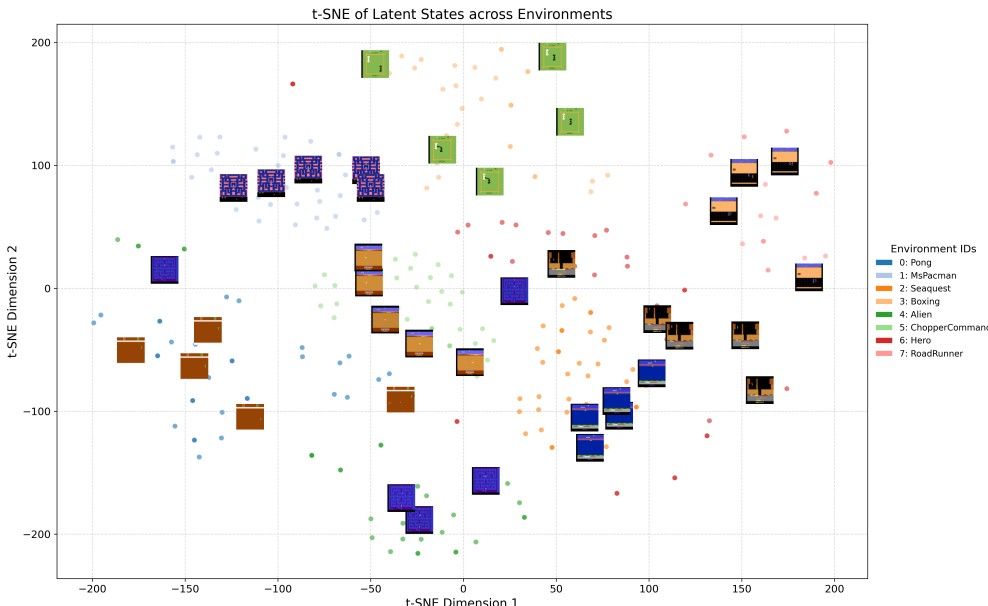

**Figure 12: T-SNE visualization of latent states learned by UniZero on eight Atari games.** The latent state spaces for the same game exhibit pronounced clustering tendencies, reflecting UniZero's ability to learn task-specific representations effectively. A representative subset of states is shown for clarity.

**Table 10:** Performance comparison between UniZero and MuZero in both multi-task and single-task settings at 200k environment steps across 26 Atari games. The results indicate that UniZero exhibits higher scalability for heterogeneous multi-task learning, achieving comparable performance to single-task learning, whereas MuZero *fails* to extend effectively to a large number of tasks. MT denotes multi-task learning and ST denotes single-task learning. Bold numbers indicate the best performance for each metric under the corresponding settings.

| Algorithm | Normalized Mean | Normalized Median |
|---|---|---|
| UniZero (MT) | **0.3133** | **0.1666** |
| MuZero (MT) | 0.0424 | 0.0004 |
| UniZero (ST) | 0.3865 | **0.2154** |
| MuZero (ST) | **0.5696** | 0.2009 |

RGB images (stack size = 1) or as `(4, 64, 64)` for grayscale images with four stacked frames (stack size = 4). This configuration differs from the `(4, 96, 96)` observation format commonly adopted in previous studies such as Ye et al. (2021); Niu et al. (2024). All implementations are based on the latest release of the open-source `LightZero` framework (Niu et al., 2024).

**Baselines.** To benchmark the performance of **UniZero** on the Atari 100K benchmark, we compare it against the following baselines:

- *MuZero* (Schrittwieser et al., 2019): The original MuZero algorithm utilizing a stack size of 4.

- *MuZero (Reproduced)*: Our reimplementation of MuZero, enhanced with self-supervised learning (SSL) and employing a stack size of 4. This variant is referred to as *MuZero w/ SSL* or *MuZero (Reproduced)* (Niu et al., 2024).

To evaluate the capacity of different algorithms for modeling short-term dependencies, both *MuZero* and *MuZero (Reproduced)* are configured with a stack size of 4, while the proposed *UniZero* operates without stacked frames (stack size = 1). All implementations are trained using uniform hyperparameters across all games, with no further game-specific tuning.

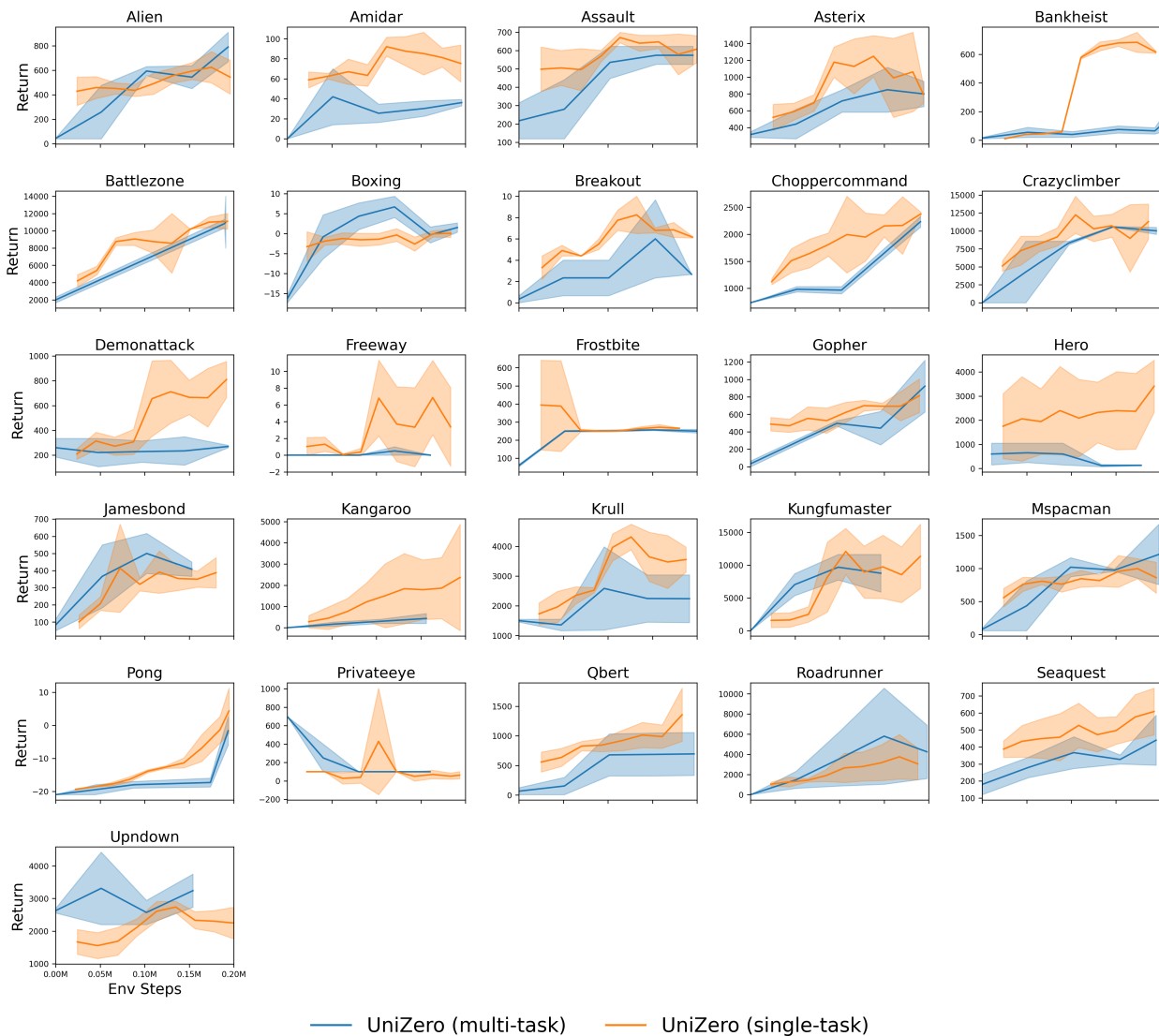

**Figure 13: Performance comparison between multi-task and single-task settings on 26 Atari games.** Multi-task training yields comparable sample efficiency across all tasks relative to single-task learning, further validating UniZero's efficacy as a scalable generalized agent.

For tasks that demand long-term dependency modeling, we extend the evaluation by comparing *UniZero* against the original *MuZero* and the transformer-based *SAC-GPT* algorithm (Ni et al., 2024). The *SAC-GPT* framework integrates transformer architectures with actor-critic methods, enabling the modeling of memory and credit assignment in reinforcement learning, thereby highlighting its potential for addressing long-term dependencies in complex RL environments.

## D.2 Atari 100K Results

Table 11 provides a comprehensive comparison of three methods: *UniZero* (stack size = 1), *MuZero (Reproduced)* (stack size = 4), and the original *MuZero* as reported in Schrittwieser et al. (2019). The results demonstrate that *UniZero* achieves a higher human-normalized median score than *MuZero (Reproduced)*, outperforming the latter in **15** out of 26 Atari games, while maintaining comparable or slightly lower performance in the remaining environments. Notably, both *UniZero* and *MuZero (Reproduced)* are implemented within the

same `LightZero` framework, ensuring a fair and controlled comparison by using identical hyperparameters across all games.

Figure 14 resents the full performance curves, further corroborating that *UniZero* consistently surpasses *MuZero (Reproduced)* in terms of human-normalized median scores. These results highlight the ability of *UniZero* to effectively model both short- and long-term dependencies, which is a critical factor in achieving robust performance on the Atari 100K benchmark.

**Table 11:** Performance comparison of *UniZero*, *MuZero (Reproduced)*, and the original *MuZero* on the Atari 100K benchmark. *UniZero* achieves a higher human-normalized median score than *MuZero (Reproduced)*, outperforming the latter in **15** out of 26 Atari games. The results for the original *MuZero* are directly taken from Schrittwieser et al. (2019) and are provided for reference. Both *UniZero* and *MuZero (Reproduced)* are reimplemented using the `LightZero` framework under identical hyperparameter settings, ensuring fairness in comparison. **Bold** entries denote the superior method between *UniZero* and *MuZero (Reproduced)*, while underlined values indicate the overall best-performing approach across all methods.

| Game | Random | Human | MuZero | MuZero (Reproduced) | UniZero (Ours) |
|---|---|---|---|---|---|
| Alien | 227.8 | 7127.7 | 530.0 | 300 | **600** |
| Amidar | 5.8 | 1719.5 | 39 | 90 | **96** |
| Assault | 222.4 | 742.0 | 500 | **609** | 608 |
| Asterix | 210.0 | 8503.3 | **1734** | 1400 | 1216 |
| BankHeist | 14.2 | 753.1 | 193 | 223 | **400** |
| BattleZone | 2360.0 | 37187.5 | 2688 | 7587 | **11410** |
| Boxing | 0.1 | 12.1 | 15 | **20** | 7 |
| Breakout | 1.7 | 30.5 | 48 | 3 | **8** |
| ChopperCommand | 811.0 | 7387.8 | 1350 | 1050 | **2205** |
| CrazyClimber | 10780.5 | 35829.4 | 56937 | **22060** | 13666 |
| DemonAttack | 152.1 | 1971.0 | 3527 | **4601** | 991 |
| Freeway | 0.0 | 29.6 | 22 | **12** | 10 |
| Frostbite | 65.2 | 4334.7 | 255 | 260 | **310** |
| Gopher | 257.6 | 2412.5 | 1256 | 346 | **853** |
| Hero | 1027.0 | 30826.4 | 3095 | **3315** | 2005 |
| Jamesbond | 29.0 | 302.8 | 88 | 90 | **405** |
| Kangaroo | 52.0 | 3035.0 | 63 | 200 | **1885** |
| Krull | 1598.0 | 2665.5 | 4891 | **5191** | 4484 |
| KungFuMaster | 258.5 | 22736.3 | 18813 | 6100 | **11400** |
| MsPacman | 307.3 | 6951.6 | 1266 | **1010** | 900 |
| Pong | -20.7 | 14.6 | -7 | -15 | **-10** |
| PrivateEye | 24.9 | 69571.3 | 56 | 100 | **500** |
| Qbert | 163.9 | 13455.0 | 3952 | **1700** | 1056 |
| RoadRunner | 11.5 | 7845.0 | 2500 | **4400** | 1100 |
| Seaquest | 68.4 | 42054.7 | 208 | 466 | **620** |
| UpNDown | 533.4 | 11693.2 | 2897 | 1213 | **2823** |
| Normalized Mean (↑) | 0.000 | 1.000 | 0.56 | **0.44** | 0.39 |
| Normalized Median (↑) | 0.000 | 1.000 | 0.23 | 0.13 | **0.22** |

## D.3 DMControl Results

Figure 15 shows the learning curves for all 18 tasks in the **Proprio Control Suite** of **DMControl**. Each solid line represents the mean performance across three seed runs, and shaded regions indicate the 95% confidence intervals. *UniZero*, leveraging *sampled policy iteration* (Hubert et al., 2021b), achieves a higher human-normalized score compared to the state-of-the-art *DreamerV3* (Hafner et al., 2023), highlighting its ability to handle continuous action spaces and diverse control tasks.

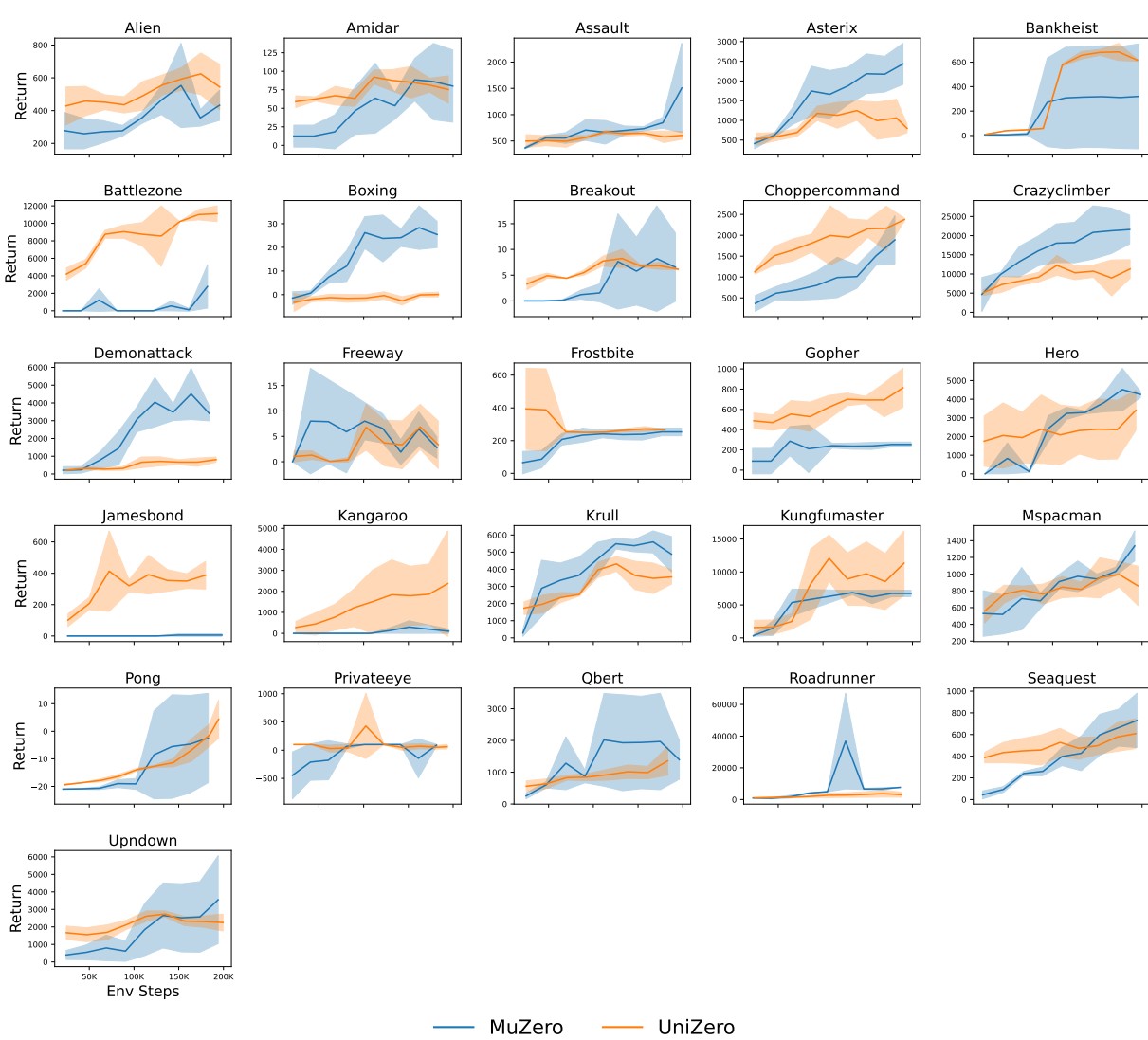

**Figure 14: Performance Comparison on 26 Atari Games between *UniZero* and *MuZero (Reproduced)* in single-task setting.** *UniZero* achieves a higher human-normalized median score. Solid lines represent the mean of three different seed runs, while shaded areas denote the 95% confidence intervals.

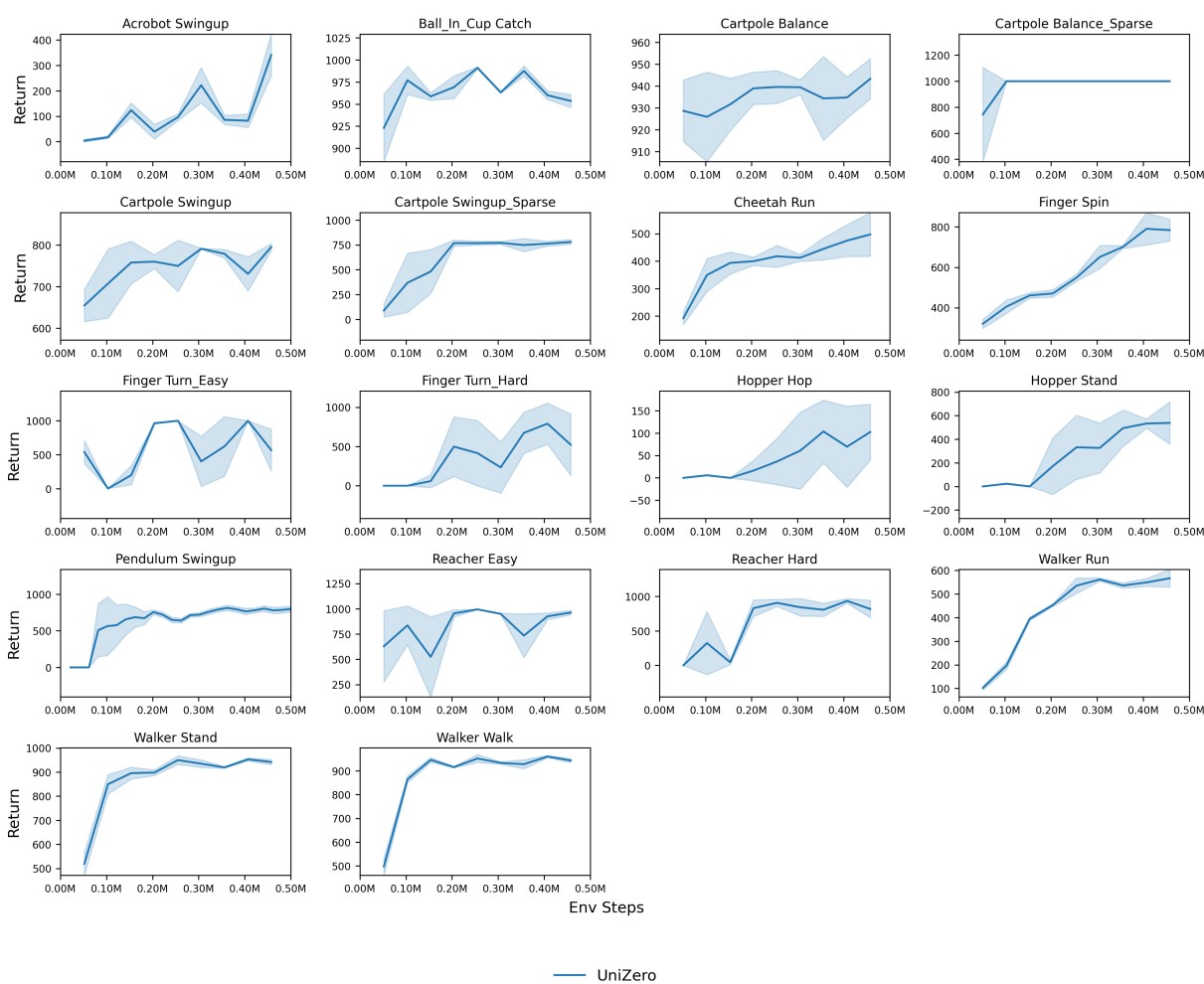

**Figure 15: Detailed Training Curves of *UniZero* on 18 DMControl Tasks.** *UniZero* demonstrates superior performance against *DreamerV3*, underscoring its effectiveness in continuous control settings. Solid lines denote the mean performance over three different seed runs, while shaded areas represent the 95% confidence intervals.

# E    Additional Ablation Study and Analysis

## E.1    Ablation Study Details

In Section 4.5, we evaluate the effectiveness and scalability of UniZero's core designs through a series of ablation experiments. These experiments include investigations into training context lengths, normalization methods, and decode regularization in the *Pong* environment and the *VisualMatch* task.

Additionally, we provide ablation studies on the target world models for both tasks and explore the impact of Transformer depth in *VisualMatch* (memlen=500).

Below, we present comprehensive experimental details and key observations.

- **Model Size Across Different Training Context Lengths ($H = 5, 10, 20, 40$)**: The number of layers in the Transformer backbone is varied, while the number of attention heads is fixed at 8. We examine how context length affects performance in both *Pong* and *VisualMatch*.

- **Latent Normalization**: We compare three normalization methods: the default *SimNorm*, *Softmax*, and *Sigmoid*. These methods are applied to both the encoded latent state and the output of the dynamics network (i.e., the predicted next latent state). Our initial experiments indicate that *LayerNorm* performs comparably to *SimNorm*. However, SimNorm's intra-group Softmax mechanism enables more attribute-specific and generalizable representations (Lavoie et al., 2022), which benefits attribute interpretation and multi-task fine-tuning. Therefore, we default to SimNorm, while retaining LayerNorm as a viable alternative.

- **Decode Regularization**: We introduce a decoder function to map latent states back into the observation space:

$$\text{Decoder: } \hat{o}_t = d_\theta(\hat{z}_t) \quad \triangleright \text{ Maps latent states to observations for regularization.}$$

  Training includes an auxiliary objective:

$$\mathcal{L}_{\text{decode\_reg}} = \|o_t - d_\theta(z_t)\|_1 + \mathcal{L}_{\text{perceptual}}(o_t, d_\theta(z_t)), \quad z_t = h_\theta(o_t),$$

  where the first term represents an $L_1$ reconstruction loss, and the second term is a perceptual loss, as defined in Ni et al. (2024). For these experiments, the decode regularization loss coefficient is set to 0.05. Notably, in *VisualMatch*, only the $L_1$ reconstruction loss is applied.

- **Target World Model**: We evaluate three configurations for the target world model:

  - **Soft Target (default)**: Leverages an Exponential Moving Average (EMA) target model (Mnih et al., 2013) for both the target latent state and target value.
  - **Hard Target**: Updates the target world model by hard-copying parameters every 100 training iterations.
  - **No Target**: Removes the target world model entirely, using the current world model to generate the target latent state.

Based on these ablation studies, we derive the following key insights:

**(1)    Training Context Length and Transformer Depth**:

- In the *Pong* environment, shorter inference contexts ($H_{\text{infer}} = 4$) outperform longer contexts ($H_{\text{infer}} = 8$) across all Transformer depths. This suggests that shorter contexts are sufficient for Atari tasks. Consequently, we set $H_{\text{infer}} = 4$ for all Atari experiments.
- In *VisualMatch*, however, the training context length must match the episode length to enable the agent to retain memory of the target color from the first phase. Accordingly, the training context length is set to $16 + \text{memory\_length}$.
- Figure 16 illustrates that deeper Transformer backbones slightly improve performance, indicating that increased capacity better captures long-term dependencies.

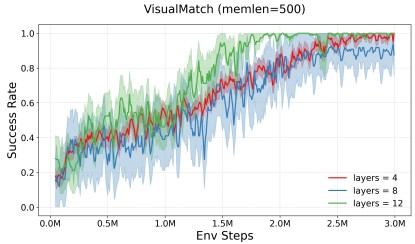

**Figure 16: Impact of Transformer depth in _VisualMatch_ (Memory Length = 500).** Performance improves slightly with the number of layers in the Transformer backbone, indicating that deeper architectures better capture long-term dependencies.

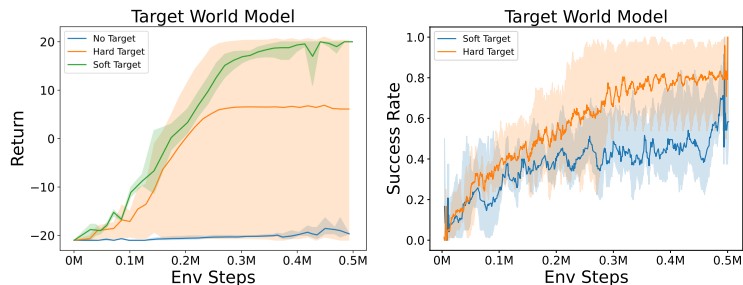

**Figure 17: Ablation results for the target world model.** Left: _Pong_, Right: _VisualMatch (MemoryLength = 60)_. Soft target models yield the most stable performance. The horizontal axis shows _Env Steps_, while the vertical axis represents the _Return_ or _Success Rate_ over 10 episodes. Results are averaged over 3 runs, with shaded areas denoting 95% confidence intervals.

**(2) Latent Normalization**:

- The _SimNorm_ method (Hansen et al., 2023) achieves the best performance, followed by _Softmax_, while _Sigmoid_ fails to converge.
- These results underscore the importance of proper normalization in the latent space for training stability. Specifically, _SimNorm_ enforces a fixed $L_1$ norm on latent states, which introduces natural sparsity and stabilizes gradient magnitudes.
- Without normalization, gradient explosion is frequently observed.

**(3) Decode Regularization**:

- Decode regularization has minimal impact on performance in both _Pong_ and _VisualMatch_.
- This suggests that latent states primarily encode task-relevant information, rendering the reconstruction of original observations unnecessary for effective decision-making.

**(4) Target World Model**:

- As shown in Figure 17, the soft target model delivers the most stable performance.
- The hard target model exhibits some instability, while removing the target world model leads to non-convergence in _Pong_ and NaN gradients in _VisualMatch_.
- This behavior aligns with the role of target networks in algorithms such as DQN (Mnih et al., 2013), where the absence of target stabilization mechanisms often causes divergence.

## E.2 World Model Analysis

**VisualMatch.** In Figure 18 and Figure 19, we present the _predictions of the learned world model_ in one _success_ and one _fail_ episode of _VisualMatch (MemoryLength=60)_, respectively. The first row indicates the predicted reward and true reward. The second row displays the original image frame. The third row outlines the predicted prior policy, and the fourth row describes the improved (MCTS) policy induced by MCTS based on the prior policy. For the sake of simplicity, we have only illustrated the first two steps ($t = \{2, 3\}$) and the last two steps ($t = \{60, 61\}$) of the _distraction_ phase. Please note that at each timestep, the agent performs the action with the highest probability value in the fourth row. As observed, the reward is accurately predicted in both cases, and the MCTS policy has shown further improvement compared to the initial predicted prior policy. For example, in Figure 18, at timestep 75, action 3, which represents moving to the right, is identified as the optimal action because the target color, green, is located on the agent's right side. While the predicted prior policy still allocates some probability to actions other than action 3, the MCTS policy refines this distribution, converging more towards action 3.

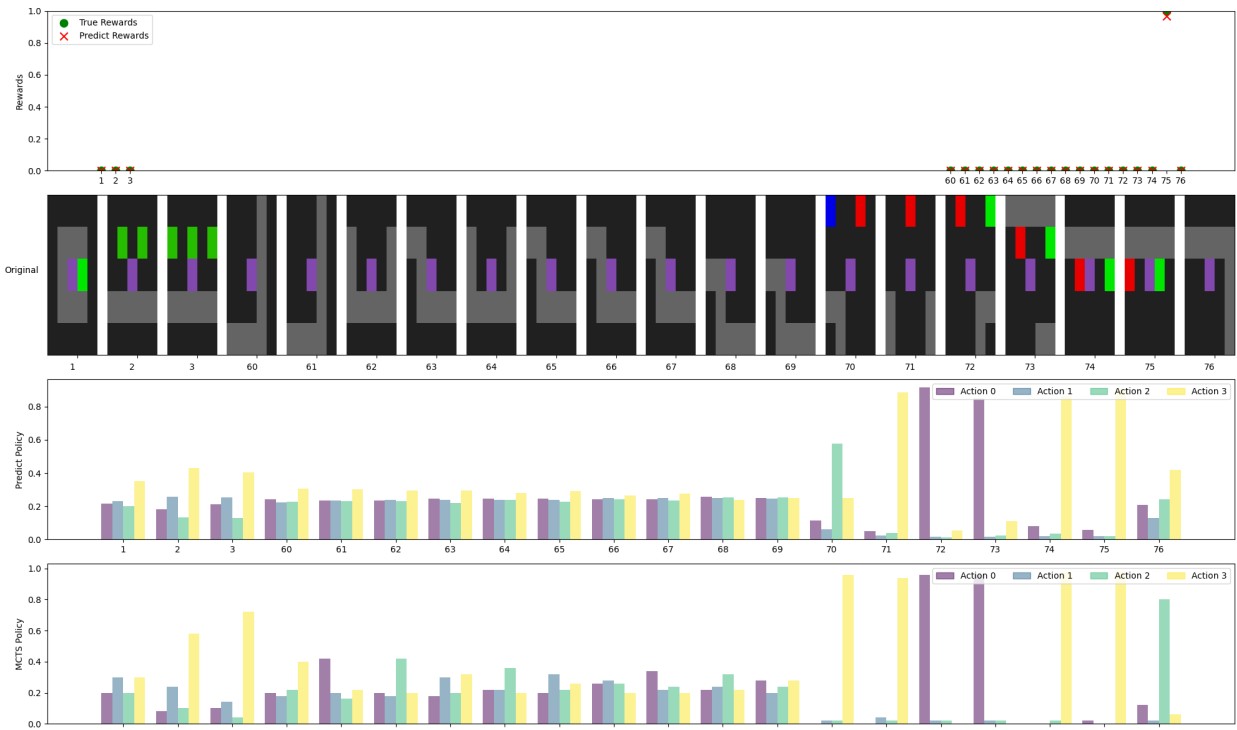

**Figure 18: Predictions of the world model** in one *success* episode of *VisualMatch (MemoryLength=60)*. The first row indicates the predicted reward and true reward. The second row displays the original image frame. The third row outlines the predicted prior policy, and the fourth row describes the improved (MCTS) policy induced by MCTS based on the prior policy. For the sake of simplicity, we have only illustrated the first two steps ($t = \{2, 3\}$) and the last two steps ($t = \{60, 61\}$) of the *distraction* phase. At timestep 75, action 3, which corresponds to moving to the right, is identified as the optimal action because the target color, green, is located on the agent's right side. Although the predicted prior policy assigns some probability to actions other than action 3, the MCTS policy refines this distribution, converging more decisively towards action 3.

Figure 21 shows the *attention maps* of the trained world model. It can be observed that in the initial layers of the Transformer, the attention is primarily focused on the *first time step* (which contains the target color that needs to be remembered) and the *most recent few time steps*, mainly for predicting potential dynamic changes. In higher-level layers, sometimes, such as in Layer3-Head2, the attention is mainly concentrated on the current time step, whereas at other times, such as in Layer4-Head4, there is a relatively broad and dispersed attention distribution, possibly indicating the fusion of some learned higher-level features.

**Pong.** Similarly, in Figure 20, we present the *predictions of the world model* in one trajectory of *Pong*. The first row indicates the predicted reward and true reward. The second row displays the original image frame. The third row outlines the predicted prior policy, and the fourth row describes the improved (MCTS) policy induced by MCTS based on the prior policy. Please note that the image in the second row (original image) has already been resized to (64,64) from the raw Atari image, so there may be some visual distortion. At each timestep, the agent performs the action with the highest probability value in the fourth row. Throughout all timesteps, the true reward remains zero due to the absence of score events. Unizero's world model can accurately predict this, with all predicted rewards consistently remaining zero. At the 8th timestep, the agent controlling the right green paddle successfully bounces the ball back. At the 7th timestep, the agent should perform the upward action 2; otherwise, it might miss the opportunity to catch the ball. The MCTS policy further concentrates the action probability on action 2 compared to the prediction policy, demonstrating the policy improvement process of MCTS.

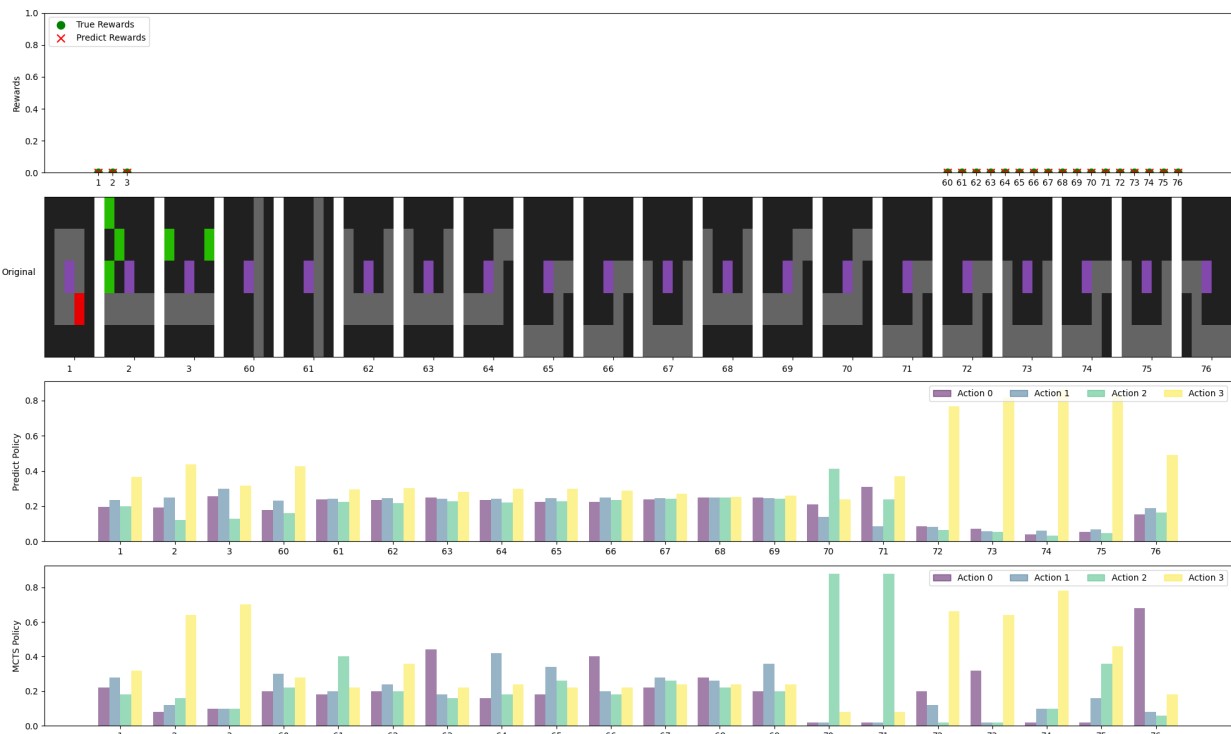

**Figure 19: Predictions of the world model** in one *fail* episode of *VisualMatch (MemoryLength=60)*. The first row indicates the predicted reward and true reward. The second row displays the original image frames. The third row outlines the predicted prior policy, and the fourth row describes the improved (MCTS) policy induced by MCTS based on the prior policy.

In Figure 22, we plot the *attention maps* in one trajectory (*train_context_length* is 10, with each time step consisting of two tokens, namely the latent state and the action) of *Pong*. It can be observed that across various levels, attention is primarily on data from the *most recent frames*. This is closely related to the *short-term dependency* characteristic of Pong. Utilizing information from only the recent frames is sufficient for dynamic prediction and policy-value learning.

### E.3 Covariate Shift in UniZero

MuZero and its recent variants Hubert et al. (2021a); Ye et al. (2021) are founded on the value-equivalence principle that guides world model training. These models comprise two primary components: a dynamics model that predicts the next latent state and reward, and a decision model that estimates both the policy and corresponding value. In the MuZero-style architecture, only the first step directly processes environment observations (e.g., multiple stacked frames or encoded history), while subsequent steps rely on recurrent predictions of latent states. This design minimizes discrepancies between training-time encoded observations and the predictions employed during Monte Carlo Tree Search (MCTS) at inference.

However, as noted in Ghugare et al. (2022b), MuZero-style latent models experience significant performance degradation with an increasing number of unroll steps during recurrent training. Our proposed UniZero addresses this limitation by employing a Transformer-based latent model that integrates new information at every step through advanced representation learning techniques, including self-supervised losses. Although this improvement enhances the overall representation and alleviates degradation over long unrolls, it inherently introduces covariate shift, consistent with the "no free lunch" principle.

In UniZero, the root node utilizes a key-value cache (kv-cache) to retain historical information, thereby preserving an unbiased process. Furthermore, during the search phase, the depth of the search tree is constrained, with long-term value estimation delegated to a dedicated value head. This design effectively

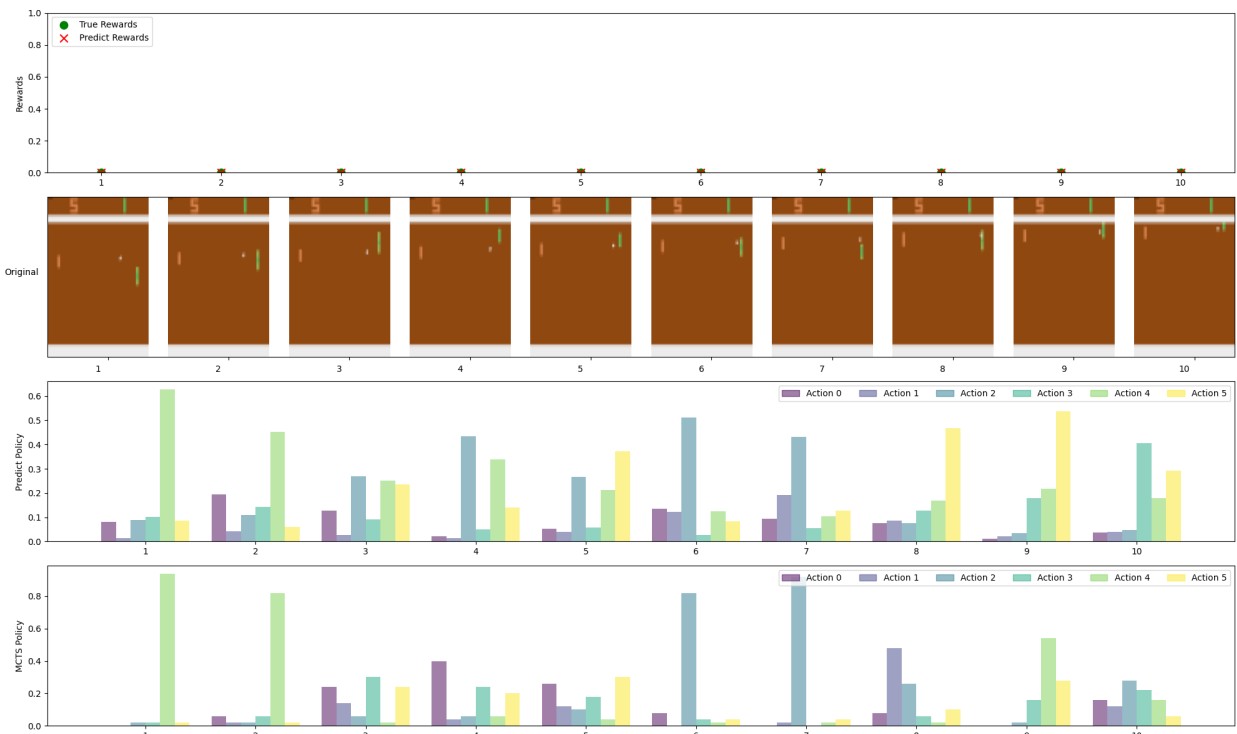

**Figure 20: Predictions of the world model** in one trajectory of *Pong*. The first row indicates the predicted reward and true reward. The second row displays the original image frame. The third row outlines the predicted prior policy, and the fourth row describes the improved (MCTS) policy induced by MCTS based on the prior policy.

minimizes biases in value and policy estimation that may arise from covariate shift. Future research will focus on both quantifying and mitigating the effects of this covariate shift, as well as exploring hybrid architectures that combine the strengths of recurrent and Transformer-based models to balance the trade-offs between extended unroll training and covariate shift.

## F    Comparison with Prior Works

**Comparison with recent approaches in world modeling.**    To provide a clear comparison, we present Table 12 outlining the key differences between UniZero and recent approaches (Hansen et al., 2023; Schrittwieser et al., 2019; Hafner et al., 2023; Micheli et al., 2022; Robine et al., 2023; Zhang et al., 2023) in world modeling. The attributes considered include the type of sequence model used, the input information introduced during a single timestep, the method for obtaining latent representations, the approach to policy improvement, and the training pipeline.

- *Sequence Model*: The architecture employed for modeling sequences.

- *Input*: The type of information fed into the sequence model at each timestep, where "Latent history" refers to the recurrent/hidden state as described in the respective papers.

- *Latent Representation*: This refers to the technique employed to extract embeddings from each observation. For instance, an "Encoder" might be a neural network such as a Convolutional Network (ConvNet) for processing images or a Multi-Layer Perceptron (MLP) for handling vector observations. The term "VQ-VAE" (van den Oord et al., 2017) denotes the vector-quantized VAE, which is utilized to obtain a discrete code for the observation. Similarly, "Categorical-VAE" (Hafner et al., 2023) represents the discrete VAE, which is used to derive the discrete distribution of the observation.

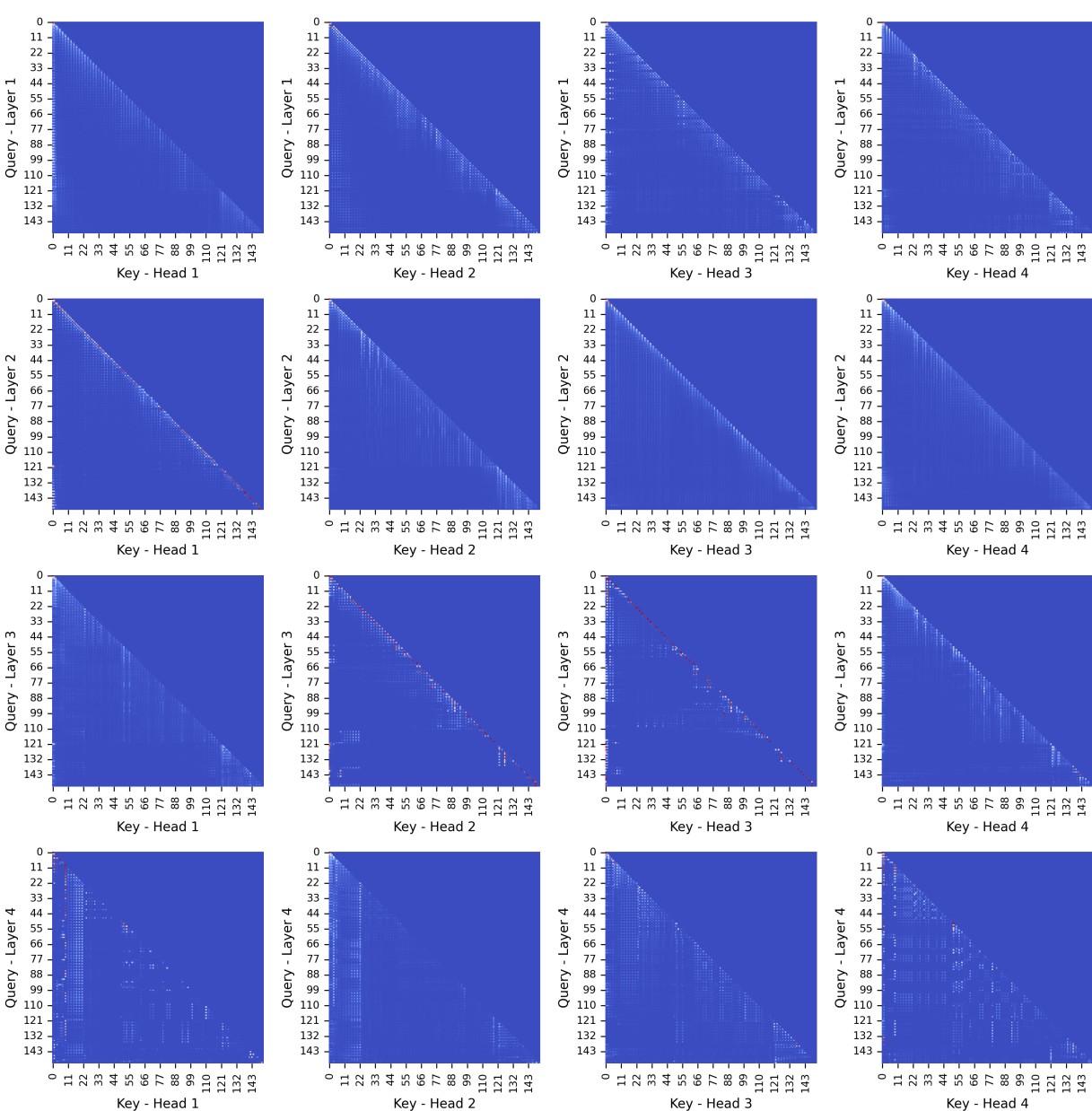

**Figure 21: Attention maps** in one *success* episode of *VisualMatch (MemoryLength=60)* (Note that the *train_context_length* is $1 + 60 + 15 = 76$, with each time step consisting of two tokens, namely the latent state and the action.). It can be observed that in the initial layers of the Transformer, the attention is primarily focused on the *first time step* (which contains the target color that needs to be remembered) and the *most recent few time steps*, mainly for predicting potential dynamic changes. In higher-level layers, sometimes, such as in `Layer3-Head2`, the attention is mainly concentrated on the current time step, whereas at other times, such as in `Layer4-Head4`, there is a relatively broad and dispersed attention distribution, possibly indicating the fusion of some learned higher-level features.

- *Policy Improvement*: The method for enhancing the policy, with "PG" standing for Policy Gradient methods (Schulman et al., 2017; Hafner et al., 2023) and "MPC" standing for Model Predictive Control (Hansen et al., 2022).

- *Training Pipeline*: The training process involves a "two-stage" approach, where we first train the world model and then use the learned model to train the policy (behavior) through imagination. On the other hand, "model-policy joint training" refers to simultaneously learning the world model and the policy (and

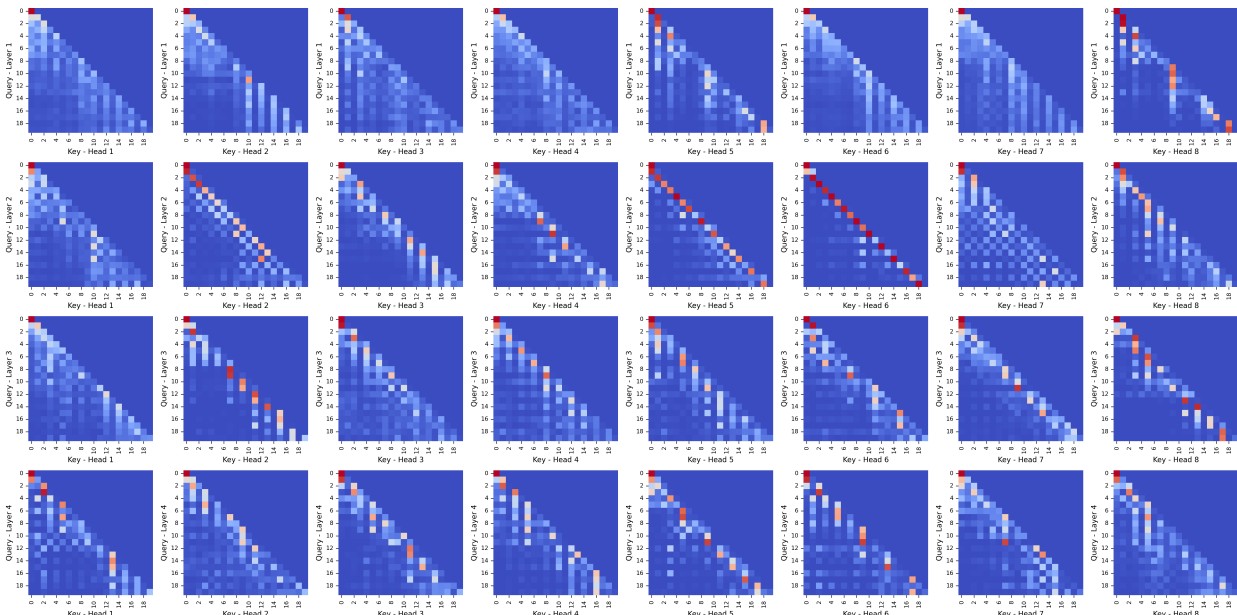

**Figure 22: Attention maps** in one trajectory (*train_context_length* is 10, with each time step consisting of two tokens, namely the latent state and the action.) of *Pong*. It can be observed that across various levels, attention is primarily on data from the *most recent frames*. This is closely related to the *short-term dependency* characteristic of Pong. Utilizing information from only the recent frames is sufficient for dynamic prediction and policy-value learning.

value), rather than following a two-stage process. This joint training approach offers several benefits, as discussed in Eysenbach et al. (2022); Ghugare et al. (2022b); Grill et al. (2020).

**Table 12: Comparison between UniZero and recent model-based RL approaches**. The main difference between UniZero and MuZero is highlighted in bold.

| Attributes | Sequence model | Input | Latent representation | Policy Improvement | Training Pipeline |
|---|---|---|---|---|---|
| TWM (Robine et al., 2023) | Transformer-XL (Dai et al., 2019) | Latent, observation, action, reward | Categorical-VAE | PG of DreamerV2 (Hafner et al., 2020) | two-stage |
| IRIS (Micheli et al., 2022) | Transformer (Vaswani et al., 2017) | Latent, observation, action | VQ-VAE | PG of DreamerV2 (Hafner et al., 2020) | two-stage |
| DreamerV3 (Hafner et al., 2023) | GRU (Cho et al., 2014) | Latent, observation, action | Categorical-VAE | PG of DreamerV3 | two-stage |
| STORM (Zhang et al., 2023) | Transformer | Latent, observation, action | Categorical-VAE | PG of DreamerV3 | two-stage |
| TD-MPC2 (Hansen et al., 2023) | MLP | Latent, observation, action | Encoder (with *SimNorm*) | MPC (Hansen et al., 2022) | two-stage |
| MuZero (Schrittwieser et al., 2019) | MLP | Latent, action | Encoder | MCTS | Model-policy Joint training |
| UniZero (ours) | **Transformer** (Lee et al., 2023) | **Latent, observation, action** | Encoder (with *SimNorm*) | MCTS | Model-policy Joint training |

**Comparison with MuZero's Extensions.** In recent years, MuZero and its extended algorithms have significantly improved efficiency and stability across various scenarios through a series of innovations. To facilitate understanding of the relationship between UniZero and the MuZero family of algorithms, we provide a qualitative comparison across the following key dimensions:

- *Action Space*: The type of action space supported by the algorithm (continuous or discrete).

- *Simulation Cost*: The computational cost of simulating latent states during the search process.

- *Sample Efficiency*: The effectiveness of utilizing sampled data for learning.

- *Explicit Stochasticity Modeling*: Whether the algorithm explicitly models the stochasticity in the environment.

The core ideas of each extended algorithm are summarized as follows:

- **Sampled MuZero** (Hubert et al., 2021b): A sample-based policy iteration framework that applies to any type of action space. It computes improved policies over a subset of the original action space and probabilistically converges to the optimal policy over the entire action space as the number of samples increases.

- **Gumbel MuZero** (Danihelka et al., 2022): is designed to enhance performance in environments with low simulation costs by leveraging the Gumbel-Top-k trick (Kool et al., 2019) to select actions that guarantee policy improvement. It seamlessly integrates the original visit counts distribution with MCTS searched values to more informed decision-making and improved performance.

- **EfficientZero** (Ye et al., 2021): Incorporates techniques such as self-supervised consistency loss, end-to-end prediction of value prefixes, and model-based off-policy correction. These enhancements significantly improve sample efficiency, achieving outstanding performance in tasks like Atari 100k.

- **Stochastic MuZero** (Antonoglou et al., 2021): Introduces stochastic modeling (including afterstates) and employs stochastic tree search to effectively handle randomness in the environment. It demonstrates superior performance in tasks such as 2048, Backgammon, and Go.

Table 13 provides a qualitative comparison of UniZero and the aforementioned MuZero family algorithms across key dimensions. It is worth noting that the improvements introduced in the MuZero family of extended algorithms are largely orthogonal to those in UniZero, making them easily transferable to UniZero. We consider integrating these extensions as part of our future work.

**Table 13:** Qualitative Comparison of UniZero and MuZero Family Algorithms Across Different Dimensions.

| Algorithm | Action Space | Simulation Cost | Sample Efficiency | Explicit Stochasticity Modeling |
|---|---|---|---|---|
| UniZero | continuous/discrete | medium | medium | no |
| Sampled MuZero | continuous/discrete | low | medium | no |
| Gumbel MuZero | discrete | low | medium | no |
| EfficientZero | discrete | medium | high | no |
| Stochastic MuZero | discrete | medium | medium | yes |

