# OpenReview forum: "UniZero: Generalized and Efficient Planning with Scalable Latent World Models"
_TMLR — Accepted by TMLR_

### Review · Reviewer_ijGr · 2025-02-07

**Summary Of Contributions:**

The paper proposes "UniZero", a variation of MuZero which uses a Transformer as world model. The main motivation is to allow the application of MuZero-style approaches to POMDPs in which the current state does not carry enough information.

**Audience:**

Yes

**Claims And Evidence:**

No

**Requested Changes:**

Please make the motivation, and limitations of UniZero clearer throughout the paper. Furthermore, for non-POMDP environments, please explain why one should expect UniZero to outperform a non-transformer based approach.

Before those changes, I cannot recommend acceptance. Furthermore, as this would require a sizable change to the current paper, I'd recommend a re-review after those changes have been made.


EDIT: After extensive discussions, the authors have added several of the requested changes.

**Strengths And Weaknesses:**

## Strengths:
* The idea of leveraging transformers as dynamics ("world") model for MCTS is interesting.

## Weaknesses

### Writing
IMO, the main weakness of the paper is that it makes a lot of vague, imprecise and confusing claims and I found it on a first pass unnecessarily hard to understand what the actual motivation (and contribution) of the paper was. For example, IMO, the main justification for using a transformer is to allow MuZero-style algorithms to apply to POMDPs where history is important. However, this is not clearly stated in the paper (POMDPs are only tangentially mentioned), instead the authors make claims about "scale[ing] in heterogeneous scenarios with diverse dependencies and task variability" or "intrinsic entanglement between latent representations and historical information" or "under-utilization of trajectory data".

### Experiments

* Baselines: I believe the obvious baseline, which is missing, is to use either a RNN or a Transformer to simply encode the history into a latent state `z_t` and use this as the initial state for the standard MuZero algorithm. In other words, replace the Encoder in MuZero with an RNN. Instead, the authors use "MuZero w/ Context" which ignores the last `k` states from the environment.
* On Figure 2: I find it worrying that UniZero-RNN and and MuZero w/ Context cannot solve the fully observable game Pong.  For "MuZero w/ Context", I believe (see above) that this might make sense and this baseline should be replaced. But For UniZero-RNN I don't see why it shouldn't solve Pong.
* Similarly, I'm not sure I understand why UniZero outperforms the baselines on (standard) Atari and Mujoco, both of which are fully observable environment where the Transformer should not bring any advantages (as historical states are irrelevant).

---

> ### Author Response · Authors · 2025-03-03
> **Author response to reviewer ijGr (1/2)**
>
> Thank you very much for your valuable time and detailed review comments. Below are our responses to your questions:
>
> **Regarding Writing**
> - Thank you very much for your suggestions, which have helped improve the clarity of our writing. However, it is important to note that reviewers cD6a and 4J8R generally agreed with the logic and main conclusions of our paper, only pointing out a few areas that could be further clarified. For example, they suggested providing a clearer explanation of related work, elaborating on the scalability of multi-task learning, and improving the clarity of some experimental descriptions. We have already addressed these points in our responses under other reviewer panels and have made corresponding revisions to the paper.
> - To validate the effectiveness of our proposed "intrinsic entanglement" hypothesis, we further analyzed the changes in MuZero's latent state after training for 100k iterations in Pong. Specifically, we set the unroll step to 5 and used the average latent cosine similarity between different unroll steps as well as the pearson correlation of consecutive unroll steps within the same minibatch as quantitative metrics to measure the degree of "entanglement." Experimental results show that for both the original MuZero and MuZero w/ SSL, the overall Cosine Similarity and Pearson Correlation are close to 1, and Pearson Correlation further increases as the unroll step grows. This indicates a high correlation among latent states during recurrent training, partially confirming the presence of "intrinsic entanglement."
>
> **Regarding UniZero-RNN**
> - It should be noted that in Figure 2, we only tested 500K envsteps. UniZero-RNN is not entirely incapable of learning; rather, due to the optimization efficiency of RNNs themselves, solving Pong in an online RL setting—where the data distribution is constantly changing—may require more training steps. In the field of reinforcement learning, using RNNs for training typically requires careful design. For example, the initial hidden state needs to be obtained through a burn-in process (as referenced in the R2D2 paper [1]). Our baseline method directly initializes from zero, which may also be one of the reasons for its weaker performance in the early stages.
>
> (1/2)

---

> ### Author Response · Authors · 2025-03-03
> **Author response to reviewer ijGr (2/2)**
>
> **Regarding Baseline Methods**
> - The ablation study in Figure 2 is primarily conducted to verify potential issues in MuZero under the recurrent training paradigm. Although "encoding historical information into a latent state z_t using an RNN or Transformer and using it as the initial state in the standard MuZero algorithm" is indeed a valid variant of MuZero, we want to emphasize that this baseline approach does not resolve the incompatibility between MuZero's recurrent training paradigm and self-supervised learning (SSL) in a POMDP setting. Therefore, we did not include the method you mentioned as a baseline in our previous work. However, we can further explore this direction in the future.
> - To address this issue, we drew inspiration from training paradigms widely adopted in recent video generation research (e.g., CogVideoX [2], Sora, etc.), where each time step is first encoded (via VAE, VQVAE, or other encoding methods) before utilizing a Transformer to perform fusion in the latent space. This approach has been validated as a general and scalable training strategy, and our UniZero framework largely follows this paradigm.
>
> **Regarding "Why Does UniZero Outperform Baselines in Non-POMDP Atari and Mujoco Tasks?"**
> - First, we have not claimed that using Transformer in UniZero would bring significant performance improvements in single-task MDP environments. Our primary focus is on enhancing the scalability of MuZero, particularly in heterogeneous scenarios such as environments with varying historical dependency lengths and multi-task learning. This has been validated in the VisualMatch and multi-task Atari environments.
> - The experiments on Atari and Mujoco tasks were primarily conducted to verify whether UniZero can maintain performance comparable to the baseline in non-memory-dependent (Non-Memory Domains) environments. In some environments, UniZero demonstrated certain advantages, which we speculate may be due to: (1) Some Atari tasks inherently exhibit POMDP characteristics (e.g., Pong and Frostbite, as noted in DRQN [3]); (2) Predicting further into the future may aid representation learning (see [4] for reference).
>
> We hope this addresses your concerns. If you have any further suggestions, we would be very grateful for your feedback. Thank you very much.
>
> [1] Kapturowski, Steven, et al. "Recurrent experience replay in distributed reinforcement learning." International conference on learning representations. 2018.
>
> [2] Yang, Zhuoyi, et al. "Cogvideox: Text-to-video diffusion models with an expert transformer." arXiv preprint arXiv:2408.06072 (2024).
>
> [3] Hausknecht, Matthew J., and Peter Stone. "Deep Recurrent Q-Learning for Partially Observable MDPs." AAAI fall symposia. Vol. 45. 2015.
>
> [4] Fang, Ching, and Kimberly L. Stachenfeld. "Predictive auxiliary objectives in deep RL mimic learning in the brain." arXiv preprint arXiv:2310.06089 (2023).
>
> (2/2)

---

> > ### Author Response · Authors · 2025-03-07
> > **Author response to reviewer ijGr**
> >
> > Dear Reviewer,
> >
> > As the discussion phase is coming to an end, we kindly invite you to review our responses to your comments. If you have any further questions, please don’t hesitate to reach out. If no further concerns remain, we would greatly appreciate it if you updated your recommendation.
> >
> > Best regards,
> >
> > Authors of UniZero (Submission 3998)

---

> ### Comment · Reviewer_ijGr · 2025-03-16
> **Thank you for your reply.**
>
> Dear Authors,
>
> Thank you for your reply and further explanations.
>
> I have read your replies, the other reviews, and the revised version of the paper.
>
> I maintain my point that the paper is often written unnecessarily convoluted and vague.
> To just give some examples of what I mean:
> * "Heterogeneous scenarios with diverse dependencies and task variability" = Multitask? (Also, I'm still not sure what "diverse dependencies" refers to here.)
> * "facilitating broader and more efficient planning" - what does broad planning mean? What does "efficient" planning mean? why would UniZero plan more efficiently by providing historical information? Did you mean more efficient training?
> * "retrospective and prospective cognitive functions in AI" = memory and planning
> * The whole "under-utilization" argument is never really explain well IMO.
>
> Furthmore, unlike suggested in your rebuttal, my reading of 4J8R's review seems largely in line with mine (albeit maybe less strong in their opinion) on the topic of clarity, e.g. writing "I thought the discussion around the limitations of MuZero could be a lot clearer.".
>
> However, **more importantly**, on the discussion about baselines:
> I'm not sure what you are referring to with "we want to emphasize that this baseline approach does not resolve the incompatibility between MuZero's recurrent training paradigm and self-supervised learning (SSL) in a POMDP setting".
> Irregardless, my main point is that the baselines in the article appear unnecessary weak:
>
> For one, the overwhelming dominating of UniZero on PongNoFrameskipv4, where none of the UniZero novelty _should_ matter, indicates to me that maybe baselines have not been sufficiently hyperparameter tuned or have bugs.
>
> However, even more importantly, MuZero w/ Context is a terrible baseline for POMDPs, or even MDPs, because _it ignores all recent observations_ except the one at `t-k`. I believe this is what you are referring to with "under-utilization" or "incomplete context" in Fig 1 - which I agree with. But I don't think this is a fair baseline because nobody would actually use this.
>
> If we zoom out to what I believe is your main argument that "entanglement" between current state and history is bad, I see effectively two different types of _fair_ architectures to integrate history into MuZero:
> * The "entangled version", i.e. "MuZero with history encoder": Take the history + current state and encode it into one latent state, from which you run the normal MuZero. Note that this is absolutely valid because we effectively say that by combining history + current state, we allow the history encoder to infer our current best guess of what the true state of the environment is, i.e. the "believe state" in POMDPs. (Btw, this is why I think formulating your arguments with the language of POMDPs would help readability). The MuZero acting on this believe state now does planning in believe space. In this architecture, the "history encoder" can be an RNN or also a Transformer from which you take the last latent state as root note for "standard" MuZero. Note that in this architecture the RNN/Transformer would only be used to encode the history, not in the forward planning step of MuZero itself!
> * "UniZero", where you learn a recurrent world model and keep the individual states "explicit". Then you use this recurrent world model inside the MuZero forward planner.
>
> Both are valid approaches and I can see the UniZero approach to be beneficial, but I think the first approach is a crucial baseline to compare against as it is the (maybe only) reasonable baseline I can see.
>
> One more point about the experiments: You write that the bad performance of the RNN method on the Pong experiment is due to "GRU's limited memory capacity". I don't think this is true in this context, as Pong only requires _one_ frame history to correctly infer all needed information (which here is the velocity and direction of the ball).

---

> ### Author Response · Authors · 2025-03-17
> **Author response to reviewer ijGr (1/2)**
>
> Dear Reviewer,
>
> We sincerely appreciate your detailed feedback and valuable suggestions. Your insights have helped us improve our manuscript, and we would like to clarify a few key points from your review. Our responses to your concerns are detailed below.
>
> **Discussion on Baseline Methods**
> - **Strength of Baselines**:
>  We respectfully disagree with the characterization that our baselines are “unnecessarily weak.” The primary aim of the experiments presented in Figure 2 is to expose the inherent limitations of the standard MuZero training paradigm rather than to serve as the sole benchmarks for our study. In subsequent experiments, we have compared our method against other state-of-the-art techniques (e.g., SAC-GPT for the VisualMatch environment and Dreamerv3 for the DMC environment) which are recognized as strong baselines in their respective domains.
> - **On the “under-utilization” Issue**:
> As you correctly noted, the MuZero w/ Context baseline suffers from an inherent “under-utilization” issue. What we want to emphasize is that, in the *standard MuZero training paradigm*, *only the first* raw observation (although presented as stacked frames) is transformed into a latent state by the encoder (or history encoder, as you suggested). For subsequent steps, the dynamics network recursively predicts rewards and latent states primarily based on this initially encoded state. During the unroll stage (illustrated in the deep-blue module of Figure 1), new observational information is not actively incorporated into the computation, leading to the fundamental “under-utilization” issue during training. We provide further details in Section 3.1 of our paper, and the additional reference [1] highlights the importance of leveraging comprehensive trajectory information in training.
> - **MuZero with History Encoder**:
> We acknowledge the validity of your proposed "MuZero with history encoder" as a baseline and recognize that its performance may surpass MuZero w/ Context. However, this method only improves the integration of information before the first step of the training sequence, without any modifications to how information is processed after the first step. While this improvement may be sufficient in environments dominated by short-term dependencies, such as Pong, it may lack flexibility and efficiency in handling more complex dependency scenarios (e.g., longer dependency problems in VisualMatch or multi-task learning environments). Additionally, as you mentioned, for certain Atari tasks such as Pong, a stack size of 4 provides sufficient information for decision-making. In Figure 2, MuZero w/ SSL (stack4) can be considered an upper bound estimate of the performance of the "MuZero with history encoder" algorithm. However, to validate this, we will provide baseline test results for "MuZero with history encoder" on Pong and VisualMatch in the coming days.
> - **UniZero (RNN) Variant**:
>  To overcome the limitation of processing only the initial observation, we propose the UniZero (RNN) variant (see Appendix, Figure 9). Our RNN architecture integrates observations at every training step, thereby addressing the “under-utilization” problem in a principled manner. However, as noted earlier, training RNN-based methods in reinforcement learning necessitates careful design (e.g., special management of hidden states), which may partly explain its suboptimal performance in our experiments.
> - **Performance on PongNoFrameskip-v4**:
>  Regarding the exceptional performance of UniZero on PongNoFrameskip-v4, we emphasize that this outcome was observed under the stack=1 configuration. Under the stack=4 setting, MuZero w/ SSL— which partially exploits full trajectory information via self-supervised loss—achieves comparable sample efficiency to UniZero. To ensure a fair comparison, we did not conduct specialized hyperparameter tuning for either the baseline methods or UniZero. The specific hyperparameter settings are provided in Table 8 and Table 9 of the appendix. While further tuning might enhance performance, it does not impact our main conclusions. To facilitate reproducibility, we will open-source our codebase after the review process is complete.
>
> (1/2)

---

> ### Author Response · Authors · 2025-03-17
> **Author response to reviewer ijGr (2/2)**
>
> **Simplicity and Clarity of the Manuscript**
> - We agree that additional clarity would benefit the manuscript, and we have revised the text accordingly. While we do not share the view that our original presentation was “unnecessarily convoluted and vague,” we acknowledge that further simplification can enhance readability. To this end, we have clarified the following points:
>   - **Diverse Dependencies**:
>  By “dependencies,” [2] we refer to the complexity inherent in the relationships that the model (e.g., its policy and value networks) must capture to solve a task. For example, if we measure it by *memory length* [2]，in VisualMatch, the agent must consistently attend to and process critical historical information (from the first frame) throughout the task, requiring the value function to appropriately weigh historical inputs. In contrast, many Atari games typically require information from only the last few frames (commonly four) to achieve satisfactory performance.
>   - **Broad and Efficient Planning**:
>  We define “broad planning” as the algorithm's versatility in adapting to a variety of environmental settings—spanning different input modalities (e.g., images or vectors) and diverse action spaces (discrete or continuous). While “efficient planning” reflects the algorithm’s ability to attain superior or comparable sample efficiency across these heterogeneous environments. Our experimental results, presented later in the paper, substantiate these claims. The description of "Efficient Planning" can also be found in the paper [3].
> - **Clarification on “GRU’s limited memory capacity”**:
>  We appreciate your suggestion on this point. As mentioned earlier, the poor performance of the GRU may be due to the need for careful handling of the initial hidden state during training. We will revise this description in the updated version of the paper.
>
> We hope that the above clarifications and revisions sufficiently address your concerns. Should you have any further feedback, we would be happy to engage in further discussion.
> Thank you once again for your constructive feedback.
>
> Sincerely,
>
> The Authors of UniZero (Submission 3998)
>
> [1] de Vries, Joery A., et al. "Visualizing muzero models." arXiv preprint arXiv:2102.12924 (2021).
>
> [2] Ni, Tianwei, et al. "When do transformers shine in rl? decoupling memory from credit assignment." Advances in Neural Information Processing Systems 36 (2023): 50429-50452.
>
> [3] Jiang, Zhengyao, et al. "Efficient planning in a compact latent action space." arXiv preprint arXiv:2208.10291 (2022).
>
>  (2/2)

---

> > ### Comment · Reviewer_ijGr · 2025-03-18
> > **Thank you**
> >
> > Dear Authors,
> >
> > Thank you for your fast and detailed response. I now understand some of your arguments better and I think I see now where some our disagreement around baselines arises.
> >
> > My argument regarding weak baseline is w.r.t to the "incomplete context" of "MuZero w/ Context" - if I understand it correctly based on Fig1, the policy does not act on the full information it has up to the current point in time, so it cannot perform well - hence I don't think this policy makes sense as a baseline as nobody would actually expect it to work well. My suggestions for the history encoder was to fix this, which I believe constitutes a more reasonable baseline. The accumulation of errors that you mention is another issue, but I believe less impactful as the lack access to up-to-date information.
> >
> > "Under-utilization" vs "full-utilization": I think one has to differentiate between using data on the level of the _training algorithm_ vs. on the level of _one forward pass_ during training. UniZero uses more information in one forward pass, but that doesn't necessarily mean it's less data-efficient on the level of the training algorithm. I don't think that discussion features in the paper? One could infer it from Fig2, but that might also be just down to insufficient hyperparameter tuning for the baselines.
> >
> > Entanglement: You say that "entanglement [of history and current state] is fundamentally incompatible with the SSL loss, as discussed later" - however I wasn't able to find this discussion. It's also not clear to me why it is incompatible. Of course, the "entangled" latent state cannot directly be compared to the encoded current state - but a projection of the entangled state, which is able to "filter" for only the current information, should be able to be used in an SSL loss?
> >
> > I think fundamentally your argument for UniZero is the same argument that Transformers have over RNNs, i.e. that attention is better than recursively updating a latent state to remember historical information (i.e. using an "entangled state"). Which I do agree with. But I think your comparison with "MuZero w/ context" is a strawman as that baseline just doesn't make sense. A more interesting comparison would be with MuZero, which uses a Transformer to encode the history into _one_ latent state, on top of which one uses the MuZero planning. This baseline would allow one to nicely compare the tradeoffs:
> > * MuZero w/ Transformer History Encoder: Needs to compress historical context into one latent state ("entanglement"), but doesn't suffer from covariate shift.
> > * UniZero: Doesn't suffer from "entanglement", but suffers from covariate shift. Covariate shift is not discussed in the paper, but I think should also be, even though it's typical of world-model algorithms. What I mean with it is that during training, the world-model only receives `z` as input, while during inference it receives `\hat{z}` as input - leading to a possible distribution shift in the inputs (=covariates).

---

> ### Author Response · Authors · 2025-03-21
> **Thank you very much for your insightful analysis.  (1/2)**
>
> Dear Reviewer,
>
> Thank you very much for your thorough and insightful analysis.
>
> We fully agree with your point that MuZero with a Transformer History Encoder serves as a reasonable and effective baseline. At present, we have implemented a preliminary version where historical information is combined with the current state (by setting history_length to determine the length of history) and encoded into a latent state, followed by planning using standard MuZero. Our initial findings are as follows:
> - For Pong (stack = 1): We set history_length = 4, encode each frame using the original MuZero encoder, and aggregate them to generate a latent state. The subsequent training and inference processes remain consistent with standard MuZero. However, our preliminary experimental results indicate that this approach struggles to converge. We hypothesize that one possible reason is the lack of effective state supervision signals.
> - For VisualMatch (stack = 1, memory_length = 60): We use a Transformer History Encoder to encode the entire historical information along with the current state into a latent state (i.e., setting history_length to be the total length of all current states). Preliminary experimental results show that the success rate fluctuates around 0.4.
> - We are carefully reviewing all implementation details to ensure the accuracy of our experimental conclusions and will provide updates as more information becomes available.
>
> Additionally, we acknowledge your point regarding the difference between using data on the level of the training algorithm vs. on the level of one forward pass during training. However, from a theoretical perspective, compared to MuZero, UniZero is able to leverage information more effectively within a single forward pass. For environments like VisualMatch, which exhibit long-term dependencies, this advantage of "full utilization" has been validated both intuitively and experimentally. Furthermore, we would like to emphasize that MuZero *does not* directly use an RNN; rather, it relies solely on a recursive mechanism, meaning that no information is explicitly passed other than the first step. This is precisely the core issue of "under-utilization" that we have highlighted.
>
> In fact, the current MuZero with SSL loss has already been implemented in a manner consistent with the method described in [1] and is also similar to your suggestion. Theoretically, this approach should enhance sample efficiency while allowing the latent state to better preserve historical information. However, due to the entanglement between the latent state and historical information, the objective introduced by SSL (precisely predicting the next latent state) *differs* from the objective implied in the original loss function (fully retaining historical information necessary for decision-making). We refer to this phenomenon as incompatibility. We have mentioned this issue in the description of this variant and plan to refine the wording in future versions. Furthermore, in long-term memory environments, the experimental results in Figure 4 have already indirectly demonstrated the limited effectiveness of this approach. Notably, the MuZero variant in Figure 4 actually incorporates SSL loss, as clarified in the appendix. Unless otherwise stated, all experiments assume the SSL loss version by default.
>
> Regarding the issue of covariate shift, although the Encoder in MuZero w/ Transformer History Encoder itself is not directly affected, MCTS— as shown in the lower-left corner of Figure 1— relies on models learned through recursive training (including the dynamics model) for inference. During the tree search process, prediction errors may still lead to discrepancies between training and inference [2]. Therefore, we believe that covariate shift is not unique to UniZero but rather an inherent challenge for model-based reinforcement learning methods, including MuZero. While its impact on performance does not currently appear to be significant, we will incorporate a brief analysis of this issue in the revised version of the paper based on your suggestion.
>
> (1/2)

---

> ### Author Response · Authors · 2025-03-21
> **Thank you very much for your insightful analysis. (2/2)**
>
> Our UniZero inference design (as shown in the lower right corner of Figure 1) can, to some extent, be divided into two key components: a Transformer history encoder and a Transformer world model. The KV cache mechanism implicitly implements the function of the Transformer historical encoder in capturing historical information, while during root node search, the Transformer world model leverages all available information for attention-based  inference. During training, UniZero follows a modular structure and a parallel paradigm (as shown in the upper right corner of Figure 1), allowing it to conveniently benefit from the latest advancements in the Transformer field.
>
> We sincerely appreciate your detailed analysis and valuable suggestions, which are highly beneficial for improving our research. We hope that the above clarifications and modifications sufficiently address your concerns. If you have any further feedback, we would be very happy to engage in further discussion.
>
> Thank you again for your constructive feedback!
>
> Sincerely,
>
> The Authors of UniZero (Submission 3998)
>
> [1] Ye, Weirui, et al. "Mastering Atari games with limited data." Advances in Neural Information Processing Systems 34 (2021): 25476-25488.
>
> [2] de Vries, Joery A., et al. "Visualizing muzero models." arXiv preprint arXiv:2102.12924 (2021).
>
> (2/2)

---

> > ### Comment · Reviewer_ijGr · 2025-03-22
> > **Thank you**
> >
> > Dear Authors,
> >
> > Thank you for the addition of those baselines as well as the reformulations you suggested, I believe those will make the paper clearer and stronger.
> >
> > On more point on the issue of covariate shift: I agree that this is always an issue - however, there's a qualitative difference between MCTS and MuZero: For MCTS the generation follows the exact same process during training and inference: Start from a real state (or latent state encoded from a real state) and roll out the MCTS forward model - so why errors can happen, they should be similar between training and inference!
> > For MuZero there's a difference in the process of how those states are generated between training and inference: During training, latent states are generated from real states and during inference they are generated from the world model.

---

> ### Author Response · Authors · 2025-03-22
> **Thank you very much for your constructive feedback.**
>
> Dear Reviewer,
>
> Thank you for your constructive feedback, which has been immensely helpful in refining our manuscript.
>
> We believe there may have been a typographical error in your comment—were you perhaps comparing MuZero and UniZero? MuZero and its variants (including UniZero) are all based on MCTS. However, traditional MCTS typically performs search within a real environment simulator, balancing exploration and exploitation through UCB-like formulas, and this process does not involve training a model. In contrast, MuZero and its variants (including UniZero) are based on the value-equivalence principle to train a world model, which consists of:
> - A dynamics model that predicts the next latent state and reward.
> - A decision model that estimates the policy and corresponding value.
>
> The search process is conducted within this world model. MuZero adopts a recurrent mechanism, where only the first step incorporates environment observations (which may be multiple stacked frames or encoded history information via a history encoder), while subsequent steps predict latent states recurrently. Since no new environment observations are introduced after the first step, this avoids the discrepancy in UniZero between $z_t$ (encoded from environment observations during training) and $\hat{z}_t$ (used in MCTS search during inference).
>
> However, as analyzed in [1], MuZero-like latent models suffer a sharp performance drop as the number of unroll steps increases in recurrent training. UniZero mitigates this issue through an improved architecture (Transformer-based structure) and enhanced representation learning (incorporating new information at each step and using self-supervised losses). While this inevitably introduces covariate shift, it is a trade-off in line with the "no free lunch" principle. Our experimental results demonstrate that UniZero achieves a well-balanced trade-off between long unroll steps training and covariate shift. Specifically:
>
> - At the root node, we use kv-cache to represent historical information, ensuring an unbiased process.
> - During search, the depth of the search tree is generally not too large (longer-term value estimation is handled by the value head), so the bias in value and policy estimation caused by covariate shift is negligible.
>
> We sincerely appreciate your insightful analysis. We hope that our clarifications and revisions have adequately addressed your concerns. Thank you once again for your constructive feedback.
>
> Sincerely,
>
> The Authors of UniZero (Submission 3998)
>
> [1] Ghugare, Raj, et al. "Simplifying model-based RL: Learning representations, latent-space models, and policies with one objective." arXiv preprint arXiv:2209.08466 (2022).

---

> > ### Comment · Reviewer_ijGr · 2025-03-23
> > **Apologies for the typo**
> >
> > Dear Authors,
> >
> > I apologize for the misnaming, I wasn't careful enough when writing my comment. It should of course be MuZero and UniZero.
> > My comment was that UniZero introduces a new "type" of covariate shift during inference that isn't present in MuZero, namely that the input latent states during MCTS rollouts are generated differently between training and inference: During training from real-state encodings, during inference from world-model predictions. And I agree with you that both have advantages and disadvantages - but I believe it shouldn't be mentioned in the paper.
> >
> > In general - I believe the proposed architecture makes a lot of sense! My only concern was with the presentation and choice of baselines.

---

> ### Author Response · Authors · 2025-03-24
> **Thank you very much for your constructive feedback.**
>
> Dear Reviewer,
>
> Thank you once again for your constructive feedback.
>
> We have revised the paper following your recommendations, with the modifications highlighted in blue. Specifically, we have expanded the discussion on the under-utilization and incompatibility issues of MuZero in Section 3.1. Additionally, we have incorporated MuZero w/ History Encoder as a baseline in Section 3.1 and Table 1. However, we observed that MuZero w/ History Encoder (stack=1) currently performs poorly on Pong and VisualMatch, which may be due to the absence of effective state regularization. After a thorough review, we currently do not find any obvious implementation bugs. We plan to update the curves in Figure 2 in a future revision of the paper.
>
> Furthermore, we have added a discussion on the covariate shift in UniZero vs. MuZero in Appendix E.3, titled "Covariate Shift in UniZero." A more in-depth investigation of covariate shift remains part of our future work.
>
> We hope these modifications strengthen the validity of our baselines and enhance the readability of the paper. We appreciate your valuable insights and hope our clarifications and revisions adequately address your concerns.
>
> Sincerely,
>
> The Authors of UniZero (Submission 3998)

---

> > ### Comment · Reviewer_ijGr · 2025-03-25
> > **Thank you**
> >
> > Dear Authors,
> >
> > Thank you, I appreciate the changes!

---

> > > ### Author Response · Authors · 2025-03-25
> > > **Thank you for the insightful discussion.**
> > >
> > > Dear Reviewer,
> > >
> > > We sincerely appreciate the insightful discussion with you, which has been immensely helpful in enhancing the quality of our manuscript. As the deadline for the formal decision recommendation approaches, we would like to kindly ask whether you would be willing to update your recommendation based on our clarifications and revisions. If you still have any concerns, we would be grateful if you could point them out so that we can further improve our work.
> > >
> > > Sincerely,
> > >
> > > The Authors of UniZero (Submission 3998)

---

> > > > ### Comment · Reviewer_ijGr · 2025-03-25
> > > >
> > > > Yes, I have added a short "EDIT" to my original review.
> > > > Overall, I still believe that the paper would be stronger if presented differently, but I'm not objecting to acceptance anymore.
> > > >
> > > > I would kindly ask you to try to make the new baseline "MuZero w/ History Encoder" work - but if it indeed does not work, that's also a valuable result.
> > > >
> > > > Thank you for the extensive discussion.

---

> > > > > ### Author Response · Authors · 2025-03-26
> > > > > **Thank you**
> > > > >
> > > > > Dear Reviewer,
> > > > >
> > > > > Thank you for your recognition. We will continue to refine the "MuZero w/ History Encoder" baseline, enhance the clarity of relevant phrasing, and incorporate these improvements into the revised version of the paper. Once again, we sincerely appreciate your insightful suggestions.
> > > > >
> > > > > Sincerely,
> > > > >
> > > > > The Authors of UniZero (Submission 3998)

---

### Review · Reviewer_4J8R · 2025-02-18

**Summary Of Contributions:**

This paper proposes a new algorithm UniZero to improve MuZero style architectures. The authors identify limitations of the previously established architecture for MuZero and consider what can be gained by incorporating in Transformers for POMDP environments. UniZero compares favorably to MuZero in partially observable settings and experiences bigger gains as the environment becomes more non-Markovian with respect to the observations while maintaining performance in Markovian environments. The authors also take a deeper look into important design choices in this setting.

**Audience:**

Yes

**Broader Impact Concerns:**

There is no broader impact statement, but I do not believe it is necessary for this work.

**Claims And Evidence:**

Yes

**Requested Changes:**

Below I said the claims made in the submission are supported when forced to choose yes or no. This is not currently totally true, but I believe that the authors can makes some reasonable changes that would make it a fully true statement:

- Important for substantiating claims: Figure 13 should be updated with comparisons to MuZero and to include normalized mean scores.
- Important for substantiating claims: The authors should provide empirical evidence directly supporting their claim regarding intrinsic entanglement. Is this just a fancy way of making a standard argument for dynamic attention as opposed to recurrent processing?
- The authors should provide a detailed look at the computational efficiency of UniZero vs. MuZero. This should be incorporated into the discussion of scaling properties.
- The authors should provide a direct justification for why UniZero helps in multi-task settings specifically. Or is the argument the same as in the MDP setting i.e. that UniZero doesn't hurt? Their should also be a more detailed comparison with the results with out task specific heads.
- For the pong experiments, UniZero -SSL nd UniZero -context should be provided as additional ablations.
- The authors should be very explicit about the differences between UniZero and Sampled MuZero / Stochastic MuZero / EfficientZero, GumbelMuZero / LightZero.
- The discussion of world models should be updated to accurately reflect the prior literature before Ha & Schmidhuber (2018).
- The authors should be more clear about what the blue and red coloring implies in and around Equation 3. Does this carry a consistent meaning to the use of the same colors in Equation 2 and Table 1?

Comment Post Revisions: After having gone through the author's response to my concerns and revisions to the document, I believe the authors have done a very good job addressing my concerns and incorporating in the suggested revisions.

**Strengths And Weaknesses:**

Strengths
- The novelty is not very high, but I do think that exploration of the incorporation of Transformers into MuZero is a topic of interest to the TMLR community. As a result, it seems like a good fit with the venue.
- The authors perform a good amount of experiments and seem to show consistent improvements or performance matching with MuZero.
- The results seem very stark as the environment becomes more non-Markovian as in the right side of Figure 2 and Figure 4. This isn't that deep of an insight, but it does makes a lot of sense.  It is also nice to see it does not hurt in Markovian environments.

Weaknesses
- I thought the discussion around the limitations of MuZero could be a lot clearer. It comes across as proposing some vague intuitions, which aren't always validated. For example, the authors write "the recurrent design introduces an intrinsic entanglement between latent representations and historical information, resulting in a bottleneck that impedes efficient information propagation". Can this particular perspective about entanglement be directly measured or validated?
- The authors also do not engage with the computational efficiency that likely motivated this original design choice in MuZero. It is unclear to what degree the amount of computation and parameters are properly accounted for between models in experiments. My assumption is that UniZero is much more expensive computationally in the experiments provided. The authors also write "the architecture suffers from an under-utilization of trajectory data during training, which restricts its ability to fully exploit the accumulated experiential data" which again seems to imply more compute will be used for UniZero. I just think the authors should be much more clear and straightforward about this tradeoff. The authors are motivated by a setting where they want to maximize sample efficiency without regard for compute per step whereas MuZero was trying to computationally efficiently do better in self-play scenarios like Go and Chess. In their setting UniZero would have gotten through fewer games with same computational budget, which could have resulted in being less practical given their set of concerns.
- The authors do a good job of motivating why UniZero should help for POMDPs, but I was not really clear on what the benefits should be in the multi-task scenario. There should be more discussion of this. It appears like the authors just thought it was an interesting setting (which I agree with) but there is not really a clear expectation on what UniZero uniquely brings to this setting more than any other.
- I found the Stack = 1 pong experiments very compelling on first glance, but something didn't sit well with me about the idea that every ablation does so much worse than UniZero. I understand this better now looking at Table 1. What I would be interested to see is UniZero -context or UniZero -SSL rather than MuZero +context or MuZero +SSL. Because the usage of the RNN makes such a big difference on its own, we cannot understand the impact of these other ablations in the partially observable context.
- The related work was a real weak point for me and I believe it must be revised before this paper can be published.  The authors mention relevant baselines in Sampled MuZero, Stochastic MuZero, EfficientZero, GumbelMuZero, and LightZero, but the connections between these works and UniZero are not clearly stated. Maybe something akin to Table 1 would be appropriate here? The authors also say "The concept of world models, first proposed in Ha & Schmidhuber (2018),..." this statement is obviously incorrect.   Ha and Schmidhuber 2018 is a weird reference to consider the first as model based RL existed in the literature for decades prior, maybe it can be considered the first "modern" model (not even totally sure about this), but the current framing seems to lack scholarly rigor. If you read their paper, you will see that they make no such claims and cite many much older references.
- There are also some seemingly incorrect statements made when presenting the results in Figure 13. The authors write "The results in Figure 13, reveal that a single model trained under a online multi-task setting achieves normalized mean scores comparable to those obtained from single-task training", but the normalized mean score is never actually reported. The authors also write "Table 2 and Figure 13 demonstrate that UniZero (MT) outperforms both UniZero (ST) and MuZero (MT) in terms of normalized mean and median scores across the evaluated environments within the 400K Env Steps setting." This statement appears unsubstantiated in the case of Figure 13 as MuZero is not compared against and the normalized mean scores are not provided.
- For multi-task learning I felt the use of independent decision and dynamics heads wasn't really justified. Did the authors try the model without this and find that it did not work? I am concerned that the separation we see in Figure 12 is just a direct result of using these different task specific heads, requiring the use of differently synced representation spaces for the different tasks to accommodate the different heads. I believe the observation spaces are sufficiently different for these tasks such that there should be no intrinsic need to have separate heads with a sufficiently powerful monolithic neural network.

---

> ### Author Response · Authors · 2025-03-03
> **Author response to reviewer 4J8R (1/2)**
>
> Thank you very much for your valuable time and detailed review comments. Below are our responses to your questions:
>
> **On the separation of independent decision and dynamics heads and the phenomenon in Figure 12**
> - In the process of multi-task learning, we adopted an independent head design, primarily based on previous research findings that suggest independent heads help reduce gradient interference [1]. Except for the decision heads specific to each task, all tasks share the same encoder and Transformer backbone, enabling representation sharing and unified latent dynamics modeling. The separation phenomenon shown in Figure 12 to some extent validates this hypothesis. It should be noted that we had not previously conducted experimental evaluations for the shared head approach.
> - To further verify the impact of the shared head, we conducted experiments with UniZero under a shared head setting on eight Atari games (with the head structure identical to that in the independent head setting for each game). The experimental results indicate that using a shared head leads to an overall performance drop of approximately 50% compared to using independent heads for each game. Therefore, although a single powerful head should theoretically be capable of handling these tasks, achieving this goal often requires more complex designs due to gradient conflict issues. In the future, we plan to explore the performance upper bound of the shared head approach by increasing network capacity, improving network architecture, or introducing gradient correction techniques. Preliminary conclusions on this matter have been added to the appendix of the paper.
>
> **Advantages of UniZero in multi-task learning**
> - Leveraging the modular encoder-backbone-head architecture, UniZero can flexibly adapt to different observation spaces and action spaces. For instance, we have validated its applicability across vector-input vs. observation-input settings, as well as discrete vs. continuous action spaces.
> - Benefiting from the strong sequential modeling capability of Transformers, UniZero effectively handles dynamics heterogeneity across tasks and maintains stable performance in POMDP and other non-Markovian environments.
> - By sharing the encoder and Transformer backbone, UniZero enables cross-task representation learning and latent dynamics modeling, improving sample efficiency and reducing the computational overhead of redundant learning. The T-SNE results in Figure 12 provide further empirical support for this hypothesis.
> - Additionally, the learned model can be efficiently fine-tuned for new tasks, which is a key direction for our future work.
>
> **Can the concept of "entanglement" be directly measured or verified?**
> - Overall, our approach revisits the issue of "entanglement" in the latent state during the propagation of historical information in MuZero and proposes an improved architecture. Notably, this issue cannot be resolved solely by introducing an attention mechanism; instead, we address it by adopting a modular encoder-backbone-head structure and a parallel training paradigm. During training, the encoder at each step receives only the current step's information, while the handling of historical information is delegated to the (Transformer) backbone. Through this decoupled training paradigm, we achieve more efficient learning across various heterogeneous environments.
> - To verify the validity of our proposed "intrinsic entanglement" hypothesis, we further analyze the changes in the latent state of MuZero after training for 100k iterations in Pong. Specifically, we set unroll step = 5 and use the Average Latent Cosine Similarity between different unroll steps and the Pearson Correlation of consecutive unroll steps within the same minibatch as quantitative metrics to measure the degree of "entanglement." Experimental results indicate that for both the original MuZero and MuZero w/ SSL, the overall Cosine Similarity and Pearson Correlation are close to 1, and the Pearson Correlation further increases as the unroll step grows. This suggests that during recurrent training, there is a high correlation between latent states, partially confirming the presence of the "intrinsic entanglement" phenomenon.
>
>  (1/2)

---

> ### Author Response · Authors · 2025-03-03
> **Author response to reviewer 4J8R (2/2)**
>
> **Discussion on Figure 13 results**
> - Upon review, we identified an indexing error in the statement: "Table 2 and Figure 13 demonstrate that UniZero (MT) outperforms both UniZero (ST) and MuZero (MT) in terms of normalized mean and median scores across the evaluated environments within the 400K Env Steps setting." The correct figure should be Figure 10. Table 2 and Figure 10 primarily compare the performance of UniZero (MT) and MuZero (MT) across 8 Atari games. We have corrected this in the revised version.
> - Since MuZero (MT) already performed poorly on the 8 tasks, we previously did not test MuZero (MT) on all 26 tasks to conserve computational resources. Instead, we directly compared the performance of UniZero (MT) and UniZero (ST) in Figure 13.
> - To enhance the completeness of our experiments, we conducted additional tests on MuZero (MT) across all 26 tasks (where the network capacity was expanded fourfold compared to the single-task version). The experimental results aligned with expectations—MuZero (MT) exhibited poor performance and failed to achieve significant progress in almost all tasks. Based on your suggestion, we have included a supplementary table in the appendix of the revised paper, comparing the normalized scores of UniZero/MuZero across MT/ST settings for all 26 tasks, providing a more comprehensive evaluation. Additionally, we briefly analyzed the primary reasons for MuZero (MT)’s poor performance in multi-task learning across 26 tasks. We hypothesize that the high dynamics diversity across these tasks makes it challenging for the recurrent structure to effectively adapt, leading to severe task conflicts and ultimately hindering learning progress.
>
>  **Additional ablation study for Figure 2**
> - Regarding UniZero-SSL, our algorithm design currently does not support this experiment because UniZero requires autoregressive prediction of the latent state during inference. Therefore, it must explicitly use the dynamics head to output the next latent state.
> - As for UniZero-context, this essentially corresponds to setting $H_{\text{infer}} = 1$ in the current UniZero implementation, meaning that the model's performance is evaluated with almost no contextual information. In POMDP environments, removing historical information theoretically leads to a performance drop. In MDP environments, the impact is relatively smaller. For example, in the DMC experiments, our default setting is $H_{\text{infer}} = 2$, under which the overall performance has already surpassed DreamerV3. In our preliminary experiments, we also tested $H_{\text{infer}} = 1$ and observed similar performance. Therefore, we did not include this ablation study in Figure 2. However, if necessary, we will add the relevant ablation experiments in future versions.
>
> **On computational efficiency**
> - As you noted, UniZero is designed to balance sample efficiency across POMDP, multi-task, and heterogeneous environments. Compared to MuZero, while UniZero may have slightly higher per-step computation, its different training paradigm (parallel training instead of MuZero’s recurrent for-loop) ensures that the actual time difference is negligible.
> - For example, under 100k env steps, UniZero training takes about 4 hours, which is comparable to the previously reported training time of MuZero. It is important to note that this result was measured on a single GPU without special computational optimizations. We have now implemented a multi-GPU version, and experiments show that its speedup is nearly linear. Furthermore, we plan to explore existing Transformer optimization techniques to further improve computational efficiency. A detailed discussion on this topic has been included in Appendix B.5: "Computational Cost".
>
> **Related work section**
> - We have corrected the description of Ha & Schmidhuber (2018) to explicitly acknowledge how the concept of World Models builds on decades of model-based RL research.
> - Additionally, we have expanded Appendix F to include detailed discussions on Sampled MuZero, Stochastic MuZero, EfficientZero, GumbelMuZero, and LightZero, comparing their differences and similarities with UniZero.
>
>  **Color annotations in figures**
> - In Equations (2) and (3), we use blue markings to indicate parts related to the dynamics head and red markings for parts related to the decision head. In Table 1, we originally used blue markings to highlight the UniZero method. However, we acknowledge that this color usage might cause ambiguity. To avoid confusion, we will remove the relevant color markings in the revised version.
>
> We hope this addresses your concerns. If you have any further suggestions, we would be very grateful for your feedback. Thank you very much.
>
> [1] Aviral Kumar, Rishabh Agarwal, Xinyang Geng, George Tucker, and Sergey Levine. Offline q-learning on
> diverse multi-task data both scales and generalizes. arXiv preprint arXiv:2211.15144, 2022.
>
> (2/2)

---

> > ### Author Response · Authors · 2025-03-07
> > **Author response to reviewer 4J8R**
> >
> > Dear Reviewer,
> >
> > As the discussion phase is coming to an end, we kindly invite you to review our responses to your comments. If you have any further questions, please don’t hesitate to reach out. If no further concerns remain, we would greatly appreciate your consideration in updating your recommendation.
> >
> > Best regards,
> >
> > Authors of UniZero (Submission 3998)

---

> > > ### Comment · Reviewer_4J8R · 2025-03-10
> > >
> > > Thank you for the comprehensive reply to my review. After having gone through the author's response to my concerns and revisions to the document, I believe the authors have done a very good job addressing my concerns and incorporating in the suggested revisions. I have also updated my review to indicate this.

---

> > > > ### Author Response · Authors · 2025-03-10
> > > > **Author response to reviewer 4J8R**
> > > >
> > > > Dear Reviewer,
> > > >
> > > > Thank you very much for recognizing the modifications we have made. We truly appreciate your updated recommendation and thoughtful evaluation, which have greatly encouraged us. Your insightful suggestions have been invaluable in refining our work, and we are deeply grateful for the time you took to review our submission.
> > > >
> > > > Best regards,
> > > >
> > > > Authors of UniZero (submission 3998)

---

### Review · Reviewer_cD6a · 2025-02-24

**Summary Of Contributions:**

UniZero introduces a transformer-based latent world model integrated with Monte Carlo Tree Search (MCTS) to address limitations of MuZero-style algorithms in heterogeneous environments. Key innovations include:

 - Disentangled Latent Representation: Separates latent states from implicit history using a transformer backbone, enabling full trajectory utilization during training.
- Joint Optimization: Combines policy, value, reward, and latent dynamics prediction into a single loss function.
- Scalable Architecture: Employs domain-specific encoders and a shared transformer for multitask learning.
- Efficient Planning: Uses a Key-Value (KV) cache for long-term memory and MCTS in latent space.

**Audience:**

Yes

**Claims And Evidence:**

Yes

**Requested Changes:**

While claims are generally supported by the evidence, I request authors to temper down claims about single-task performance and scalability. I have mentioned the reasons in the weakness section (For instance, underperforming MuZero for single-task RL in ~42% of the games).

**Strengths And Weaknesses:**

### Strengths

- Outperforms MuZero and SAC-GPT on VisualMatch, achieving >80% success rate at memory length=60 vs. MuZero’s ~20%. Transformer architecture mitigates recurrent model limitations (e.g., error accumulation in MuZero w/ Context).
-  Single-model training on 26 Atari games matches single-task performance (Figure 13), with mean normalized score of 0.455 vs. 0.322 for single-task UniZero (Table 2). Figure 12 show task-specific clustering, indicating effective shared representation learning.
- Matches MuZero in Atari 100K (Figure 5) and surpasses DreamerV3 in DMControl (Table 3: UniZero mean=787.2 vs. DreamerV3=743.7).
- SimNorm (latent normalization) proves critical for stability (Figure 7), outperforming Softmax/Sigmoid. No performance drop when omitting observation reconstruction losses, aligning with "decision-relevant latent encoding" hypothesis.
- Authors have discussed limitations of the work which is highly appreciated.

### Weaknesses

- Method has modest scalability in multitask learning, only 13% mean improvement over single-task suggests limited cross-task synergy. Also, method fails to outperform single-task UniZero in 7/26 Atari games (e.g., Breakout: 8 vs. MuZero's 48 [Table 10]).

- The method is claimed to be competitive in standard single-task RL, however, the method Underperforms MuZero in 11/26 Atari games (e.g., RoadRunner: 1100 vs. MuZero's 4400 [Table 10]). Fails short-horizon tasks like Pong (-10 vs. MuZero's -7 [Table 10]).

- Authors suggest that SimNorm is critical for stability, however they don't make comparisons with other norm methods like LayerNorm. The ablation only has the experiment with and without SimNorm.

### Suggestions

- Authors can adopt rotary or relative positional embeddings to enhance temporal modeling.
- It will be interesting if authors can devise a strategy to adjust  $H_{infer}$ automatically per task.

---

> ### Author Response · Authors · 2025-03-03
> **Author response to reviewer cD6a (1/2)**
>
> Thank you very much for your valuable time and detailed review comments. Below are our responses to your questions:
>
> **Regarding Atari Single-Task Performance**
> - We emphasize that the comparative results in Table 10 were obtained at 100k environment steps. Due to the larger capacity and unique network structure of the Transformer, UniZero may require more training data and gradient updates to fully realize its potential. As shown in Figure 14, at 200k environment steps, UniZero's performance is already approaching that of MuZero (e.g., around 8 points in Pong and approximately 5000 points in RoadRunner).
> - Furthermore, the current version of UniZero has not yet been fully fine-tuned in terms of network architecture and hyperparameters. We believe that with careful optimization, it is possible to further narrow the gap with MuZero at 100k environment steps.
> - Based on the existing experimental results, we have revised the corresponding descriptions in the paper regarding single-task performance to ensure a neutral tone consistent with the experimental data. Thank you for your valuable suggestions!
>
> **Regarding Scalability**
> - Compared to MuZero, which employs a recurrent mechanism, UniZero adopts a decoupled architecture, allowing it to directly benefit from the scalability of Transformers (a characteristic widely validated in NLP and CV fields). The 13% performance improvement in UniZero’s multi-task learning over single-task learning may be attributed to the significant variations among different Atari tasks. However, we believe that the performance gain from multi-task over single-task learning represents only one aspect of "scalability." Scalability also encompasses UniZero’s performance dynamics as model size increases and training task diversity expands.
> - Figure 11 illustrates UniZero’s scalability as model size grows, while Figure 13 demonstrates its ability to scale to the full Atari100 benchmark (26 tasks in total). Additionally, our latest supplementary experiments show that MuZero fails entirely in multi-task learning across these 26 tasks. We have included relevant analyses in the appendix and believe that with the introduction of larger-scale data and system optimizations, UniZero’s synergistic performance in multi-task learning will further improve. Moreover, we have adjusted the wording regarding scalability as per your suggestions.
>
> **Regarding Rotary or Relative Positional Encoding**
> - In early experiments, we tested the RoPE relative positional encoding technique. Experimental results indicated that under the current experimental settings, RoPE performs comparably to learnable absolute positional encoding. Our preliminary analysis suggests that in decision-making tasks like Atari, temporal dependencies are usually short and fixed. Compared to the NLP domain, the advantages of relative position encoding may not be as significant. Consequently, we chose learnable absolute positional encoding for the main experiments. However, we will retain the RoPE option in the final open-source code and plan to further explore its performance in scenarios with complex temporal dependencies in future research.
>
> **Regarding SimNorm**
> - The concept of SimNorm was first introduced in the paper [1], with its core idea being to impose constraints on the representation space during the pretraining phase, introducing an inductive bias that promotes group sparsity in representations. Theoretical analysis indicates that compared to unnormalized representations, those processed with SimNorm achieve superior generalization performance. TD-MPC2 [2] incorporated SimNorm into the field of RL and demonstrated a similar effect. Therefore, we initially only compared ungrouped Softmax (i.e., SimNorm with V=768) and Sigmoid, without testing other normalization methods.
> - Currently, we have tested LayerNorm on the Pong single-task setting, and the results indicate comparable performance. SimNorm enhances the attribute-specificity and generalization capability of latent state representations through its within-group Softmax mechanism, which may facilitate subsequent interpretation of specific attributes and multi-task fine-tuning. Therefore, we adopt SimNorm by default, though LayerNorm is also a viable alternative. The relevant analysis has been added to the ablation section in the appendix.
>
> (1/2)

---

> > ### Author Response · Authors · 2025-03-03
> > **Author response to reviewer cD6a (2/2)**
> >
> > **Regarding Adaptive Adjustment of H_{infer}**
> > - Thank you for your suggestion—we completely agree with your perspective. Adapting $H_{\text{infer}}$ based on environmental differences is indeed a highly valuable research direction. In MDP tasks, a short context is usually sufficient, whereas tasks with long-term dependencies require a longer context window. Let valid_context_length represent the effective context length required for a given task. A feasible approach is as follows: during the training phase, we can use mini-batches of varying lengths and indirectly infer the required context length by analyzing the reward prediction error and TD error. The intuition behind this idea is that as valid_context_length increases, the sequence length required for loss reduction also grows accordingly. During the testing phase, we can adaptively adjust valid_context_length in the inference process based on feedback from training errors. We plan to further explore this direction in future work.
> >
> > We hope this addresses your concerns. If you have any further suggestions, we would be very grateful for your feedback. Thank you very much.
> >
> > [1] Lavoie, Samuel, et al. "Simplicial embeddings in self-supervised learning and downstream classification." arXiv preprint arXiv:2204.00616 (2022).
> > [2] Hansen, Nicklas, Hao Su, and Xiaolong Wang. "Td-mpc2: Scalable, robust world models for continuous control." arXiv preprint arXiv:2310.16828 (2023).
> >
> > (2/2)

---

> > ### Comment · Reviewer_cD6a · 2025-03-06
> > **Some comments.**
> >
> > I thank authors for adopting a neutral tone, especially for single-task performance.  I don't have anymore queries and have updated my recommendation.

---

> > > ### Author Response · Authors · 2025-03-07
> > > **Author response to reviewer cD6a**
> > >
> > > Dear Reviewer,
> > >
> > > Thank you for acknowledging our work. We are very pleased that you recognized the changes we made to our tone. Your updated recommendation and thoughtful evaluation are greatly appreciated. Once again, thank you for your valuable suggestions!
> > >
> > > Best regards,
> > >
> > > Authors of UniZero (submission 3998)

---

### Author Response · Authors · 2025-03-03
**General response and appreciation to reviewers**

We sincerely appreciate all reviewers for their constructive feedback and valuable suggestions. Based on your comments, we have revised our manuscript, and a summary of the changes is provided below. Additionally, we have addressed each of your comments individually.

Below, we summarize the main revisions made to the paper. These changes have been highlighted in blue in the new version:
- Following reviewer cD6a’s suggestion, we refined the description of single-task and multi-task performance to ensure more objective wording that aligns with the experimental data.
- Following reviewer cD6a’s suggestion, we added a comparative analysis between SimNorm and LayerNorm, which has been included in Appendix E.1.
- Following reviewer 4J8R’s suggestion, we added an analysis on whether to use independent heads (Appendix C.1). Theoretically, a sufficiently strong single head should be able to handle multiple tasks. However, due to gradient conflict issues, achieving this typically requires a more complex design. Therefore, we have adopted an independent head structure.
- Following reviewer 4J8R’s suggestion, we included a table comparing the normalized scores of UniZero/MuZero and MT/ST across 26 tasks in the appendix of the revised paper (Appendix C.3, Table 10) to provide a more comprehensive evaluation. The results indicate that UniZero’s multi-task model performs comparably to the single-task trained UniZero in terms of normalized human scores, whereas MuZero fails to scale effectively in large-scale task scenarios (Appendix C.3).
- Following reviewer 4J8R’s suggestion, we added a brief discussion on the computational efficiency of UniZero and MuZero in Appendix B.5, "Computational Cost."
- Following reviewer 4J8R’s suggestion, we revised the description of World Models and updated the relevant references. Additionally, in Appendix F, we provided explanations of baseline methods such as Sampled MuZero, Stochastic MuZero, EfficientZero, and GumbelMuZero, along with a detailed comparative analysis of UniZero and these methods.
- Following reviewer 4J8R’s suggestion, to avoid confusion, we removed the related color markings in the revised version.
- Following reviewer ijGr’s suggestion, we clarified certain expressions to improve readability.

We sincerely appreciate all reviewers for their valuable feedback. This revision addresses most of the comments, but we are happy to make further adjustments if the reviewers have additional suggestions.

Best,

Authors of UniZero (submission 3998)

---

### Decision · Action_Editor_18rx · 2025-04-10

**Recommendation:** Accept as is

**Comment:**

The paper proposes a new method to address weaknesses of the MuZero agent. The transformer-based world model is a better fit for POMDP environments. Initially reviewers criticized the presentation of the paper, motivation and lack of experimental results to support the claims. Authors provided additional experiments, rearranged the paper and added significant content through rebuttal phase. All reviewers agreed that concerns are addressed and the work is ready for publication. I recommend authors to make the code public for reproducibility.

**Audience:**

The paper explores transformer architecture for MuZero and provides detailed analysis and ablation study. The TMLR audience can benefit from the findings of this work.

**Claims And Evidence:**

The paper presents UniZero, a transformer-based architecture with Monte Carlo Tree Search to improve MuZero. Initially certain claims such as efficient planning and disentangled latent representation were not well-supported. However, through very long rebuttal discussion, authors added significant content and analysis, providing clear evidence for claims. Reviewers' concerns have largely been addressed.